# Focus and Dilution: The Multi-stage Learning Process of Attention

**Zheng-An Chen** [* 1]   **Pengxiao Lin** [* 1 2]   **Zhi-Qin John Xu** [1 2 3 4]   **Tao Luo** [1 2 3 5]

## Abstract

Transformer-based models have achieved remarkable success across a wide range of domains, yet our understanding of their training dynamics remains limited. In this work, we identify a recurrent focus–dilution cycle in attention learning and provide a rigorous explanation in a one-layer Transformer setting for Markovian data via gradient-flow analysis. Using stage-wise linearization around critical points, we show that a single focus–dilution cycle can be decomposed into a sequence of distinct stages. First, embedding and projection rapidly condense to a rank-one structure, while attention parameters remain effectively frozen. Then, the attention parameters begin to increase, inducing a frequency-driven focus toward high-frequency tokens. As attention continues to evolve, it generates next-order perturbations in embeddings, leading to a mass-redistribution mechanism that progressively dilutes this focus. Finally, small asymmetries among low-frequency tokens lift a degenerate critical point, opening new embedding directions and initiating the next cycle. Experiments on synthetic Markovian data as well as WikiText and TinyStories corroborate the predicted stages and cyclical dynamics.

## 1. Introduction

Transformer models (Vaswani et al., 2017) have become the dominant architecture for sequence modeling. While their approximation power is now well understood in a variety of

regimes (Pérez et al., 2019; Yun et al., 2020a;b), we still lack a mechanistic theory for how attention itself evolves during training. Most existing analyses gain tractability by introducing additional technical conditions, such as reparameterizations (Zhang et al., 2024a) or proxy dynamics (Tarzanagh et al., 2023), which may obscure the native coupling among embeddings, projection, and attention. Moreover, recent work suggests that Transformer training often undergoes multiple stages (Chang et al., 2024; Varre et al., 2025), and that attention can shift from highly concentrated to more diffuse patterns (Tian et al., 2024). These observations point to a need for a dynamical picture that remains faithful to the coupled dynamics and can explain both attention amplification and its subsequent dissipation.

In this work, we combine theory and experiments to show that attention can be understood as a cyclical learning process. Within each cycle, attention first amplifies a frequency-driven preference over tokens (*focus*), and then gradually weakens this preference as the embedding structure adapts (*dilution*). We identify the dynamical origin of each stage and explain how the interaction between embeddings and attention progressively decomposes the learning problem.

To keep the analysis tractable while preserving essential sequential structure, we study population gradient flow for a one-layer Transformer trained by cross-entropy on Markov data (Chang et al., 2024; Makkuva et al., 2024; 2025). Our explanation is stage-wise and built on linearizations around critical points. Under small initialization, the trajectory is first governed by the linearization near the origin, which forces a rank-one condensation of the embedding and projection components, consistent with the condensation phenomenon in Chen & Luo (2025). We further observed that the condensed direction is explicitly determined by the stationary distribution. In contrast, the attention parameters remain small in this initial stage because the leading-order driving term for $(W_Q, W_K)$ vanishes at the origin.

After condensation, the trajectory follows the same low-rank ray until it reaches a second critical point. We show that this point is generically a saddle: the Jacobian admits a local block decomposition into (i) a contracting embedding/output subsystem and (ii) an attention subsystem with a single unstable mode. Consequently, once the trajectory enters this neighborhood, $(W_Q, W_K)$ align with the unsta-

---
[*]Equal contribution . Code is available at Transformer Training Dynamics.   [1]School of Mathematical Sciences, Shanghai Jiao Tong University. [2]Institute of Natural Sciences, Shanghai Jiao Tong University. [3]MOE-LSC, Shanghai Jiao Tong University. [4]Shanghai Seres Information Technology Co., Ltd, Shanghai 200040, China. [5]CMA-Shanghai, Shanghai Jiao Tong University. Correspondence to: Zhi-Qin John Xu <xuzhiqin@sjtu.edu.cn>, Tao Luo <luotao41@sjtu.edu.cn>.

*Proceedings of the 43rd International Conference on Machine Learning*, Seoul, South Korea. PMLR 306, 2026. Copyright 2026 by the author(s).

ble eigendirection and grow exponentially, and attention acquires a bias toward high-frequency tokens, initiating the focus phase.

Going beyond the focus phase requires a more refined description than the local saddle analysis. Once $(W_Q, W_K)$ align with the unstable direction, dynamics enters to a rank-one invariant manifold and induces a closed reduced system. This reduced flow exposes a mass-redistribution mechanism in the embeddings. As the attention amplitude evolves, it generates next-order perturbations, causing the embeddings of the main token and the remaining tokens to move in opposite directions. As a result, the earlier high-frequency focus is gradually weakened, leading to an attention dilution phase.

Finally, the model must learn new embedding directions that distinguish low-frequency tokens. However, we show that the training dynamics become trapped at a degenerate critical point on the rank-one manifold, where the driving forces vanish and no new directions can emerge. To model realistic asymmetries and eliminate degeneration, we introduce a small symmetry-breaking perturbation among low-frequency tokens and analyze the resulting bifurcation of critical points. This mechanism explains how new embedding directions are unlocked, thereby initiating the next focus–dilution cycle. Experiments on synthetic Markovian data as well as WikiText and TinyStories corroborate the predicted stages and cyclical dynamics.

**Our contributions.**

1. We identify a focus–dilution cycle in the training dynamics of attention and introduce a minimal tractable setting that captures this phenomenon.

2. We develop a stage-wise analysis based on linearization at critical points that explains the different stages within single cycle.

3. We empirically validate the predicted stages and transitions, demonstrating that the focus–dilution cycle persists on synthetic Markov data as well as realistic data.

## 2. Related Works

**Training dynamics of attention and multi-stage analysis** Studying the training dynamics of attention remains a challenging problem. A common practice is to introduce various simplifications to the problem, such as constructing task-specific synthetic data, utilizing reparameterization or using simplified model and target function (Sheen et al., 2024; Kim & Suzuki, 2024; Varre et al., 2023; Chen et al., 2024a; Wu et al., 2025; Gao et al., 2024; Zhang et al., 2025a; Vasudeva et al., 2025; Yang et al., 2025). Among these, (Lu

et al., 2021) establishes key dynamical identities using a controllable text classification task. (Snell et al., 2021) suggests that models first capture word co-occurrence before adjusting attention to focus on relevant tokens. (Li et al., 2023) examines the dynamical effects of fixing specific attention components within a topic-word task framework. Following these previous works, (Tian et al., 2024) proposed a novel mathematical framework for analyzing the joint dynamics of MLP and attention blocks, successfully explaining the sparsity of attention score matrices. Recent works further study sparse attention emergence and dynamical separation in broader settings (Zucchet et al., 2025; Chen & Luo, 2025). Meanwhile, multi-stage dynamics have also been studied. For feedforward-style models, (Wang & Ma, 2023; Chen et al., 2024c) characterize the multi-stage training dynamics of two-layer networks, while (Xu et al., 2025a) studies cross-stage dynamics in LoRA. For Transformers, (Varre et al., 2025) explains plateau-to-drop dynamics in an analyzable in-context $n$-gram setting. However, the aforementioned literature either relies on data settings that deviate significantly from real-world scenarios or requires overly stringent analytical conditions.

**Transformers on Markov chains** A significant body of influential work employs Markov chains to understand how Transformers, as probabilistic models, learn continuous linguistic data. (Chang et al., 2024) discovers that LLM learning can be summarized as "early n-gram learning followed by the gradual refinement of low-probability (tail) n-gram predictions." (Bietti et al., 2023) analyzes the formation mechanism of induction heads using Markov-like data. (Rajaraman et al., 2024) investigates the impact of tokenization on Markovian data, proving that appropriate tokenization assists Transformers in modeling Markov processes. Additionally, (Makkuva et al., 2024) and (Makkuva et al., 2025) explore training dynamics and convergence analysis specifically under Markovian data settings.

**Small initialization** The initialization of a neural network significantly affects its learning outcomes (Arora et al., 2019; Williams et al., 2019; Mei et al., 2018; Jacot et al., 2018; Rotskoff & Vanden-Eijnden, 2018; Zhang et al., 2020). Small initialization is a common setting investigated in the study of neural network optimization dynamics, which contrasts with the Neural Tangent Kernel (NTK) perspective prevalent in infinitely wide networks. For linear models, (Ji & Telgarsky, 2019) theoretically establish results regarding matrix alignment. For nonlinear models, (Zhou et al., 2022) found that small initialization similarly promotes parameter condensation, thereby reducing model complexity. Theoretically, (Luo et al., 2021; Chen et al., 2024b; Zhou et al., 2025; Kumar & Haupt, 2025) have further deepened the understanding of this phenomenon. A recent survey article (Xu et al., 2025b) systematically synthesizes these empirical

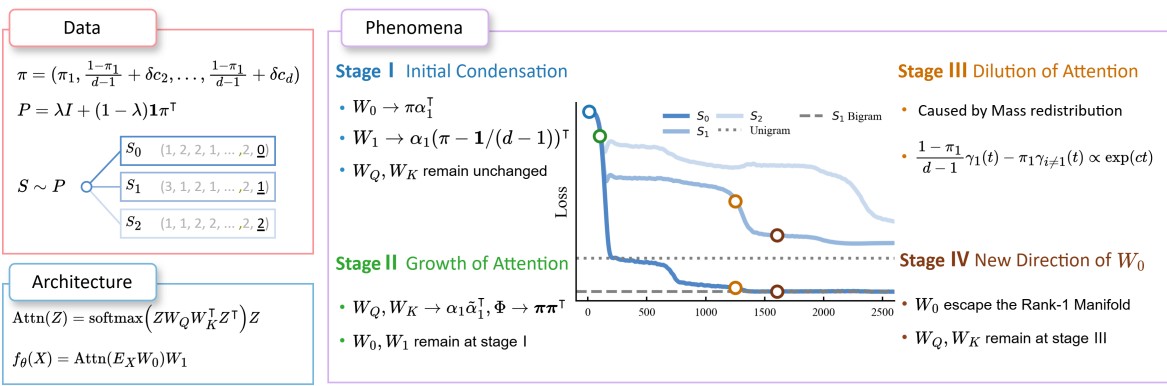

*Figure 1.* **Overview of the setting and the focus–dilution training pattern.** (*Left*) Sequences are generated by a Markov chain with stationary distribution $\pi$. We extract dataset $S_0, S_1, S_2$ from the training set, which differ only in the identity of the final token in each sequence. (*Right*) The loss curves exhibit four stages: initial condensation, attention growth, attention dilution, and the emergence of a new direction.

## 3. Preliminaries

### 3.1. Basic Notations.

For any $N \in \mathbb{N}$, let $[N] := \{1, \ldots, N\}$. Let $\mathcal{V} := [d]$ be the vocabulary set with $d \geq 2$. We identify tokens with indices in $[d]$ and write $\{e_i\}_{i=1}^d$ for the canonical basis of $\mathbb{R}^d$. For $\alpha \in \mathbb{R}^n$ and $A \in \mathbb{R}^{n \times n}$, $\|\alpha\|_2$ and $\|A\|_F$ denote Euclidean norm and Frobenius norm separately, with the subscript omitted when clear from context. We write $\|\alpha\|_C := \sqrt{\alpha^\mathsf{T} C \alpha}$ for the seminorm induced by positive semidefinite matrix $C \succeq 0$. For $\alpha \in \mathbb{R}^n$, define the variance matrix $\mathrm{Var}(\alpha) := \mathrm{diag}(\alpha) - \alpha\alpha^\mathsf{T}$.

### 3.2. Markov data generation

We generate the dataset $\mathcal{D} = \{(X_i, y_i)\}_{i=1}^N$ with $X_i = (x_{i,1}, \ldots, x_{i,s}) \in \mathcal{V}^s$ and $y_i := x_{i,s+1}$, by a Markov chain.

**Definition 3.1** (Markovian data). Let $P \in \mathbb{R}^{d \times d}$ be row-stochastic. For each $i \in [N]$, sample $x_{i,1} \sim \mathrm{Unif}(\mathcal{V})$ and $x_{i,j} \sim P_{x_{i,j-1}}$ for $j = 2, \ldots, s+1$. Set $X_i = (x_{i,1}, \ldots, x_{i,s})$ and $y_i = x_{i,s+1}$.

To model one high-frequency token together with a group of low-frequency tokens that may exhibit mild heterogeneity, we consider a stationary distribution of the form $\pi^\mathsf{T}(\delta) := (\pi_1, \ldots, \pi_d)$ with

$$\pi_i = \frac{1 - \pi_1}{d - 1} + c_i \delta, \quad \forall\, 2 \leq i \leq d, \tag{1}$$

where $\frac{1-\pi_1}{d-1} < \pi_1 < 1$, $\sum_{i=2}^d c_i = 0$, and $\delta \geq 0$ is a small parameter chosen so that $\pi(\delta)$ remains entrywise nonnegative. The first term describes two-group setting, one high-frequency token and the rest symmetrical low-frequency tokens. The second term is an $O(\delta)$ perturbation that breaks

symmetry within the low-frequency group. Unless stated otherwise, we treat the first term as the leading-order component and regard the second term as a small perturbation that can be neglected in early-stage analyses.

The transition matrix is defined as

$$P = \lambda I + (1 - \lambda)\mathbf{1}\pi^\mathsf{T}, \qquad 0 < \lambda < 1, \tag{2}$$

where $\mathbf{1} \in \mathbb{R}^d$ is the all-ones vector. A direct computation verifies that $\pi^\mathsf{T} P = \pi^\mathsf{T}$, hence $\pi$ is stationary for $P$.

### 3.3. One-layer transformer

Since the next token depends only on the current token under the Markov assumption, a single attention block is sufficient to capture the relevant dependency. We therefore study a one-layer Transformer and its training dynamics.

**Definition 3.2** (One-layer transformer). Given input sequence $X = (x_1, \ldots, x_s)$, let $E_X = (e_{x_1}, \ldots, e_{x_s})^\mathsf{T} \in \mathbb{R}^{s \times d}$. Let $W_0 \in \mathbb{R}^{d \times m}$ be the embedding matrix and define the embedded sequence $E_X W_0 \in \mathbb{R}^{s \times m}$. For any $Z \in \mathbb{R}^{s \times m}$, the attention block is

$$\mathrm{Attn}(Z) = \mathrm{softmax}\!\left(Z W_Q W_K^\mathsf{T} Z^\mathsf{T}\right) Z.$$

Let $W_1 \in \mathbb{R}^{m \times d}$ be the output projection. The output logits are

$$f_\theta(X) = \mathrm{Attn}(E_X W_0) W_1$$

For notational convenience, we define $W_{QK} := W_Q W_K^\mathsf{T}$, $\Phi := W_0 W_Q W_K^\mathsf{T} W_0^\mathsf{T}$ and $M := W_0 W_1$.

### 3.4. Training objective and gradient-flow dynamics

Given $(X_i, y_i) \in \mathcal{D}$, define the cross-entropy at the last token $\ell\left(f_\theta(X_i)_s, y_i\right) := -\log \frac{\exp\left(f_\theta(X_i)_{s,y_i}\right)}{\sum_{j=1}^d \exp(f_\theta(X_i)_{s,j})}$. Then

$$\mathcal{L}(\theta) = \frac{1}{N} \sum_{i=1}^N \ell\left(f_\theta(X_i)_s, y_i\right). \tag{3}$$

We study the gradient flow $\dot{\theta} = -\nabla\mathcal{L}(\theta)$.

**Proposition 3.3** (Gradient flow and population-gradient limit). *The gradient flow dynamics satisfy*

$$\begin{cases} \dfrac{dW_0}{dt} = -\dfrac{\partial\mathcal{L}}{\partial M}W_1^\mathsf{T} - \dfrac{\partial\mathcal{L}}{\partial\Phi}W_0 W_{QK}^\mathsf{T} - \left(\dfrac{\partial\mathcal{L}}{\partial\Phi}\right)^\mathsf{T} W_0 W_{QK}, \\ \dfrac{dW_1}{dt} = -W_0^\mathsf{T}\dfrac{\partial\mathcal{L}}{\partial M}, \\ \dfrac{dW_Q}{dt} = -W_0^\mathsf{T}\dfrac{\partial\mathcal{L}}{\partial\Phi}W_0 W_K, \\ \dfrac{dW_K}{dt} = -W_0^\mathsf{T}\left(\dfrac{\partial\mathcal{L}}{\partial\Phi}\right)^\mathsf{T} W_0 W_Q. \end{cases} \tag{4}$$

*Moreover, define the token-level proxy attention matrix $\mathbb{A} \in \mathbb{R}^{d\times d}$ by $\mathbb{A}_{i,j} = \frac{\pi_j \exp(e_i^\mathsf{T}\Phi e_j)}{\sum_{j'} \pi_{j'}\exp(e_i^\mathsf{T}\Phi e_{j'})}$ and the model output distribution $\mathbb{P} \in \mathbb{R}^{d\times d}$ by $\mathbb{P}_{i,j} = \frac{\exp(\mathbb{A}_i M e_j)}{\sum_{j'}\exp(\mathbb{A}_i M e_{j'})}$, where $\mathbb{A}_i$ and $\mathbb{P}_i$ denote the i-th row.*

*Then, in the large sample-size and long-context limit $(N, s) \to \infty$, the empirical gradients converge to*

$$\lim_{N,s\to\infty} \frac{\partial\mathcal{L}}{\partial M} = -\sum_{i=1}^d \pi_i \mathbb{A}_i^\mathsf{T}\left(P_i - \mathbb{P}_i\right),$$
$$\lim_{N,s\to\infty} \frac{\partial\mathcal{L}}{\partial\Phi} = -\sum_{i=1}^d \pi_i e_i\left(P_i - \mathbb{P}_i\right)M^\mathsf{T}\mathrm{Var}(\mathbb{A}_i). \tag{5}$$

## 4. Theoretical results

### 4.1. Idea: stage-wise linearization around saddle points

Under small initialization, attention training often exhibits a multi-stage pattern: the trajectory spends a long time near a low-dimensional structure and then abruptly departs in a new direction. We explain this behavior via a stage-wise analysis around successive critical points. At each stage, the parameters enter a neighborhood of a saddle point where the gradient flow is well-approximated by its linearization. The linearized dynamics exposes (i) stable directions that keep the trajectory confined to a low-dimensional manifold, and (ii) unstable directions that eventually dominate and trigger the transition to the next stage.

Concretely, we consider the gradient flow $\dot{\theta} = -\nabla\mathcal{L}(\theta)$. Let $\theta_*$ be a critical point, and define $\Delta\theta := \theta - \theta_*$. A

Taylor expansion yields

$$\frac{d}{dt}\Delta\theta = -\nabla^2\mathcal{L}(\theta_*)\Delta\theta + \text{higher-order terms.}$$

The next lemma characterizes the linearization in which the nonlinear flow is governed by the linearized system, and formalizes the alignment with the most unstable direction.

**Lemma 4.1** (Linearization near a saddle point). *Let $\dot{\theta} = F(\theta)$ be an ODE with $F \in C^2$, and let $\theta_*$ satisfy $F(\theta_*) = 0$. Let $J := DF(\theta_*)$ and assume there exist $r > 0$ and $L > 0$ such that for all $\|\Delta\theta\| \leq r$,*

$$\|F(\theta_* + \Delta\theta) - J\Delta\theta\| \leq L\|\Delta\theta\|^2. \tag{6}$$

*Let $\theta(t)$ be the solution with $\|\Delta\theta(0)\| = \varepsilon \leq r/2$, and $\tilde{\Delta}\theta(t) := e^{Jt}\Delta\theta(0)$ be the solution of the linearized system $\dot{\tilde{\Delta}}\theta = J\tilde{\Delta}\theta$. Define $\mu := \sup\{\Re(\lambda) : \lambda \in \sigma(J)\}$. Then for all $t$ such that $\|\tilde{\Delta}\theta(t)\| \leq r/2$,*

$$\|\Delta\theta(t) - \tilde{\Delta}\theta(t)\| \leq C\varepsilon^2 e^{2\mu t} \tag{7}$$

*for some constant $C = C(J, L)$. In particular, if $\mu > 0$, then the nonlinear dynamics is well-approximated by the linearized dynamics up to times $t = \Theta(\log(1/\varepsilon))$.*

*Moreover, suppose $J$ is symmetric and has a simple eigenvalue $\mu > 0$ with eigenvector $v_u$ and a spectral gap $\rho > 0$ in the sense that $\lambda \leq \mu - \rho$ for all $\lambda \in \sigma(J) \setminus \{\mu\}$. Then for any initialization with $\langle\Delta\theta(0), v_u\rangle \neq 0$,*

$$\frac{\Delta\theta(t)}{\|\Delta\theta(t)\|} \to \pm\frac{v_u}{\|v_u\|} \tag{8}$$

*for any sequence $t = t(\varepsilon)$ with $t(\varepsilon) \to \infty$ and $\varepsilon e^{\mu t(\varepsilon)} \to 0$.*

At initialization, each entry of every parameter matrix is sampled i.i.d. from $\mathcal{N}(0, \varepsilon^2)$ with $\varepsilon \ll 1$. Thus $\theta(0)$ lies in an $\mathcal{O}(\varepsilon)$-neighborhood of the origin, which is a critical point of the gradient flow. By Lemma 4.1, the dynamics in the early time window of length $\Theta(\log(1/\varepsilon))$ is governed by the linearization at $\theta = 0$. A key consequence is that the linearized system admits a single unstable direction, so trajectories rapidly align with a rank-one direction. In our setting, this direction is not arbitrary: it is explicitly pinned down by the stationary distribution $\pi$ of the underlying token Markov chain.

**Theorem 4.2** (Initial condensation (rephrased from Thm. 2 in (Chen & Luo, 2025))). *The origin is a critical point and*

$$\frac{\partial\mathcal{L}}{\partial M}\bigg|_{\theta=0} = -\pi\left(\pi - \frac{1}{d}\mathbf{1}\right)^\mathsf{T}, \qquad \frac{\partial\mathcal{L}}{\partial\Phi}\bigg|_{\theta=0} = 0. \tag{9}$$

*The effective dynamics near $\theta = 0$ is*

$$\frac{d\Delta W_0}{dt} = -\frac{\partial\mathcal{L}}{\partial M}\bigg|_{\theta=0}\Delta W_1^\mathsf{T}, \frac{d\Delta W_1}{dt} = -\Delta W_0^\mathsf{T}\frac{\partial\mathcal{L}}{\partial M}\bigg|_{\theta=0} \tag{10}$$

*Consequently, there exist a vector $\alpha_1$ such that the following limit holds as $\varepsilon \to 0$ at $t = \Theta(\log \frac{1}{\varepsilon})$:*

$$\frac{W_0}{\|W_0\|} \to \frac{\pi}{\|\pi\|}\alpha_1^\intercal, \quad \frac{W_1}{\|W_1\|} \to \alpha_1 \frac{\pi^\intercal - \frac{1}{d}\mathbf{1}^\intercal}{\|\pi^\intercal - \frac{1}{d}\mathbf{1}^\intercal\|}. \quad (11)$$

Theorem 4.2 characterizes the first stage of training dynamics in our model. Although it is rephrased from Thm. 2 in (Chen & Luo, 2025), we emphasize a more concrete interpretation relevant to data. As a result, $(W_0, W_1)$ rapidly condense onto a $\pi$-driven rank-one structure within time $T_1 = \Theta(\log(1/\varepsilon))$. In contrast, the attention block $(W_Q, W_K)$ stays $\mathcal{O}(\varepsilon)$ throughout this stage because the linear term in its dynamics vanishes at the origin, i.e., $\partial \mathcal{L}/\partial \Phi|_{\theta=0} = 0$.

### 4.2. Focus of Attention

After initial condensation stage, outer parameters $(W_0, W_1)$ rapidly become approximately rank-one, while the attention parameters $(W_Q, W_K)$ remain $O(\varepsilon)$. Empirically, the trajectory then stays close to the rank-one condensation ray where the outer parameters evolve along the same direction until it enters a neighborhood of a second critical point.

**Proposition 4.3** (Existence of a second critical point on the condensation ray). *Assume $\pi_1 > \pi_2 = \cdots = \pi_d$. Then there exists $\kappa_1 > 0$ such that the parameter tuple $\theta_c^1$*

$$W_0 = \kappa_1 \frac{\pi}{\|\pi\|}\alpha_1^\intercal, W_1 = \kappa_1 \alpha_1 \frac{\pi^\intercal - \frac{1}{d}\mathbf{1}^\intercal}{\|\pi - \frac{1}{d}\mathbf{1}\|}, W_Q, W_K = 0,$$
$$(12)$$

*satisfies $\mathbb{P}_i = \pi^\intercal$ for all $i$ and $\frac{\partial \mathcal{L}}{\partial M}\big|_{\theta=\theta_c^1} = 0$. Moreover, $\theta_c^1$ is a critical point of the full gradient flow.*

The key point is that $\theta_c^1$ is typically a saddle: the $(W_0, W_1)$-subsystem is contracting (or neutrally stable due to symmetry), while the $(W_Q, W_K)$-subsystem contains an unstable mode.

**Proposition 4.4** (Linearized dynamics and its unique unstable direction). *At critical point $\theta_c^1$ defined in Prop. 4.3, the linearization of the gradient flow admits the block form*

$$\frac{\mathrm{d}}{\mathrm{d}t}\begin{pmatrix} \Delta W_0 \\ \Delta W_1 \\ \Delta W_Q \\ \Delta W_K \end{pmatrix} = \begin{pmatrix} J_{\text{out}} & 0 \\ 0 & J_{\text{att}} \end{pmatrix}\begin{pmatrix} \Delta W_0 \\ \Delta W_1 \\ \Delta W_Q \\ \Delta W_K \end{pmatrix} \quad (13)$$

*where $J_{\text{out}}$ is negative semi-definite and $J_{\text{att}}$ is positive semi-definite. It indicates that the current dynamics are dominated by the attention subsystem. In particular, the attention block satisfies the explicit closed system*

$$\frac{\mathrm{d}}{\mathrm{d}t}\Delta W_Q = c\,\alpha_1 \alpha_1^\intercal \Delta W_K, \quad \frac{\mathrm{d}}{\mathrm{d}t}\Delta W_K = c\,\alpha_1 \alpha_1^\intercal \Delta W_Q,$$
$$(14)$$

*where $c = \lambda \kappa_1^4 \frac{\|\pi\|_{\text{Var}(\pi)}^4}{\|\pi - \frac{1}{d}\mathbf{1}\|\|\pi\|^3} > 0$. Therefore the attention block has an exponentially unstable mode.*

By Lemma 4.1, once the trajectory enters an $O(\varepsilon)$-neighborhood of $\theta_c^1$, the dynamics is governed by the linearization for a duration $\Theta(\log(1/\varepsilon))$. It implies that the attention parameters $(W_Q, W_K)$ converge into the direction depending on the condensation direction. As a result, the attention structure prioritizes tokens that appear frequently in the steady-state distribution, indicating that the attention mechanism has become specific.

**Theorem 4.5** (High frequency token bias). *Suppose the trajectory enters an $O(\varepsilon)$-neighborhood of $\theta_c^1$. Within the linearization neighborhood of Lemma 4.1, there exists a unit vector $\tilde{\alpha}_1 \in \mathbb{R}^m$ such that, for generic small initialization of $(W_Q, W_K)$,*

$$\frac{W_Q(t)}{\|W_Q(t)\|} \to \alpha_1 \tilde{\alpha}_1^\intercal, \quad \frac{W_K(t)}{\|W_K(t)\|} \to \alpha_1 \tilde{\alpha}_1^\intercal. \quad (15)$$

*Consequently, along the unstable ray $W_Q = W_K = \kappa \alpha_1 \tilde{\alpha}_1^\intercal$, the attention score matrix satisfies*

$$\frac{\Phi}{\|\Phi\|} = \frac{W_0 W_Q W_K^\intercal W_0^\intercal}{\|W_0 W_Q W_K^\intercal W_0^\intercal\|} = \pi\pi^\intercal. \quad (16)$$

*Then for each $i \in [d]$ the attention distribution exhibits a high-frequency bias:*

$$\lim_{\kappa \to \infty} \mathbb{A}_i(W_0, W_1, \kappa\alpha_1\tilde{\alpha}_1^\intercal, \kappa\alpha_1\tilde{\alpha}_1^\intercal) = e_1^\intercal. \quad (17)$$

### 4.3. Dilution of Attention

Sec. 4.2 shows that the second critical point $\theta_c^1$ is a saddle whose unique unstable direction lies in the attention subsystem: after a transient of length $\Theta(\log(1/\varepsilon))$, the attention parameters become approximately rank-1 and aligned while the outer parameters remain close to their initial values on the condensation ray. In this subsection, we stay in the same neighborhood of $\theta_c^1$ but go beyond linearization: Conditioned on the rank-1 manifold, we resolve the next-order perturbations in embeddings induced by the evolution of attention. This refinement reveals a redistribution effect in the embeddings that gradually undermines the previously formed focus, leading to the dilution phase.

Motivated by the alignment result in Sec. 4.2, we model the post-transient phase by the rank-one parametrization

$$W_0 = \gamma(t)\,\alpha_1^\intercal, \quad W_1 = \alpha_1\,\beta(t)^\intercal,$$
$$W_Q = \lambda_Q(t)\,\alpha_1\,\tilde{\alpha}_1^\intercal, \quad W_K = \lambda_K(t)\,\alpha_1\,\tilde{\alpha}_1^\intercal, \quad (18)$$

where $\gamma(t), \beta(t) \in \mathbb{R}^d$ and $\lambda_Q(t), \lambda_K(t) \in \mathbb{R}$. Meanwhile, at the entry time $t_0$ of this phase, we assume

$$\gamma(t_0) = \kappa_1 \frac{\pi}{\|\pi\|}, \quad \beta(t_0) = \kappa_1 \frac{\pi - \frac{1}{d}\mathbf{1}}{\|\pi - \frac{1}{d}\mathbf{1}\|},$$
$$\lambda_Q(t_0) = \lambda_K(t_0) = o(1). \quad (19)$$

For notational convenience, we also define the attention amplitude $\eta(t) := \lambda_Q(t)\lambda_K(t)$ which is the only combination that enters the reduced dynamics below.

**Proposition 4.6** (Invariant rank-one manifold). *Assume $\pi_1 > \pi_2 = \cdots = \pi_d$. Define*

$$\mathcal{W} := \left\{ \theta \text{ satisfying (18) for some } (\gamma, \beta, \lambda_Q, \lambda_K) \right\}.$$

*If $(W_0, W_1, W_Q, W_K) \in \mathcal{W}$ at time $t_0$, then the gradient flow (4) remains in $\mathcal{W}$ for all $t \geq t_0$. Moreover, if $\gamma_2(t_0) = \cdots = \gamma_d(t_0)$ and $\beta_2(t_0) = \cdots = \beta_d(t_0)$, it will be preserved for any $t \geq t_0$.*

Restricting the gradient flow to $\mathcal{W}$ yields a closed system in $(\gamma, \beta, \eta)$:

$$
\begin{aligned}
\dot{\gamma} &= -\frac{\partial \mathcal{L}}{\partial M}\beta - \eta\Big(\frac{\partial \mathcal{L}}{\partial \Phi} + \Big(\frac{\partial \mathcal{L}}{\partial \Phi}\Big)^{\mathsf{T}}\Big)\gamma, \\
\dot{\beta} &= -\Big(\frac{\partial \mathcal{L}}{\partial M}\Big)^{\mathsf{T}}\gamma, \quad \dot{\eta} = -2\eta\,\gamma^{\mathsf{T}}\frac{\partial \mathcal{L}}{\partial \Phi}\gamma.
\end{aligned}
\tag{20}
$$

By Proposition 4.6, it suffices to track two-group coordinates

$$
\begin{aligned}
\gamma_1, \quad \gamma_2 &= \cdots = \gamma_{|\mathcal{V}|} := \gamma_{i \neq 1}, \\
\beta_1, \quad \beta_2 &= \cdots = \beta_{|\mathcal{V}|} := \beta_{i \neq 1},
\end{aligned}
$$

and denote $\Delta\gamma := \gamma_1 - \gamma_{i \neq 1}$, $\Delta\beta := \beta_1 - \beta_{i \neq 1}$. Intuitively, this reduces the post-alignment dynamics to an effective two-group system (token 1 versus all others). Importantly, we are *still* analyzing the flow near the same critical point $\theta_c^1$; the difference from Sec. 4.2 is that we can keep the attention direction fixed and resolve the next-order feedback that governs redistribution on the rank-one manifold.

**Theorem 4.7** (Mass redistribution). *Consider the linearization of reduced dynamics (20) on $\mathcal{W}$ at critical point corresponding to $\theta_c^1$. There exists $c > 0$ such that*

$$(1 - \pi_1)\gamma_1(t) - (d-1)\pi_1\gamma_{i \neq 1}(t) \propto \exp(ct). \tag{21}$$

*Consequently, $\gamma_1(t)$ and $\gamma_{i \neq 1}(t)$ cannot move in the same direction: a weighted contrast between high-frequency token and the remaining tokens is exponentially amplified.*

Theorem 4.7 explains the mechanism behind the dilution phase. After alignment, the attention direction is essentially fixed, and the attention amplitude $\eta(t)$ feeds back into $\dot{\gamma}$ through the $\eta(\partial \mathcal{L}/\partial \Phi)\gamma$ term in (20). The redistribution effect forces a growing separation between $\gamma_1$ and $\gamma_{i \neq 1}$, so the embedding mass cannot remain concentrated along $\pi$.

Therefore, the logit difference that causes high-frequency bias gradually weakens: the attention weights corresponding to low-frequency tokens are no longer concentrated on high-frequency tokens. This marks a shift from focus to dilution.

## 4.4. Emergence of a new direction via data asymmetry

In Sec. 4.3, the dynamics collapses onto a rank-one invariant manifold, effectively reducing learning to a "token 1 vs. all others" two-group system. Further learning requires separating low-frequency states, which demands growth of embeddings along directions that distinguish low-frequency tokens.

However, for perfectly symmetric data among low-frequency tokens, the rank-one manifold may contain a degenerate critical point where both driving forces vanish: $\partial \mathcal{L}/\partial M = 0$ and $\partial \mathcal{L}/\partial \Phi = 0$. Crucially, the degeneracy is not only tangential, but also transverse. As a consequence, linearization does not generate a mechanism that pushes the trajectory away from the rank-one manifold.

**Proposition 4.8** (Degenerate critical point). *Assume perfect symmetry among low-frequency tokens. On the rank-one invariant manifold $\mathcal{W}$ (Proposition 4.6), there exists a critical point, which is a neutrally stable equilibrium for the linearized dynamics, such that $\frac{\partial \mathcal{L}}{\partial M} = 0$ and $\frac{\partial \mathcal{L}}{\partial \Phi} = 0$.*

The solution to remove degeneracy is to introduce perturbations that breaks the symmetry. In practice, low-frequency tokens rarely have identical frequencies. To model a minimal asymmetry while keeping calculations simple, we focus on $d = 3$ and perturb the stationary distribution by a small parameter $\delta$:

$$\pi^{\mathsf{T}} = (\pi_1, \tfrac{1-\pi_1}{2}, \tfrac{1-\pi_1}{2}) \Rightarrow \tilde{\pi}^{\mathsf{T}} = (\pi_1, \tfrac{1-\pi_1}{2}+\delta, \tfrac{1-\pi_1}{2}-\delta)$$

We study stationary points of the perturbed gradient field $-\nabla_\theta \mathcal{L}(\theta, \delta) = 0$ near the degenerate critical point. After shifting coordinates so that $\theta = 0$ corresponds to the degenerate critical point, a formal expansion takes the form

$$-\nabla \mathcal{L}(\theta, \delta) = J_0 \theta + \delta f_1 + \text{h.o.t.}, \tag{22}$$

where $J_0 = -\nabla_\theta^2 \mathcal{L}$ at $\delta = 0$.

If $J_0$ were invertible, the implicit function theorem would apply, and we could directly obtain the solution $\theta(\delta)$. Unfortunately, due to symmetry, $J_0$ is degenerate, so we use the standard Lyapunov–Schmidt reduction. Let $Q_K = (k_1, \ldots, k_{d_K})$ and $Q_R = (q_1, \ldots, q_{d_R})$ be orthonormal bases for the kernel and range subspaces of $J_0$:

$$\mathbb{R}^p = \text{Ker}(J_0) \oplus \text{Ran}(J_0), \qquad \theta = Q_K x + Q_R y,$$

where $p$ is the parameter dimension. Projecting the stationarity condition onto the range and kernel yields the equivalent system

$$-Q_R^{\mathsf{T}}\nabla \mathcal{L}(\theta, \delta) = 0, \quad -Q_K^{\mathsf{T}}\nabla \mathcal{L}(\theta, \delta) = 0. \tag{23}$$

The range equation can be solved by the implicit function theorem since $Q_R^{\mathsf{T}} J_0 Q_R$ is invertible, yielding a smooth

map $y = \zeta(x, \delta)$. Substituting back into the kernel equation produces a reduced low-dimensional problem in $x$ whose solutions describe nearby stationary points.

The key effect of the perturbation is that it splits the previously flat transverse directions. A genuinely transverse positive eigenvalue of order $\Theta(\delta)$ appears, while tangential instability is at most $\mathcal{O}(\delta^2)$. This fast transverse instability is what drives the trajectory away from the rank-one manifold and seeds a new embedding direction.

**Theorem 4.9** (Asymmetry lifts degeneracy and induces a new direction). *Consider the perturbed stationary distribution with parameter $\delta$ above. There exists a point $\theta(\delta)$ near the degenerate critical point such that*

$$\|\nabla_\theta \mathcal{L}(\theta(\delta), \delta)\| = O(\delta^3). \tag{24}$$

*Moreover, the Hessian at $\theta(\delta)$ exhibits two distinct scales:*

*1. Slow tangential instability. Any positive eigenvalues created from the previously degenerate directions are at most $O(\delta^2)$.*

*2. Fast normal instability. Under mild condition, in directions transverse to the rank-one manifold, there exists a positive eigenvalue of order $\Theta(\delta)$.*

Theorem 4.9 shows that any generic low-frequency asymmetry lifts this degeneracy and produces a fast transverse unstable mode of size $\Theta(\delta)$. Once the trajectory enters the neighborhood of $\theta(\delta)$, this transverse instability drives it away from the degenerate rank-one configuration and enables the emergence of a genuinely new embedding direction, allowing the model to further differentiate low-frequency tokens beyond the two-group description.

## 5. Empirical evidence

In this section, leveraging the simplified transformer model, we analyze the training behavior on Markovian data and empirically validate the theoretical derivation describing the transition of attention from focus to dilution. In parallel, we evaluate the model on real-world WikiText corpora and on the TinyStories corpus, which exhibits basic linguistic structure, to assess whether our observations generalize to the training dynamics of large-scale language models in realistic settings.

### 5.1. Synthetic Experiments

We construct synthetic datasets using four distinct transition matrices $P$, designed to share a common stationary distribution $\pi = (0.75, 0.19, 0.05, 0.01)$. To reveal the low-rank structure of parameters caused by condensation, we measured the cosine similarity between neuronal input weights for analysis (Chen & Luo, 2025; Xu et al., 2025b).

Additionally, we visualize the embedding trajectory evolution by applying Principal Component Analysis (PCA) to the concatenated embedding snapshots across all steps (Lorch, 2016; Antognini & Sohl-Dickstein, 2018). Detailed experimental setups are provided in Appendix D. The overall evolution of the training dynamics is visualized in Fig. 2. We identify four distinct stages during the training process. In the following, we provide a detailed analysis of each stage to demonstrate the consistency between our experimental observations and theoretical results.

**Stage I: Initial Condensation**  Our theoretical analysis predicts that during this stage, the outer layers $W_0, W_1$ evolve from an initialized full-rank state to a low-rank structure, while the inner attention parameters remain largely invariant. This is depicted by Fig. 2(A), which demonstrates that the outer weights rapidly evolve into rank-1, whereas the $W_Q, W_K$ maintain the high-rank nature of their initialization. Simultaneously, Fig. 2(B1) illustrates that the embeddings of all tokens evolve towards a uniform direction, further validating our theoretical analysis.

**Stage II: Growth of Attention**  During this stage, the outer parameters remain largely invariant, while $W_Q$ and $W_K$ transition into a condensed state. This phase coincides with a significant drop in training loss, marking the evolution of parameters from the origin to the next critical point. According to our theory, the attention mechanism evolves such that high-frequency tokens are gradually focused by the remaining tokens. This phenomenon is clearly visualized in Fig. 2(C).

**Stage III: Dilution of Attention**  In Stage III, although the parameters remain confined to the rank-1 manifold (evidenced by the unchanged condensation heatmap in Fig. 2(A)), Fig. 2(B2) reveals that all tokens, except for token 0, exhibit a retraction trajectory. This implies that while the training dynamics are strictly constrained within the low-rank manifold, the model begins to differentiate between tokens. As shown in Fig. 2(C, D), low-frequency tokens pay less attention to high-frequency token in this phase, accompanied by a significant drop in the embedding norms of low-frequency tokens. Consequently, outer parameters of the network revert to an unstable state.

**Stage IV: Emergence of New Direction**  In Stage IV, the accumulated instability drives the model to escape the constraints of the rank-1 manifold, initiated by the growth of new directions in the outer layers. This transition is clearly observable in Fig. 2(B3). To further quantify this, in Fig. 2(D3), we project the embeddings of low-frequency tokens ($W_0[1:]$) onto the direction orthogonal to token 0 (denoted as $W_0[0]_\perp$) and calculate the projection norms. The results indicate that, for the first time, the remaining

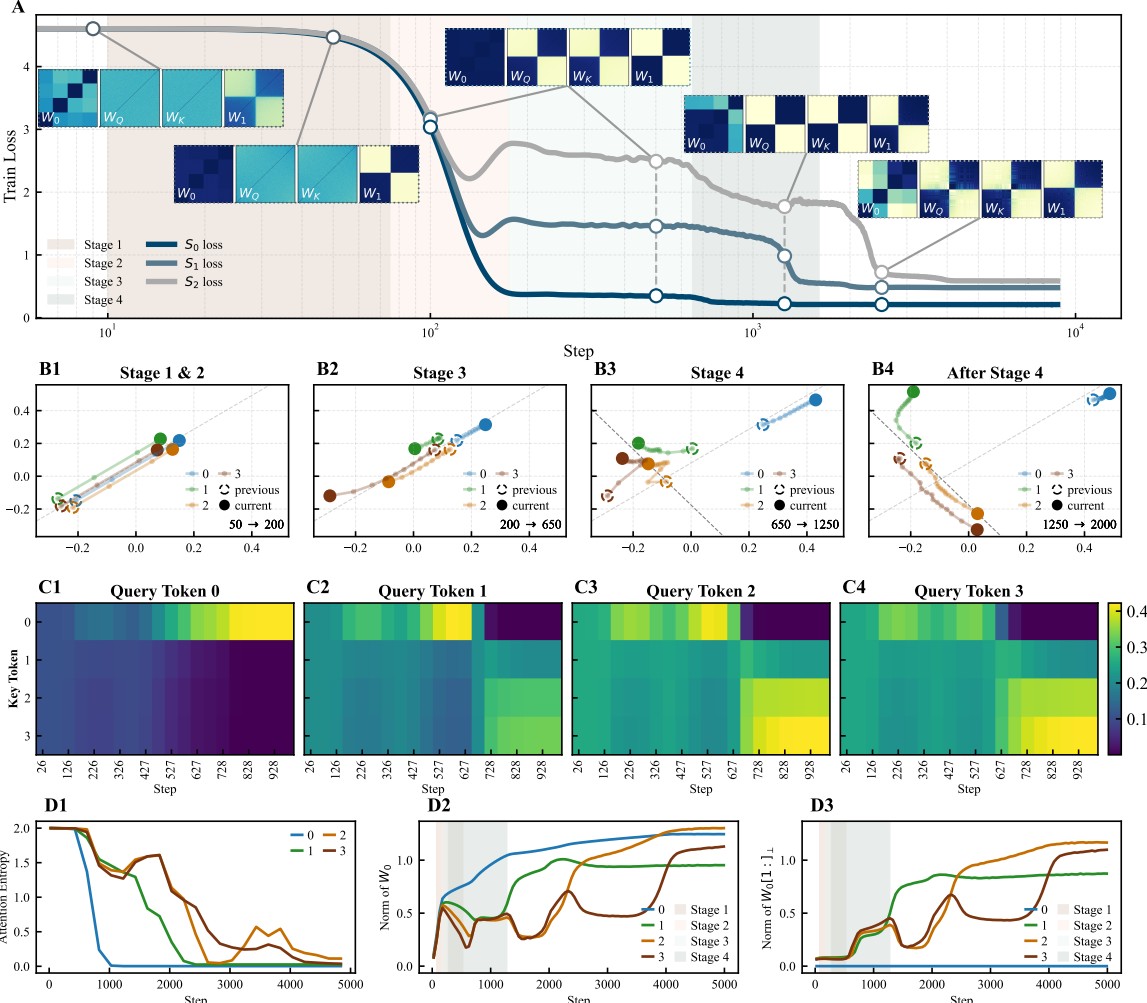

*Figure 2.* **Empirical focus–dilution cycle.** (A) Loss curves for $S_0, S_1, S_2$, accompanied by cosine-similarity matrices for $(W_0, W_Q, W_K, W_1)$. (B) PCA of embeddings reveals one-directional growth (Stage I & II), retraction during dilution (Stage III), and expansion into new directions (Stage IV). (C) Attention maps transition from focus to dilution. Between steps 350 and 550, the attention given to the first token by all query pairs increases synchronously. Afterward, the first token focuses attention on itself, while other tokens reduce their attention towards it. (D1) Attention entropy; (D2), (D3) The embedding norm and the norm of perpendicular component onto the first token.

token embeddings significantly deviate from the direction of token 0.

**After Stage IV: Subsequent Training Dynamics** Fig. 2(D) further illustrates the later stages of the training process, revealing a distinct periodicity in the embedding norms. Specifically, the focus and dilution pattern repeats recursively: as the network proceeds to learn Token 1, the remaining tokens (2–3) undergo the same retraction and regrowth process, continuing sequentially until training concludes. We hypothesize that after Stage IV, the model has effectively converged on Token 0. Consequently, the system evolves into a sub-dynamic regime governed by Tokens 1–3. In this reduced state, the parameter dynamics can be re-analyzed within our original theoretical framework.

### 5.2. Real-world Experiments

**Experimental Results.** We validate the correctness of our theorem on two real-world datasets: WikiText (Merity et al., 2016) and TinyStories (Eldan & Li, 2023). We employ the same simplified Transformer architecture and maintain hyperparameter settings consistent with the synthetic data experiments. To investigate the "focus-and-dilution" characteristics of the attention mechanism, we track the top-three most frequent tokens alongside three randomly sampled medium-frequency tokens (frequency $> 10\%$) from the training set.

As illustrated in the figure 3, across both WikiText and TinyStories, the attention mechanism exhibits a consistent

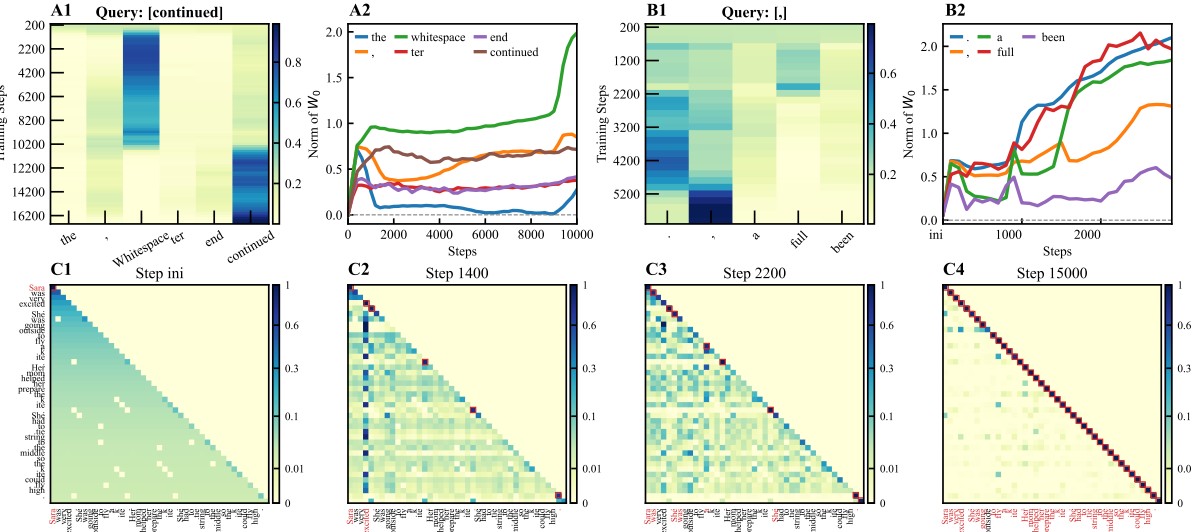

*Figure 3.* **Experimental results on real-world datasets.** (A) Results on WikiText. (A1) The attention evolution of the medium-frequency token 'continue' given the input sequence [the, *comma*, *whitespace*, ter, end, continue]. The attention scores exhibit a distinct four-phase transition: dilution → focus on the high-frequency whitespace → secondary dilution → final focus on continue itself. (A2) The evolution of $\|W_0\|_2$ for selected tokens. The dominant *whitespace* token shows continuous growth, while other tokens display a clear retraction. (B) Results on TinyStories. We observe similar dynamics using the input sequence [the, *comma*, a, full, been], mirroring the phenomena in (A). (C) Visualization of attention shifts. The evolution of attention for a single test sample across different training steps, with tokens exhibiting self-attention scores $\geq 0.75$ highlighted.

pattern: it initially prioritizes high-frequency tokens (e.g., "the", "a", and whitespace) before subsequently losing this focus—a process we term "dilution." Concurrently, by monitoring the embedding evolution of these selected tokens, we observe a distinct "retraction" phenomenon. Notably, due to the high variance in batch composition inherent to the 1-epoch training regime on real-world corpora, we occasionally observe this retraction even in the most frequent tokens.

**Validity of the Markov Approximation.** Existing studies (Chang & Bergen, 2022; Chang et al., 2024) indicate a curriculum in Transformer learning, starting from 1-gram to n-gram statistics. Our analysis of attention patterns in TinyStories supports this: distinct tokens gradually shift from uniform attention to self-attention. This behavior indicates that the model functions as a pseudo-2-gram model during early training phases, despite the non-Markovian nature of real text. These observations indirectly validate our experimental design, confirming that our synthetic Markov data acts as a suitable proxy for understanding real-world training dynamics.

**Extension to More Complex Architectures.** We conduct additional experiments to examine whether the focus–dilution and embedding retraction phenomena persist in more complex settings. Specifically, we consider two more realistic architectural variants: attention with residual connections and Transformers with multiple attention layers.

We find that residual connections do not qualitatively alter the overall training dynamics: as shown in Fig. 4, both the focus–dilution pattern and the evolution of embedding norms remain consistent with those observed in Figures 2(C,D). In the multilayer setting, although the presence of multiple attention modules modifies the detailed manifestation of the focus–dilution phenomenon, the embedding retraction effect remains evident, as shown in Fig. 5. Moreover, since multi-head attention may disrupt the low-rank dynamical structure analyzed in our theoretical setting, we further examine models with multi-head attention and find that embedding retraction still persists. Together, these results indicate that the proposed mechanism remains relevant beyond our simplified theoretical setting.

## 6. Conclusion and Limitations

This work provides a mechanistic account of transformer training dynamics by identifying a recurring focus–dilution cycle in attention learning. Through a stage-wise gradient-flow analysis, we offer a rigorous explanation for several empirically observed early-stage phenomena. Although derived in a simplified single-layer setting, the mechanism appears to extend beyond the idealized model. A key limitation is that the current analysis does not yet capture layer interactions, multi-head structure, or components such as LayerNorm; extending the framework to more realistic transformer architectures remains an important direction for future work.

## Impact Statement

This paper aims to advance the theoretical understanding of transformer training dynamics by providing a mechanistic analysis of attention evolution. While improved understanding of learning dynamics may inform future model design and training practices, we do not foresee any direct negative ethical or societal consequences arising specifically from this work.

## Acknowledgements

This work is sponsored by the National Key R&D Program of China Grant No. 2022YFA1008200 (T. L., Z. X.). We also thank Shanghai Institute for Mathematics and Interdisciplinary Sciences (SIMIS) for their financial support. This research was funded by SIMIS under grant number SIMISID-2025-ST (T. L.). The authors are grateful for the resources and facilities provided by SIMIS, which were essential for the completion of this work. This work is also sponsored by the National Natural Science Foundation of China Grant No. 92270001 (Z. X.), 12371511 (Z. X.), 12422119 (Z. X.), 2025 Key Technology R&D Program "New Generation Information Technology" Project 25511103100 of Shanghai Municipal Science and Technology Commission (Z. X.). This work is also supported by the Fundamental Research Funds for the Central Universities under Project No. YG2024ZD03 (Z. X.).

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

# A. Theoretical details in Sec. 3

## A.1. Property of markov process

We will use two standard asymptotic properties of Markov chains. For the sake of completeness, we provide a detailed proof.

**Proposition A.1** (Basic Markov properties). *Given the transition matrix $P$ in* (2) *and any initial distribution $\mu_0$:*

1. ***Convergence.*** *The marginal distribution converges to $\pi$. That is $\lim_{t \to \infty} \mu_0^\intercal P^t = \pi^\intercal$.*

2. ***Ergodicity.*** *Along a single trajectory, the empirical state frequencies converge to $\pi$. That is $\lim_{s \to \infty} \frac{1}{s} \sum_{j=1}^s \mathbb{1}_{x_j} = \pi^\intercal$, where $\mathbb{1}_{x_j} \in \mathbb{R}^d$ is the one-hot vector of token $x_j$.*

*Proof.* Throughout, we work on the finite state space $\mathcal{V} = \{1, \ldots, |\mathcal{V}|\}$. By the definition of the transition matrix $P$ defined in (2), $P$ is irreducible and aperiodic with strictly positive entries. Thus, $P$ is ergodic. In particular, $P$ admits a unique stationary distribution $\pi$ satisfying $\pi^\intercal P = \pi^\intercal$ and $\pi_i > 0$ for all $i$.

**1) Convergence of marginals.** Since $P$ is ergodic on a finite state space, it is primitive. By the Perron–Frobenius theorem, the eigenvalue 1 of $P$ is simple and all other eigenvalues satisfy $|\lambda| < 1$. Let $\mathbf{1} \in \mathbb{R}^{|\mathcal{V}|}$ denote the all-ones vector. Because $P$ is row-stochastic, we have $P\mathbf{1} = \mathbf{1}$; because $\pi$ is stationary, we have $\pi^\intercal P = \pi^\intercal$. Define the rank-one projector

$$\Pi := \mathbf{1}\,\pi^\intercal.$$

Then $\Pi^2 = \Pi$, $P\Pi = \Pi P = \Pi$, and we can write

$$P = \Pi + Q, \qquad \text{where } Q := P - \Pi.$$

Note that $Q\mathbf{1} = 0$ and $\pi^\intercal Q = 0$. Moreover, the spectrum of $Q$ equals the spectrum of $P$ with the eigenvalue 1 removed, hence its spectral radius satisfies $\rho(Q) < 1$. Therefore, $Q^t \to 0$ as $t \to \infty$ (in any matrix norm), and

$$P^t = (\Pi + Q)^t = \Pi + Q^t \xrightarrow[t \to \infty]{} \Pi = \mathbf{1}\,\pi^\intercal. \tag{25}$$

For any initial distribution $\mu_0$ (a row vector with nonnegative entries summing to 1),

$$\mu_0^\intercal P^t \xrightarrow[t \to \infty]{} \mu_0^\intercal (\mathbf{1}\,\pi^\intercal) = (\mu_0^\intercal \mathbf{1})\,\pi^\intercal = \pi^\intercal,$$

which proves the convergence claim.

**2) Ergodicity of empirical frequencies.** Let $(X_t)_{t \geq 0}$ be the Markov chain with transition matrix $P$ and arbitrary initial distribution $\mu_0$. Fix a reference state, say state 1, and define the (strict) return times

$$\tau_0 := 0, \qquad \tau_{k+1} := \inf\{t > \tau_k : X_t = 1\}, \qquad k \geq 0.$$

By irreducibility on a finite state space, the chain is positive recurrent, hence $\tau_k < \infty$ almost surely for all $k$ and $\mathbb{E}_1[\tau_1] < \infty$.

For each cycle $k \geq 0$, define the cycle length and the state-$i$ visit count within the cycle:

$$S_k := \tau_{k+1} - \tau_k, \qquad R_k(i) := \sum_{t=\tau_k}^{\tau_{k+1}-1} \mathbb{1}\{X_t = i\}.$$

By the strong Markov property, conditional on $X_{\tau_k} = 1$ the post-$\tau_k$ evolution is independent of the past, and therefore the pairs $\{(S_k, R_k(i))\}_{k \geq 1}$ are i.i.d. under $\mathbb{P}_1^{\text{MC}}$ (and also after the chain first hits state 1 when started from an arbitrary $\mu_0$). Let $N(T) := \max\{k : \tau_k \leq T\}$ be the number of completed cycles up to time $T$. Then for each fixed $i$,

$$\sum_{t=0}^{T-1} \mathbb{1}\{X_t = i\} = \underbrace{\sum_{t=0}^{\tau_1-1} \mathbb{1}\{X_t = i\}}_{\text{initial transient}} + \sum_{k=1}^{N(T)-1} R_k(i) + \underbrace{\sum_{t=\tau_{N(T)}}^{T-1} \mathbb{1}\{X_t = i\}}_{\text{remainder}}. \tag{26}$$

Divide by $T$. The initial transient term is $O(1/T)$ almost surely. The remainder term is at most one cycle, hence bounded by $S_{N(T)}$, and thus also negligible after dividing by $T$ because $\tau_{N(T)} \leq T < \tau_{N(T)+1}$ implies $S_{N(T)} \leq \tau_{N(T)+1}$ and $\tau_{N(T)} \to \infty$.

It remains to analyze the dominant sum over complete cycles. By the strong law of large numbers applied to the i.i.d. sequences $\{S_k\}$ and $\{R_k(i)\}$,

$$\frac{1}{n}\sum_{k=1}^{n} S_k \to \mathbb{E}_1[S_1] = \mathbb{E}_1[\tau_1], \qquad \frac{1}{n}\sum_{k=1}^{n} R_k(i) \to \mathbb{E}_1[R_1(i)] \quad \text{a.s.} \tag{27}$$

Moreover, by the definition of $\tau_k$, we know

$$\tau_{N(T)+1} = \sum_{k=0}^{N(T)} S_k \geq T, \quad \tau_{N(T)} = \sum_{k=0}^{N(T)-1} S_k \leq T-1$$

which implies

$$\frac{N(T)}{T} \to \frac{1}{\mathbb{E}_1[\tau_1]} \quad \text{a.s.}$$

Combining with (26) and (27), we obtain

$$\frac{1}{T}\sum_{t=0}^{T-1} \mathbb{1}\{X_t = i\} \xrightarrow[T\to\infty]{a.s.} \frac{\mathbb{E}_1[R_1(i)]}{\mathbb{E}_1[\tau_1]}. \tag{28}$$

Take $i = 1$, we get the right-hand side is $\pi_1$ by the computation about expectation of first return time. Since the above proof process is independent of the choice of the reference state, by considering all possible reference states, we obtain the result. $\qquad\square$

### A.2. Gradient-flow dynamics

In this section, we will supplement the proof details of Proposition 3.3.

*Proof.* We first derive Eq. (4), using the standard trace theorem and chain rule. Taking the total differential of the loss, we get

$$d\mathcal{L} = \left\langle \frac{\partial \mathcal{L}}{\partial M}, dM \right\rangle + \left\langle \frac{\partial \mathcal{L}}{\partial \Phi}, d\Phi \right\rangle \tag{29}$$

Using the chain rule, we get

$$\left\langle \frac{\partial \mathcal{L}}{\partial M}, dM \right\rangle = \left\langle \frac{\partial \mathcal{L}}{\partial M}, dW_0 W_1 + W_0 dW_1 \right\rangle$$
$$\left\langle \frac{\partial \mathcal{L}}{\partial \Phi}, d\Phi \right\rangle = \left\langle \frac{\partial \mathcal{L}}{\partial \Phi}, dW_0 W_Q W_K^\mathsf{T} W_0^\mathsf{T} + W_0 dW_Q W_K^\mathsf{T} W_0^\mathsf{T} + W_0 W_Q dW_K^\mathsf{T} W_0^\mathsf{T} + W_0 W_Q W_K^\mathsf{T} dW_0^\mathsf{T} \right\rangle \tag{30}$$

We derive the evolution equation for $W_0$, and the other derivations are similar. We collect items related to $dW_0$:

$$\left\langle \frac{\partial \mathcal{L}}{\partial M}, dW_0 W_1 \right\rangle + \left\langle \frac{\partial \mathcal{L}}{\partial \Phi}, dW_0 W_Q W_K^\mathsf{T} W_0^\mathsf{T} + W_0 W_Q W_K^\mathsf{T} dW_0^\mathsf{T} \right\rangle$$
$$= \mathrm{tr}\left(\left(\frac{\partial \mathcal{L}}{\partial M}\right)^\mathsf{T} dW_0 W_1\right) + \mathrm{tr}\left(\left(\frac{\partial \mathcal{L}}{\partial \Phi}\right)^\mathsf{T} (dW_0 W_Q W_K^\mathsf{T} W_0^\mathsf{T} + W_0 W_Q W_K^\mathsf{T} dW_0^\mathsf{T})\right)$$
$$= \mathrm{tr}\left(W_1 \left(\frac{\partial \mathcal{L}}{\partial M}\right)^\mathsf{T} dW_0\right) + \mathrm{tr}\left(W_Q W_K^\mathsf{T} W_0^\mathsf{T} \left(\frac{\partial \mathcal{L}}{\partial \Phi}\right)^\mathsf{T} dW_0\right) + \mathrm{tr}\left(W_K W_Q^\mathsf{T} W_0^\mathsf{T} \left(\frac{\partial \mathcal{L}}{\partial \Phi}\right) dW_0\right) \tag{31}$$
$$= \left\langle \frac{\partial \mathcal{L}}{\partial M} W_1^\mathsf{T} + \frac{\partial \mathcal{L}}{\partial \Phi} W_0 W_K W_Q^\mathsf{T} + \left(\frac{\partial \mathcal{L}}{\partial \Phi}\right)^\mathsf{T} W_0 W_Q W_K^\mathsf{T}, dW_0 \right\rangle$$

Therefore, we obtain an expression for $\frac{\partial \mathcal{L}}{\partial W_0}$. Since we are considering gradient descent, the evolution of the parameters follows the direction of the negative gradient. Then we derive the expression of $\frac{\partial \mathcal{L}}{\partial M}$ and $\frac{\partial \mathcal{L}}{\partial \Phi}$ in large $N$ and $s$ limit. Taking the total differential of the loss and taking the term about $\mathrm{d}M$, we get

$$\left\langle \frac{\partial \mathcal{L}}{\partial M}, \mathrm{d}M \right\rangle = \frac{1}{N} \sum_{i=1}^{N} - \sum_{l=1}^{s} A_{s,l}(X_i) e_{x,l}^{\intercal} \mathrm{d}M e_{y_i} + \frac{1}{N} \sum_{i=1}^{N} \sum_{j} p_{y_j}(X_i) \sum_{l=1}^{s} A_{s,l}(X_i) e_{x,l}^{\intercal} \mathrm{d}M e_{y_j} \tag{32}$$

Here, $A_{s,l}(X_i) = \frac{\exp(e_{x_{i,s}}^{\intercal} \Phi e_{x_{i,l}})}{\sum_{l'=1}^{s} \exp(e_{x_{i,s}}^{\intercal} \Phi e_{x_{i,l'}})}$. Based on Proposition A.1, we find that

$$A_{s,l}(X_i) = \frac{\frac{1}{s} \exp(e_{x_{i,s}}^{\intercal} \Phi e_{x_{i,l}})}{\frac{1}{s} \sum_{l'=1}^{s} \exp(e_{x_{i,s}}^{\intercal} \Phi e_{x_{i,l'}})} = \frac{\frac{1}{s} \exp(e_{x_{i,s}}^{\intercal} \Phi e_{x_{i,l}})}{\sum_{j=1}^{d} \pi_j \exp(e_{x_{i,s}}^{\intercal} \Phi e_j)} \tag{33}$$

Then, for sufficiently large sequence length $s$,

$$\begin{aligned}
\sum_{l=1}^{s} A_{s,l}(X_i) e_{x_{i,l}}^{\intercal} &= \frac{1}{\sum_{j=1}^{d} \pi_j \exp(e_{x_{i,s}}^{\intercal} \Phi e_j)} \sum_{l=1}^{s} \frac{1}{s} \exp(e_{x_{i,s}}^{\intercal} \Phi e_{x_{i,l}}) e_{x_{i,l}}^{\intercal} \\
&= \frac{1}{\sum_{j=1}^{d} \pi_j \exp(e_{x_{i,s}}^{\intercal} \Phi e_j)} \sum_{j'=1}^{d} \pi_{j'} \exp(e_{x_{i,s}}^{\intercal} \Phi e_{j'}) e_{j'}^{\intercal} = \mathbb{A}_{x_{i,s}}
\end{aligned} \tag{34}$$

It implies that the output probability $p(X_i)$ actually depends on the last token $x_{i,s}$. That is

$$p_j(X_i) = \frac{\exp(\mathbb{A}_{x_{i,s}} M e_{y_j})}{\sum_{j'=1}^{d} \exp(\mathbb{A}_{x_{i,s}} M e_{y'_j})} = \mathbb{P}_{x_{i,s},j} \tag{35}$$

Based on this fact and Eq. (34), Eq. (32) can be reformulated by using the notations about $\mathbb{A}$ and $\mathbb{P}$.

$$\left\langle \frac{\partial \mathcal{L}}{\partial M}, \mathrm{d}M \right\rangle = -\frac{1}{N} \sum_{i=1}^{N} \mathbb{A}_{x_{i,s}} \mathrm{d}M e_{y_i} + \frac{1}{N} \sum_{i=1}^{N} \mathbb{A}_{x_{i,s}} \mathrm{d}M \mathbb{P}_{x_{i,s}}^{\intercal}. \tag{36}$$

Based on Proposition A.1, we have $x_{i,s} \sim \pi^{\intercal}$ when $s$ is sufficiently large. Thus, we get

$$\left\langle \frac{\partial \mathcal{L}}{\partial M}, \mathrm{d}M \right\rangle = -\sum_{i=1}^{d} \pi_i \mathbb{A}_i \mathrm{d}M (P_i - \mathbb{P}_i)^{\intercal} \tag{37}$$

Using the trace theorem again, we have

$$\frac{\partial \mathcal{L}}{\partial M} = -\sum_{i=1}^{d} \pi_i \mathbb{A}_i^{\intercal} (P_i - \mathbb{P}_i). \tag{38}$$

Then we derive $\frac{\partial \mathcal{L}}{\partial \Phi}$. By direct computation, we get

$$\begin{aligned}
\left\langle \frac{\partial \mathcal{L}}{\partial \Phi}, \mathrm{d}\Phi \right\rangle &= \frac{1}{N} \sum_{i=1}^{N} - \left( \sum_{l=1}^{s} \left( A_{s,l} \mathrm{d}\left( e_{x,s}^{\intercal} \Phi e_{x,l} \right) - A_{s,l} \sum_{l'} A_{s,l'} \mathrm{d}\left( e_{x,s}^{\intercal} \Phi e_{x,l'} \right) \right) e_{x,l}^{\intercal} M e_{y_i} \right) \\
&\quad + \frac{1}{N} \sum_{i=1}^{N} \sum_{j} p_{y_j} \left( \sum_{l=1}^{s} \left( A_{s,l} \mathrm{d}\left( e_{x,s}^{\intercal} \Phi e_{x,l} \right) - A_{s,l} \sum_{l'} A_{s,l'} \mathrm{d}\left( e_{x,s}^{\intercal} \Phi e_{x,l'} \right) \right) e_{x,l}^{\intercal} M e_{y_j} \right)
\end{aligned} \tag{39}$$

Using the notations we introduced, the derivative can be reformulated as

$$\begin{aligned}
\left\langle \frac{\partial \mathcal{L}}{\partial \Phi}, \mathrm{d}\Phi \right\rangle &= -\sum_{i} \pi_i \left( \sum_{l=1}^{s} A_{x_i,l} e_{x_i}^{\intercal} \mathrm{d}\Phi (e_l - \mathbb{A}_i^{\intercal}) \right) e_l^{\intercal} M (P_i - \mathbb{P}_i)^{\intercal} \\
&= -\sum_{i} \pi_i e_{x_i}^{\intercal} \mathrm{d}\Phi \left( \mathrm{diag}(\mathbb{A}_i^{\intercal}) - \mathbb{A}_i^{\intercal} \mathbb{A}_i \right) M (P_i - \mathbb{P}_i)^{\intercal}
\end{aligned} \tag{40}$$

As a result, using the trace theorem, we get the expression about $\frac{\partial \mathcal{L}}{\partial \Phi}$ as follows:

$$\frac{\partial \mathcal{L}}{\partial \Phi} = -\sum_i \pi_i e_{x_i} (P_i - \mathbb{P}_i) M^{\mathsf{T}} \left( \operatorname{diag}(\mathbb{A}_i^{\mathsf{T}}) - \mathbb{A}_i^{\mathsf{T}} \mathbb{A}_i \right) \tag{41}$$

$\square$

# B. Theoretical details in Sec. 4

## B.1. Theoretical details in Sec. 4.1

### Proof of Lemma 4.1

*Proof.* Let $\Delta\theta(t) := \theta(t) - \theta_*$. Since $F(\theta_*) = 0$ and $J = DF(\theta_*)$, we can write

$$\dot{\Delta\theta}(t) = J\Delta\theta(t) + R(\Delta\theta(t)), \qquad R(\Delta\theta) := F(\theta_* + \Delta\theta) - J\Delta\theta. \tag{42}$$

By assumption (6), for all $\|\Delta\theta\| \leq r$,

$$\|R(\Delta\theta)\| \leq L\|\Delta\theta\|^2. \tag{43}$$

**Step 1: Variation-of-constants representation.** Let $\tilde{\Delta\theta}(t) := \mathrm{e}^{Jt}\Delta\theta(0)$ be the solution of the linearized system. From (42), the solution satisfies the Duhamel formula

$$\Delta\theta(t) = \mathrm{e}^{Jt}\Delta\theta(0) + \int_0^t \mathrm{e}^{J(t-s)} R(\Delta\theta(s))\mathrm{d}s = \tilde{\Delta\theta}(t) + \int_0^t \mathrm{e}^{J(t-s)} R(\Delta\theta(s))\mathrm{d}s. \tag{44}$$

Define the linearization error $E(t) := \Delta\theta(t) - \tilde{\Delta\theta}(t)$. Then $E(0) = 0$ and by (44),

$$E(t) = \int_0^t \mathrm{e}^{J(t-s)} R(\Delta\theta(s))\mathrm{d}s. \tag{45}$$

**Step 2: A standard bound on the semigroup $\mathrm{e}^{Jt}$.** Let $\mu := \sup\{\Re(\lambda) : \lambda \in \sigma(J)\}$. In finite dimension, for the chosen operator norm there exists a constant $K \geq 1$ such that

$$\|\mathrm{e}^{Jt}\| \leq K\mathrm{e}^{\mu t}, \qquad \forall t \geq 0. \tag{46}$$

**Step 3: Bootstrap control inside the neighborhood.** Fix a time horizon $T > 0$ such that

$$\|\tilde{\Delta\theta}(t)\| \leq \frac{r}{2}, \qquad \forall t \in [0, T]. \tag{47}$$

We will show that for $\varepsilon := \|\Delta\theta(0)\|$ sufficiently small (depending on $J, L, r$), the trajectory stays in the ball $\|\Delta\theta(t)\| \leq r$ on $[0, T]$, so that (43) applies.

Indeed, from (44), (46), and (43), as long as $\|\Delta\theta(s)\| \leq r$ for all $s \in [0, t]$, we have

$$\|E(t)\| \leq \int_0^t \|\mathrm{e}^{J(t-s)}\| \, \|R(\Delta\theta(s))\|\mathrm{d}s \leq KL \int_0^t \mathrm{e}^{\mu(t-s)}\|\Delta\theta(s)\|^2\mathrm{d}s. \tag{48}$$

Also $\|\tilde{\Delta\theta}(t)\| \leq \|\mathrm{e}^{Jt}\| \, \|\Delta\theta(0)\| \leq K\varepsilon\mathrm{e}^{\mu t}$.

We now bootstrap the bound

$$\|\Delta\theta(t)\| \leq 2\|\tilde{\Delta\theta}(t)\| \quad \text{for all } t \in [0, T]. \tag{49}$$

Assuming (49) holds on $[0, t]$, then $\|\Delta\theta(s)\| \leq 2\|\tilde{\Delta\theta}(s)\| \leq r$ by (47), so (48) applies and yields

$$\|E(t)\| \leq KL \int_0^t \mathrm{e}^{\mu(t-s)} \big(2\|\tilde{\Delta\theta}(s)\|\big)^2 \mathrm{d}s = 4KL \int_0^t \mathrm{e}^{\mu(t-s)}\|\tilde{\Delta\theta}(s)\|^2\mathrm{d}s$$

$$\leq 4KL \int_0^t \mathrm{e}^{\mu(t-s)} \big(K\varepsilon\mathrm{e}^{\mu s}\big)^2 \mathrm{d}s = 4K^3 L \varepsilon^2 \mathrm{e}^{\mu t} \int_0^t \mathrm{e}^{\mu s}\mathrm{d}s. \tag{50}$$

If $\mu > 0$, then $\int_0^t e^{\mu s} ds = (e^{\mu t} - 1)/\mu \leq e^{\mu t}/\mu$, and thus

$$\|E(t)\| \leq \frac{4K^3 L}{\mu} \varepsilon^2 e^{2\mu t}. \tag{51}$$

If $\mu = 0$, then $\int_0^t e^{\mu s} ds = t$ and (50) gives

$$\|E(t)\| \leq 4K^3 L \varepsilon^2 t. \tag{52}$$

(For $\mu < 0$, one may similarly bound the integral by a constant and obtain a uniform $O(\varepsilon^2)$ error.)

Now choose $\varepsilon$ small enough such that on $[0, T]$,

$$\|E(t)\| \leq \|\tilde{\Delta}\theta(t)\| \quad \forall t \in [0, T]. \tag{53}$$

This is possible because by (47) we have $\|\tilde{\Delta}\theta(t)\| \leq r/2$, while (51) shows $\|E(t)\|$ is $O(\varepsilon^2)$ times an exponential factor; e.g. it suffices to require

$$\frac{4K^3 L}{\mu} \varepsilon e^{\mu T} \leq \frac{1}{2} \quad (\mu > 0), \qquad \text{or} \qquad 4K^3 L \varepsilon T \leq \frac{1}{2} \quad (\mu = 0),$$

and recall $T$ is such that $\|\tilde{\Delta}\theta(t)\| \leq r/2$ on $[0, T]$, hence $e^{\mu T}$ is at most on the order of $1/\varepsilon$ when $\mu > 0$. Under (53),

$$\|\Delta\theta(t)\| \leq \|\tilde{\Delta}\theta(t)\| + \|E(t)\| \leq 2\|\tilde{\Delta}\theta(t)\| \leq r,$$

so the bootstrap is self-consistent and (51) (or (52)) holds for all $t \in [0, T]$. This proves (7) with $C = \frac{4K^3 L}{\mu}$ for $\mu > 0$ and $C = 4K^3 L$ for $\mu = 0$.

**Step 4: The $\Theta(\log(1/\varepsilon))$ window when $\mu > 0$.** If $\mu > 0$, then $\|\tilde{\Delta}\theta(t)\| \leq K\varepsilon e^{\mu t}$. Therefore the condition $\|\tilde{\Delta}\theta(t)\| \leq r/2$ holds at least up to times

$$t \leq \frac{1}{\mu} \log \frac{r}{2K\varepsilon} = \Theta(\log(1/\varepsilon)),$$

which is exactly the linearization window claimed in the lemma.

**Step 5: Alignment with the unstable eigenvector with positive spectral gap.** Assume now that $J$ has a simple eigenvalue $\mu > 0$ with eigenvector $v_u$ and a spectral gap: $\Re(\lambda) \leq \mu - \delta$ for all other eigenvalues. Let $\Pi_u$ be the spectral projection onto $\text{span}\{v_u\}$ and $\Pi_s = I - \Pi_u$. Then there exist constants $K_u, K_s$ such that

$$\|\Pi_u e^{Jt}\| \leq K_u e^{\mu t}, \qquad \|\Pi_s e^{Jt}\| \leq K_s e^{(\mu - \delta)t}, \qquad \forall t \geq 0. \tag{54}$$

Write $\tilde{\Delta}\theta(t) = \Pi_u \tilde{\Delta}\theta(t) + \Pi_s \tilde{\Delta}\theta(t)$. If $\langle \Delta(0), v_u \rangle \neq 0$, then $\Pi_u \tilde{\Delta}\theta(t) = a_0 e^{\mu t} v_u$ for some $a_0 \neq 0$, while

$$\|\Pi_s \tilde{\Delta}\theta(t)\| \leq K_s \varepsilon e^{(\mu - \delta)t}.$$

Hence

$$\frac{\tilde{\Delta}\theta(t)}{\|\tilde{\Delta}\theta(t)\|} \rightarrow \pm \frac{v_u}{\|v_u\|} \quad \text{as } t \rightarrow \infty, \tag{55}$$

and the convergence rate is $O(e^{-\delta t})$.

For the nonlinear trajectory, decompose $\Delta\theta(t) = u(t) + s(t)$ with $u(t) := \Pi_u \Delta\theta(t)$ and $s(t) := \Pi_s \Delta\theta(t)$. Projecting (44) onto the two subspaces and using (54) gives

$$u(t) = \Pi_u \tilde{\Delta}\theta(t) + \int_0^t \Pi_u e^{J(t-s)} R(\Delta\theta(s)) \, ds, \quad s(t) = \Pi_s \tilde{\Delta}\theta(t) + \int_0^t \Pi_s e^{J(t-s)} R(\Delta\theta(s)) \, ds. \tag{56}$$

Inside the linearization window we have $\|\Delta\theta(s)\| \leq r$, so (43) applies and, using also $\|\Delta\theta(s)\| \lesssim \varepsilon e^{\mu s}$ from the bootstrap in Step 3, we obtain

$$\|R(\Delta(s))\| \leq L \|\Delta\theta(s)\|^2 \lesssim L\varepsilon^2 e^{2\mu s}.$$

Plugging into (56) yields, for $t$ in the linearization window,

$$\|u(t) - \Pi_u \tilde{\Delta}\theta(t)\| \leq \int_0^t \|\Pi_u e^{J(t-s)}\| \, \|R(\Delta\theta(s))\| \mathrm{d}s \lesssim \varepsilon^2 e^{2\mu t}, \tag{57}$$

$$\|s(t) - \Pi_s \tilde{\Delta}\theta(t)\| \leq \int_0^t \|\Pi_s e^{J(t-s)}\| \, \|R(\Delta\theta(s))\| \mathrm{d}s \lesssim \varepsilon^2 e^{2\mu t}. \tag{58}$$

Therefore,

$$\|s(t)\| \leq \|\Pi_s \tilde{\Delta}\theta(t)\| + \|s(t) - \Pi_s \tilde{\Delta}\theta(t)\| \lesssim \varepsilon e^{(\mu-\delta)t} + \varepsilon^2 e^{2\mu t},$$

while

$$\|u(t)\| \geq \|\Pi_u \tilde{\Delta}\theta(t)\| - \|u(t) - \Pi_u \tilde{\Delta}\theta(t)\| \gtrsim \varepsilon e^{\mu t} - \varepsilon^2 e^{2\mu t}.$$

Hence, for times $t$ such that $\varepsilon e^{\mu t}$ is still sufficiently small (which holds throughout a $\Theta(\log(1/\varepsilon))$ interval inside the linearization window), we have

$$\frac{\|s(t)\|}{\|u(t)\|} \lesssim e^{-\delta t} + \varepsilon e^{\mu t}. \tag{59}$$

Now let $t$ increase while remaining in the linearization window (so $t \to \infty$ is possible as $\varepsilon \to 0$), and choose any sequence $t = t(\varepsilon)$ such that

$$t(\varepsilon) \to \infty, \qquad \varepsilon e^{\mu t(\varepsilon)} \to 0 \quad (\text{e.g. } t(\varepsilon) = \tfrac{1}{2\mu}\log(1/\varepsilon)).$$

Then (59) implies $\|s(t)\|/\|u(t)\| \to 0$, so

$$\frac{\Delta\theta(t)}{\|\Delta\theta(t)\|} = \frac{u(t) + s(t)}{\|u(t) + s(t)\|} \to \pm\frac{v_u}{\|v_u\|}.$$

This proves (8). □

**Proof of Theorem 4.2**

*Proof.* The claim follows by a direct evaluation of the gradients at the origin. By Lemma 4.1, the early-time dynamics is governed by the linearization at $\theta = 0$, so we substitute $\theta = 0$ into (5).

At $\theta = 0$, the definitions of $\mathbb{A}_i$ and $\mathbb{P}_i$ yield

$$\mathbb{A}_i = \pi^\mathsf{T}, \qquad \mathbb{P}_i = \frac{1}{d}\mathbf{1}^\mathsf{T}, \quad \text{for all } i.$$

Moreover, since $\pi^\mathsf{T}$ is stationary, we have $\pi^\mathsf{T} P = \pi^\mathsf{T}$, and hence

$$\sum_i \pi_i P_i = \pi^\mathsf{T}.$$

Plugging these identities into the expression of $\frac{\partial \mathcal{L}}{\partial M}$ in (5), we obtain

$$\left.\frac{\partial \mathcal{L}}{\partial M}\right|_{\theta=0} = -\pi\left(\pi - \frac{1}{d}\mathbf{1}\right)^\mathsf{T}.$$

This is exactly the desired formula, completing the proof. □

**Analysis of transition between stage I and II**   Although linearization reveals the characteristics of the first stage, gaps remain in the transition from the first to the second stage. At this point, simple linearization provides an error estimate that is too loose to accurately represent the stage transition. Therefore, a more refined error estimate is needed. To achieve this, we extend the techniques in (Chen et al., 2024c) or (Xu et al., 2025a) and prove a rigorous stage transition. Without loss of generality, we assume $\frac{\pi^\mathsf{T}}{\|\pi\|}W_0, W_1 \frac{\pi - \frac{1}{d}\mathbf{1}}{\|\pi - \frac{1}{d}\mathbf{1}\|} \sim \Theta(\varepsilon)$. First, we introduce two quantities to show the magnitudes that distinguish the outer and inner parameters:

$$W_{\text{out, max}} = \max\{\|W_0\|_{\text{F}}, \|W_1\|_{\text{F}}\}, \quad W_{\text{in, max}} = \max\{\|W_Q\|_{\text{F}}, \|W_K\|_{\text{F}}\}. \tag{60}$$

Then, recalling the definition of dynamics (4), we have the estimate of error term in transition of stage I and stage II:

$$\left\|\frac{\partial\mathcal{L}}{\partial\Phi}W_0W_{QK}^\mathsf{T}\right\|, \left\|\left(\frac{\partial\mathcal{L}}{\partial\Phi}\right)^\mathsf{T}W_0W_{QK}\right\| \lesssim W_{\text{out, max}}^3 W_{\text{in, max}}^2, \tag{61}$$

and

$$\left\|W_0^\mathsf{T}\frac{\partial\mathcal{L}}{\partial\Phi}W_0W_K\right\|, \left\|W_0^\mathsf{T}\left(\frac{\partial\mathcal{L}}{\partial\Phi}\right)^\mathsf{T}W_0W_Q\right\| \lesssim W_{\text{out, max}}^4 W_{\text{in, max}}. \tag{62}$$

Moreover, we decompose $W_0, W_1$ into condensation directions and normal directions:

$$W_0 = \frac{\pi}{\|\pi\|}\frac{\pi^\mathsf{T}}{\|\pi\|}W_0 + R_0, \quad W_1 = W_1\frac{\pi-\frac{1}{d}\mathbf{1}}{\|\pi-\frac{1}{d}\mathbf{1}\|}\frac{\pi^\mathsf{T}-\frac{1}{d}\mathbf{1}^\mathsf{T}}{\|\pi-\frac{1}{d}\mathbf{1}\|} + R_1. \tag{63}$$

We begin the estimate by decomposing $\frac{\partial\mathcal{L}}{\partial M}$:

$$\frac{\partial\mathcal{L}}{\partial M} = -\sum_{i=1}^d \pi_i\mathbb{A}_i(P_i - \mathbb{P}_i) = -\pi(\pi^\mathsf{T} - \mathbb{P}) + E_1, \tag{64}$$

where $\mathbb{P}_j = \frac{\exp(\pi^\mathsf{T}Me_j)}{\sum_{j'}\exp(\pi^\mathsf{T}Me_{j'})}$ and $\|E_1\| \lesssim W_{\text{out, max}}^2 W_{\text{in, max}}^2$. Here, we use the expansion of $\mathbb{A}_i$. Next, we will further elaborate on $\mathbb{P}$. By the definition of $\mathbb{P}$, it can be viewed as $\text{softmax}(\pi^\mathsf{T}M)$. Using the decompositions of $W_0$ and $W_1$, we get

$$\mathbb{P} = \text{softmax}\left(\pi^\mathsf{T}qq^\mathsf{T}W_0W_1uu^\mathsf{T} + \pi^\mathsf{T}R_0W_1uu^\mathsf{T} + \pi^\mathsf{T}qq^\mathsf{T}W_0R_1 + R_0R_1\right).$$

Here we denote $q = \frac{\pi}{\|\pi\|}$ and $u = \frac{\pi-\frac{1}{d}\mathbf{1}}{\|\pi-\frac{1}{d}\mathbf{1}\|}$ for simplicity. Using the fact that $\text{d}\,\text{softmax}(z) = \text{Var}(\text{softmax}(z))$, we have

$$\begin{aligned}\mathbb{P} &= p_{\text{sym}} + \left(\pi^\mathsf{T}R_0W_1uu^\mathsf{T} + \pi^\mathsf{T}qq^\mathsf{T}W_0R_1 + \pi^\mathsf{T}R_0R_1\right)\text{Var}(p_{\text{sym}}) + \tilde{E}_2 \\ &= p_{\text{sym}} + \left(\pi^\mathsf{T}R_0W_1uu^\mathsf{T} + \pi^\mathsf{T}qq^\mathsf{T}W_0R_1\right)\text{Var}(p_{\text{sym}}) + E_2\end{aligned} \tag{65}$$

where $E_2$ satisfies $\|E_2\| \lesssim \|R_0\|^2 + \|R_1\|^2$. Substituting this equation into the expression of $\frac{\partial\mathcal{L}}{\partial M}$, we get

$$\frac{\partial\mathcal{L}}{\partial M} = -\pi(\pi^\mathsf{T} - p_{\text{sym}}) + \pi\left(\pi^\mathsf{T}R_0W_1uu^\mathsf{T} + \pi^\mathsf{T}qq^\mathsf{T}W_0R_1\right)\text{Var}(p_{\text{sym}}) + E_1 + E_2 \tag{66}$$

Before we begin the error estimate, we emphasize the following two facts. First, $\pi^\mathsf{T} - p_{\text{sym}} \propto u^\mathsf{T}$ under the setting of one high frequency token and other low frequency tokens. Second, for vector $u_\perp$ which is orthogonal to $u$ and $\mathbf{1}$, we have $\text{Var}(p_{\text{sym}})u_\perp = p_{\text{sym},i\neq 1}u_\perp$. For $\mathbf{1}$, we have $\text{Var}(p_{\text{sym}})\mathbf{1} = 0$. Both of these two facts can be verified directly. Then we begin the error estimate. For $q_\perp^\mathsf{T}W_0$, we have

$$\frac{\text{d}}{\text{d}t}q_\perp^\mathsf{T}W_0 = q_\perp^\mathsf{T}(E_1 + E_2) + q_\perp^\mathsf{T}\frac{\partial\mathcal{L}}{\partial\Phi}W_0W_{QK}^\mathsf{T} + \left(\frac{\partial\mathcal{L}}{\partial\Phi}\right)^\mathsf{T}W_0W_{QK} \tag{67}$$

Thus, we have

$$\|q_\perp^\mathsf{T}W_0(t)\| \leq \|q_\perp^\mathsf{T}W_0(0)\| + C\int_0^t W_{\text{out, max}}^2 W_{\text{in, max}}^2 + \|R_0\|^2 + \|R_1\|^2\text{d}s. \tag{68}$$

Sum them up, we get

$$\|R_0(t)\| \leq \|R_0(0)\| + C\int_0^t W_{\text{out, max}}^2 W_{\text{in, max}}^2 + \|R_0\|^2 + \|R_1\|^2\text{d}s. \tag{69}$$

For $W_1u_\perp$, we have

$$\frac{\text{d}}{\text{d}t}W_1u_\perp = -W_0^\mathsf{T}\pi\left(\pi^\mathsf{T}R_0W_1uu^\mathsf{T} + \pi^\mathsf{T}qq^\mathsf{T}W_0R_1\right)\text{Var}(p_{\text{sym}})u_\perp - W_0^\mathsf{T}(E_1 + E_2)u_\perp \tag{70}$$

Using the fact that $\text{Var}(p_{\text{sym}})u_\perp = p_{\text{sym},i\neq 1}u_\perp$, we get

$$\frac{\text{d}}{\text{d}t}W_1u_\perp = -p_{\text{sym},i\neq 1}W_0^\mathsf{T}\pi\left(\pi^\mathsf{T}qq^\mathsf{T}W_0R_1\right)u_\perp - W_0^\mathsf{T}(E_1 + E_2)u_\perp \tag{71}$$

Then, we get

$$\frac{\mathrm{d}}{\mathrm{d}t}\|W_1 u_\perp\|^2 = -2\|\pi\|^2 p_{\text{sym},i\neq 1} u_\perp^\mathsf{T} W_1^\mathsf{T} W_0^\mathsf{T} qq^\mathsf{T} W_0 W_1 u_\perp - 2u_\perp^\mathsf{T} W_1^\mathsf{T} W_0^\mathsf{T}(E_1 + E_2)u_\perp$$
$$\lesssim \|R_1\| \left(W_{\text{out, max}}^2 W_{\text{in, max}}^2 + \|R_0\|^2 + \|R_1\|^2\right) \tag{72}$$

Sum them up, we get

$$\frac{\mathrm{d}}{\mathrm{d}t}\|R_1\|^2 \lesssim \|R_1\| \left(W_{\text{out, max}}^2 W_{\text{in, max}}^2 + \|R_0\|^2 + \|R_1\|^2\right) \tag{73}$$

After establish the error estimate of the normal terms, We formally begin the proof of the phase transition. We divide the entire phase transition into two parts. The first part can be seen as a plateau period, where the condensation direction grows significantly and is about to become $O(1)$, while the normal direction remains small due to our fine error estimation. In the second part, the condensation direction rapidly reaches the neighborhood of the critical point.

We define

$$T_1 = \sup\left\{t \geq 0 \mid W_{\text{out,max}} \lesssim \varepsilon^{\frac{1}{n}}\right\} \tag{74}$$

Moreover, let $T_p$ be the first time when the condensation component reaches the plateau scale:

$$T_p := \frac{1 - \frac{1}{n}}{\|\pi\|\|\pi - \frac{1}{d}\mathbf{1}\|} \log\frac{1}{\varepsilon}. \tag{75}$$

**Lemma B.1** (Refined error estimate). *For every $t \leq \min\{T_1, T_p\}$, one has*

$$\|W_Q(t)\|_\mathrm{F} + \|W_K(t)\|_\mathrm{F} \leq C\varepsilon, \tag{76}$$

*and*

$$\|R_0(t)\|_\mathrm{F} + \|R_1(t)\|_\mathrm{F} \leq C\varepsilon, \tag{77}$$

*provided the initialization satisfies $\|W_Q(0)\|_\mathrm{F} + \|W_K(0)\|_\mathrm{F} + \|R_0(0)\|_\mathrm{F} + \|R_1(0)\|_\mathrm{F} \lesssim \varepsilon$ and $\varepsilon > 0$ is sufficiently small.*

*Proof.* We start from the equations

$$\frac{dW_Q}{dt} = -W_0^\mathsf{T}\frac{\partial\mathcal{L}}{\partial\Phi}W_0 W_K, \qquad \frac{dW_K}{dt} = -W_0^\mathsf{T}\left(\frac{\partial\mathcal{L}}{\partial\Phi}\right)^\mathsf{T} W_0 W_Q.$$

By the definition of $W_{\text{out,max}}$, on $[0, \min\{T_1, T_p\}]$ we have

$$\left\|W_0^\mathsf{T}\frac{\partial\mathcal{L}}{\partial\Phi}W_0\right\| + \left\|W_0^\mathsf{T}\left(\frac{\partial\mathcal{L}}{\partial\Phi}\right)^\mathsf{T} W_0\right\| \leq CW_{\text{out,max}}^4 \leq C\varepsilon^{\frac{4}{n}}.$$

Hence

$$\frac{d}{dt}\left(\|W_Q\|_\mathrm{F} + \|W_K\|_\mathrm{F}\right) \leq C\varepsilon^{\frac{4}{n}}\left(\|W_Q\|_\mathrm{F} + \|W_K\|_\mathrm{F}\right).$$

By Gronwall's inequality and the assumption on the initialization, (76) follows.

Next, recall the expansion

$$\frac{\partial\mathcal{L}}{\partial M} = -\pi(\pi^\mathsf{T} - p_{\text{sym}}) + \pi\left(\pi^\mathsf{T} R_0 W_1 uu^\mathsf{T} + \pi^\mathsf{T} qq^\mathsf{T} W_0 R_1\right)\text{Var}(p_{\text{sym}}) + E, \tag{78}$$

where

$$\|E\| \leq C\left(W_{\text{out,max}}^2 W_{\text{in,max}}^2 + \|R_0\|_\mathrm{F}^2 + \|R_1\|_\mathrm{F}^2\right). \tag{79}$$

Projecting the $W_0$-equation to the orthogonal complement of $q$, we obtain

$$\frac{d}{dt}\left((I - qq^\mathsf{T})W_0\right) = (I - qq^\mathsf{T})\left(-\frac{\partial\mathcal{L}}{\partial M}W_1^\mathsf{T} - \frac{\partial\mathcal{L}}{\partial\Phi}W_0 W_{QK}^\mathsf{T} - \left(\frac{\partial\mathcal{L}}{\partial\Phi}\right)^\mathsf{T} W_0 W_{QK}\right).$$

Since $\pi(\pi^{\mathsf{T}} - p_{\text{sym}})$ lies in the span of $q$, its contribution vanishes under the projection $I - qq^{\mathsf{T}}$. Thus

$$\frac{d}{dt}\|R_0\|_{\text{F}} \leq C\Big(W_{\text{out,max}}^2 W_{\text{in,max}}^2 + \|R_0\|_{\text{F}}^2 + \|R_1\|_{\text{F}}^2\Big).$$

Similarly, projecting the $W_1$-equation to the orthogonal complement of $u$, we get

$$\frac{d}{dt}(W_1(I - uu^{\mathsf{T}})) = -W_0^{\mathsf{T}}\frac{\partial\mathcal{L}}{\partial M}(I - uu^{\mathsf{T}}).$$

Using the key identities

$$(\pi^{\mathsf{T}} - p_{\text{sym}})(I - uu^{\mathsf{T}}) = 0, \qquad \text{Var}(p_{\text{sym}})\mathbf{1} = 0,$$

and the fact that $\text{Var}(p_{\text{sym}})$ acts as a scalar on the subspace $\{v : \langle v, \mathbf{1}\rangle = \langle v, u\rangle = 0\}$, we obtain

$$\frac{d}{dt}\|R_1\|_{\text{F}}^2 \leq C\|R_1\|_{\text{F}}\Big(W_{\text{out,max}}^2 W_{\text{in,max}}^2 + \|R_0\|_{\text{F}}^2 + \|R_1\|_{\text{F}}^2\Big).$$

Combining the two inequalities, using (76), and applying a standard bootstrap argument yields (77). $\qquad\square$

Next, we show that $T_p \leq T_1$ by error estimate.

**Proposition B.2.** *For sufficiently small $\varepsilon$, one has*
$$T_p \leq T_1.$$

*Proof.* By Lemma B.1, for all $t \leq \min\{T_1, T_p\}$,

$$W_{\text{in,max}}(t) \leq C\varepsilon, \qquad \|R_0(t)\|_{\text{F}} + \|R_1(t)\|_{\text{F}} \leq C\varepsilon.$$

The only possible mechanism that can terminate the bootstrap interval before $T_1$ is the growth of the condensation component itself.

Define the two condensation variables

$$a(t) := q^{\mathsf{T}}W_0(t) \in \mathbb{R}^m, \qquad b(t) := W_1(t)u \in \mathbb{R}^m. \tag{80}$$

By projecting the dynamics of $W_0$ and $W_1$ onto $q$ and $u$, and using (78), we obtain

$$\dot{a} = \|\pi\|\,\|\pi - p_{\text{sym}}(z)\|\,b + \mathcal{E}_a, \qquad \dot{b} = \|\pi\|\,\|\pi - p_{\text{sym}}(z)\|\,a + \mathcal{E}_b, \tag{81}$$

where

$$z(t) := \langle a(t), b(t)\rangle, \tag{82}$$

and the error terms satisfy

$$\|\mathcal{E}_a\| + \|\mathcal{E}_b\| \leq C\Big(W_{\text{out,max}}^2 W_{\text{in,max}}^2 + W_{\text{out,max}}^2\|R_0\|_{\text{F}} + W_{\text{out,max}}^2\|R_1\|_{\text{F}}\Big). \tag{83}$$

Considering the linearized version:

$$\dot{\tilde{a}} = \lambda_*\tilde{b}, \quad \dot{\tilde{b}} = \lambda_*\tilde{a} \tag{84}$$

Since the linearized system is just the system linearized at $\theta = 0$, using the same argument as Lemma 4.1, we get for $t \in [0, T_p]$

$$\|a(t) - \tilde{a}(t)\| + \|b(t) - \tilde{b}(t)\| \lesssim \varepsilon^{\frac{2}{n}}. \tag{85}$$

Considering the solution of linearized system, we get:

$$\|\tilde{a}(T_p)\|, \|\tilde{b}(T_p)\| \sim \varepsilon^{\frac{1}{n}} \tag{86}$$

Combined this with the error estimate, we finish the proof. $\qquad\square$

*Remark* B.3. At first glance, this proposition resembles the previous linearization result. However, the key distinction is that here we obtain a significantly stronger control of the dynamics in the normal directions. This stronger control is essential for carrying out the cross-stage analysis.

To complete the cross-stage analysis, we need a dynamics so that we can work with a significant condensation direction. Specifically, observing the Eq. (81), we find that

$$\dot{z} = \|\pi\| \|\pi - p_{\text{sym}}(z)\| (\|a\|^2 + \|b\|^2) + \langle \mathcal{E}_a, b \rangle + \langle \mathcal{E}_b, a \rangle \tag{87}$$

Thus, if can establish the conservation law in this case. we can get the dynamics of $z$ which is solvable. For given small $\delta$, we define

$$T_2 = \sup \left\{ t \geq 0 \mid \|\pi - p_{\text{sym}}\| \geq \delta \right\}, \tag{88}$$

$$T_3 = \sup \left\{ t \geq 0 \mid W_{\text{out,max}} \lesssim 1 \right\} \tag{89}$$

and give a estimate of $T_2, T_3$ by

$$T_d = T_p + \frac{2}{\|\pi\| \sqrt{\frac{d}{d-1}} \, n \left( \pi_1 - \frac{1}{d} \right)} \log \frac{1}{\varepsilon}. \tag{90}$$

Then first we get similar error estimate, the proof is similar to Lemma B.1. Thus we omit its proof.

**Lemma B.4** (Refined error estimate). *For every $t \leq \min\{T_2, T_3, T_d\}$, one has*

$$\|W_Q(t)\|_{\text{F}} + \|W_K(t)\|_{\text{F}} \leq C\varepsilon^{\frac{2}{3}}, \tag{91}$$

*and*

$$\|R_0(t)\|_{\text{F}} + \|R_1(t)\|_{\text{F}} \leq C\varepsilon. \tag{92}$$

Then we show the monotonicity of key variables. Before we do this, we derive the aprroximate conservation law:

$$\frac{\mathrm{d}}{\mathrm{d}t} \langle a, b \rangle - \frac{1}{2}(\|a\|^2 + \|b\|^2) = \|\pi\| \|\pi - p_{\text{sym}}(z)\| (\|a\|^2 + \|b\|^2 - 2\langle a, b \rangle) + \langle \mathcal{E}_a, b \rangle + \langle \mathcal{E}_b, a \rangle - \langle \mathcal{E}_a, b \rangle - \langle \mathcal{E}_b, a \rangle$$
$$\geq \langle \mathcal{E}_a, b \rangle + \langle \mathcal{E}_b, a \rangle - \langle \mathcal{E}_a, b \rangle - \langle \mathcal{E}_b, a \rangle \tag{93}$$

Then we have

$$\langle a, b \rangle(t) - \frac{1}{2}(\|a\|^2 + \|b\|^2)(t) \gtrsim -\varepsilon^{\frac{3}{n}} - \int_{T_p}^t |\langle \mathcal{E}_a, b \rangle| + |\langle \mathcal{E}_b, a \rangle| + |\langle \mathcal{E}_a, b \rangle| + |\langle \mathcal{E}_b, a \rangle| \mathrm{d}s \tag{94}$$

For $t$ such that $\frac{1}{2}(\|a\|^2 + \|b\|^2)(t) \gtrsim \varepsilon^{\frac{2}{n}}$ and $t \leq \min\{T_2, T_3, T_d\}$, the right hand can be controlled by

$$\varepsilon^{\frac{1}{2n}}(\|a\|^2 + \|b\|^2) \tag{95}$$

As a result, we get for $t$ such that $\frac{1}{2}(\|a\|^2 + \|b\|^2)(t) \gtrsim \varepsilon^{\frac{2}{n}}$ and $t \leq \min\{T_2, T_3, T_d\}$, there is

$$(1 - \varepsilon^{\frac{1}{n}}) \frac{1}{2}(\|a\|^2 + \|b\|^2)(t) \leq \langle a, b \rangle(t) \leq \frac{1}{2}(\|a\|^2 + \|b\|^2)(t) \tag{96}$$

**Proposition B.5.** *For $T_p \leq t \leq \min\{T_2, T_3, T_d\}$, one has $\langle a, b \rangle$, $\|a\|^2 + \|b\|^2$ increase monotonically.*

*Proof.* We begin the discussion with $z = \langle a, b \rangle$. Taking the derivative, we get

$$\dot{z} = \|\pi\| \|\pi - p_{\text{sym}}(z)\| (\|a\|^2 + \|b\|^2) + \langle \mathcal{E}_a, b \rangle + \langle \mathcal{E}_b, a \rangle \tag{97}$$

From the estimate at $T_p$, we find that

$$|\langle \mathcal{E}_a, b \rangle| \lesssim \varepsilon^{\frac{1}{2}} \delta \|b\|^2, \quad |\langle \mathcal{E}_b, a \rangle| \lesssim \varepsilon^{\frac{1}{2}} \delta \|a\|^2 \tag{98}$$

Here, we use the fact that $\|\mathcal{E}_a\|, \|\mathcal{E}_b\| \lesssim \varepsilon$ and $\|a\|, \|b\| \gtrsim \varepsilon^{\frac{1}{n}}$. Using the continuity, we get $z$ will increase monotonically at a time period. Then we consider $\|a\|^2 + \|b\|^2$:

$$\frac{\mathrm{d}}{\mathrm{d}t}\left(\|a\|^2 + \|b\|^2\right) = 4\|\pi\|\|\pi - p_{\mathrm{sym}}\|\langle a, b\rangle + 2\langle \mathcal{E}_a, b\rangle + 2\langle \mathcal{E}_b, a\rangle$$

$$\geq 2\|\pi\|\|\pi - p_{\mathrm{sym}}\|(1 - \varepsilon^{\frac{1}{n}} - \varepsilon^{\frac{1}{2}})(\|a\|^2 + \|b\|^2) \tag{99}$$

Thus $\|a\|^2 + \|b\|^2$ will increase at a time period. Since the above estimate we use will maintain if $z$ and $\|a\|^2 + \|b\|^2$ increase. We find that $z$ and $\|a\|^2 + \|b\|^2$ will increase during $T_p \leq t \leq \min\{T_2, T_3, T_d\}$. □

We are now ready to finish the entire cross-stage analysis. First, we find that during $T_p \leq t \leq \min\{T_2, T_3, T_d\}$, and there exists a constant

$$c_\pi := \frac{\|\pi\|}{\|\pi - \frac{1}{d}\mathbf{1}\|}\left(\pi_1 - \pi_{i \neq 1}\right) \tag{100}$$

such that

$$p_{\mathrm{sym},1}(z) = \frac{1}{1 + (d-1)e^{-c_\pi z}}, \qquad p_{\mathrm{sym},j}(z) = \frac{1 - p_{\mathrm{sym},1}(z)}{d - 1}, \quad j \geq 2. \tag{101}$$

Consequently,

$$\|\pi - p_{\mathrm{sym}}(z)\| = \sqrt{\frac{d}{d-1}}\left(\pi_1 - p_{\mathrm{sym},1}(z)\right). \tag{102}$$

Thus, we get the following dynamics based on similar estimate used in Proposition B.5

$$(2 - \varepsilon^{\frac{1}{2n}})\|\pi\|\sqrt{\frac{d}{d-1}}\,z\left(\pi_1 - \frac{1}{1 + (d-1)e^{-c_\pi z}}\right) \leq \dot{z} \leq (2 + \varepsilon^{\frac{1}{2n}})\|\pi\|\sqrt{\frac{d}{d-1}}\,z\left(\pi_1 - \frac{1}{1 + (d-1)e^{-c_\pi z}}\right). \tag{103}$$

Here we abuse the notation between critical point and critical value. We identify the critical value

$$\theta_c^1 := \frac{1}{c_\pi}\log\frac{(d-1)\pi_1}{1 - \pi_1}, \tag{104}$$

which is exactly the unique solution of

$$p_{\mathrm{sym},1}(\theta_c^1) = \pi_1.$$

Let

$$A_\pm := (2 \pm \varepsilon^{\frac{1}{2n}})\|\pi\|\sqrt{\frac{d}{d-1}}, \qquad G(z) := \pi_1 - \frac{1}{1 + (d-1)e^{-c_\pi z}}. \tag{105}$$

Then the reduced dynamics satisfies

$$A_- zG(z) \leq \dot{z} \leq A_+ zG(z). \tag{106}$$

Recall that the critical value is

$$\theta_c^1 = \frac{1}{c_\pi}\log\frac{(d-1)\pi_1}{1 - \pi_1}, \tag{107}$$

and $G(\theta_c^1) = 0$. Moreover, $G(z) > 0$ for $0 < z < \theta_c^1$, hence $z(t)$ is strictly increasing as long as $z(t) < \theta_c^1$.

For $0 < \delta \ll 1$, define $z_\delta \in (0, \theta_c^1)$ by

$$G(z_\delta) = \delta. \tag{108}$$

Equivalently,

$$z_\delta = \frac{1}{c_\pi}\log\frac{(d-1)(\pi_1 - \delta)}{1 - \pi_1 + \delta}. \tag{109}$$

Then the definition of $T_2$ can be rewritten as

$$T_2 := \inf\{t \geq T_p : z(t) \geq z_\delta\}. \tag{110}$$

This is exactly the first time when

$$\pi_1 - \frac{1}{1 + (d-1)e^{-c_\pi z(t)}} = O(\delta).$$

**Proposition B.6** (Time to the $O(\delta)$-neighborhood of the critical point). *Assume*

$$z(T_p) = \Theta\left(\varepsilon^{\frac{2}{n}}\right).$$

*Then*

$$\frac{1}{A_+} \int_{z(T_p)}^{z_\delta} \frac{dz}{z\,G(z)} \leq T_2 - T_p \leq \frac{1}{A_-} \int_{z(T_p)}^{z_\delta} \frac{dz}{z\,G(z)}. \tag{111}$$

*Moreover, there exists a constant $C > 0$, independent of sufficiently small $\varepsilon, \delta$, such that*

$$\left| \int_{z(T_p)}^{z_\delta} \frac{dz}{z\,G(z)} - \frac{1}{\pi_1 - \frac{1}{d}} \log \frac{1}{z(T_p)} - \frac{1}{\theta_c^1 c_\pi \pi_1 (1 - \pi_1)} \log \frac{1}{\delta} \right| \leq C. \tag{112}$$

*Consequently,*

$$T_2 - T_p = \frac{1}{2\|\pi\|\sqrt{\frac{d}{d-1}}} \left[ \frac{2}{n\left(\pi_1 - \frac{1}{d}\right)} \log \frac{1}{\varepsilon} + \frac{1}{\theta_c^1 c_\pi \pi_1 (1 - \pi_1)} \log \frac{1}{\delta} \right] + O(1). \tag{113}$$

*In particular, if $\delta > 0$ is fixed, then*

$$T_2 - T_p = \frac{1}{\|\pi\|\sqrt{\frac{d}{d-1}}\, n\left(\pi_1 - \frac{1}{d}\right)} \log \frac{1}{\varepsilon} + O(1). \tag{114}$$

*Proof.* Since $G(z) > 0$ on $(0, \theta_c^1)$, the solution is monotone increasing there. Separating variables in (106) yields (111) immediately.

Hence it remains to estimate the integral

$$I(z_0, z_\delta) := \int_{z_0}^{z_\delta} \frac{dz}{z\,G(z)}, \qquad z_0 := z(T_p).$$

The point is that the integrand has two logarithmic singularities: one at $z = 0$, coming from the factor $1/z$, and one at $z = \theta_c^1$, coming from the simple zero of $G$.

First, near $z = 0$,

$$G(z) = G(0) + O(z) = \left(\pi_1 - \frac{1}{d}\right) + O(z), \tag{115}$$

since

$$G(0) = \pi_1 - \frac{1}{1 + (d-1)} = \pi_1 - \frac{1}{d}.$$

Therefore

$$\frac{1}{z\,G(z)} = \frac{1}{\pi_1 - \frac{1}{d}} \frac{1}{z} + O(1), \qquad z \to 0. \tag{116}$$

Next, near $z = \theta_c^1$, we use that $G(\theta_c^1) = 0$ and

$$G'(z) = -\frac{c_\pi(d-1)e^{-c_\pi z}}{\left(1 + (d-1)e^{-c_\pi z}\right)^2}. \tag{117}$$

By the defining relation of $\theta_c^1$,

$$(d-1)e^{-c_\pi \theta_c^1} = \frac{1 - \pi_1}{\pi_1},$$

hence

$$G'(\theta_c^1) = -c_\pi \pi_1 (1 - \pi_1). \tag{118}$$

Thus

$$G(z) = c_\pi \pi_1 (1 - \pi_1)(\theta_c^1 - z) + O\left((\theta_c^1 - z)^2\right), \qquad z \to \theta_c^1-, \tag{119}$$

and so

$$\frac{1}{z\,G(z)} = \frac{1}{\theta_c^1 c_\pi \pi_1 (1-\pi_1)} \frac{1}{\theta_c^1 - z} + O(1), \qquad z \to \theta_c^1 - . \tag{120}$$

Therefore, after subtracting the two poles,

$$R(z) := \frac{1}{z\,G(z)} - \frac{1}{\pi_1 - \frac{1}{d}} \frac{1}{z} - \frac{1}{\theta_c^1 c_\pi \pi_1 (1-\pi_1)} \frac{1}{\theta_c^1 - z} \tag{121}$$

extends to a bounded function on $(0, \theta_c^1)$. Integrating, we obtain

$$\begin{aligned}
I(z_0, z_\delta) &= \frac{1}{\pi_1 - \frac{1}{d}} \log \frac{z_\delta}{z_0} + \frac{1}{\theta_c^1 c_\pi \pi_1 (1-\pi_1)} \log \frac{\theta_c^1 - z_0}{\theta_c^1 - z_\delta} + O(1) \\
&= \frac{1}{\pi_1 - \frac{1}{d}} \log \frac{1}{z_0} + \frac{1}{\theta_c^1 c_\pi \pi_1 (1-\pi_1)} \log \frac{1}{\theta_c^1 - z_\delta} + O(1),
\end{aligned} \tag{122}$$

which is (112) up to the relation between $\theta_c^1 - z_\delta$ and $\delta$.

Now from (109),

$$\theta_c^1 - z_\delta = \frac{1}{c_\pi} \log \frac{\pi_1 (1 - \pi_1 + \delta)}{(1-\pi_1)(\pi_1 - \delta)} = \frac{\delta}{c_\pi \pi_1 (1-\pi_1)} + O(\delta^2), \tag{123}$$

hence

$$\log \frac{1}{\theta_c^1 - z_\delta} = \log \frac{1}{\delta} + O(1). \tag{124}$$

Also, by assumption,

$$z(T_p) = \Theta\left(\varepsilon^{\frac{2}{n}}\right), \qquad \text{so} \qquad \log \frac{1}{z(T_p)} = \frac{2}{n} \log \frac{1}{\varepsilon} + O(1). \tag{125}$$

Substituting these two estimates into (112) gives

$$I(z(T_p), z_\delta) = \frac{2}{n\left(\pi_1 - \frac{1}{d}\right)} \log \frac{1}{\varepsilon} + \frac{1}{\theta_c^1 c_\pi \pi_1 (1-\pi_1)} \log \frac{1}{\delta} + O(1). \tag{126}$$

Finally, substituting this into (111) and using

$$A_\pm^{-1} = \frac{1}{2\|\pi\|\sqrt{\frac{d}{d-1}}} + o(1)$$

yields (113). $\qquad\square$

Finally, we can prove that $T_2 \leq \min\{T_3, T_d\}$ by direct computation. Thus, we finish the cross stage analysis and show that parameters will enter into a $\delta$ neighborhood of $\theta_c^1$.

### B.2. Theoretical details in Sec. 4.2

**Proof of Proposition 4.3**

*Proof.* Since attention parameters $(W_Q, W_K)$ are chosen to be zero, for any $i$ we have $\mathbb{A}_i = \pi^\intercal$, and hence $\mathbb{P}_i = \mathbb{P}_j$ for $i \neq j$. By definition,

$$\mathbb{P}_{i,j} = \frac{\exp\left(\kappa^2 \|\pi\|^2 (\pi_j - \frac{1}{|\mathcal{V}|})\right)}{\sum_{j'} \exp\left(\kappa^2 \|\pi\|^2 (\pi_{j'} - \frac{1}{|\mathcal{V}|})\right)} = \frac{\exp\left(\kappa^2 \|\pi\|^2 \pi_j\right)}{\sum_{j'} \exp(\kappa^2 \|\pi\|^2 \pi_{j'})}. \tag{127}$$

As $\kappa \to 0$, $\mathbb{P}_i \to \frac{1}{|\mathcal{V}|} \mathbf{1}^\intercal$; as $\kappa \to \infty$, $\mathbb{P}_i \to e_1^\intercal$ since $\pi_1 = \max_j \pi_j$. The map $\kappa \mapsto \mathbb{P}_{i,1}$ is continuous, hence by the intermediate value theorem there exists $\kappa_1 > 0$ such that $\mathbb{P}_{i,1} = \pi_1$. Together with the symmetry assumption $\pi_i = \pi_j$ for $i, j \geq 2$, this implies $\mathbb{P}_i = \pi^\intercal$.

Substituting $\mathbb{P}_i = \pi^\mathsf{T}$ and $\mathbb{A}_i = \pi^\mathsf{T}$ into $\frac{\partial \mathcal{L}}{\partial M} = -\sum_{i=1}^{|\mathcal{V}|} \pi_i \mathbb{A}_i^\mathsf{T}(P_i - \mathbb{P}_i)$ yields

$$\frac{\partial \mathcal{L}}{\partial M} = -\pi \sum_{i=1}^{|\mathcal{V}|} \pi_i (P_i - \pi^\mathsf{T}). \tag{128}$$

Using $\pi^\mathsf{T} P = \pi^\mathsf{T}$, we obtain $\frac{\partial \mathcal{L}}{\partial M} = 0$. Finally, $W_Q = W_K = 0$ at this point, so it is indeed a critical point. $\qquad \square$

We first record the derivatives needed for the linearization.

**Proposition B.7** (Derivatives at the second critical point). *At $\theta_c^1$ in (12), we have*

$$\left. \frac{\partial \mathcal{L}}{\partial \Phi} \right|_{\theta_c^1} = -\lambda \kappa_1^2 \operatorname{Var}(\pi) \frac{\pi - \frac{1}{|\mathcal{V}|}\mathbf{1}}{\left\| \pi - \frac{1}{|\mathcal{V}|}\mathbf{1} \right\|} \frac{\pi^\mathsf{T}}{\|\pi\|} \operatorname{Var}(\pi), \tag{129}$$

*and the total differential of $\frac{\partial \mathcal{L}}{\partial M}$ satisfies*

$$\left. \mathrm{d}\frac{\partial \mathcal{L}}{\partial M} \right|_{\theta_c^1} = \pi\pi^\mathsf{T} \, \mathrm{d}M \left( \operatorname{diag}(\pi) - \pi\pi^\mathsf{T} \right). \tag{130}$$

**Proof of Proposition B.7**

*Proof.* First, we compute the specific expression of $\frac{\partial \mathcal{L}}{\partial \Phi}$. Using the expression derived in Eq. (5),

$$\frac{\partial \mathcal{L}}{\partial \Phi} = -\sum_i \pi_i e_i \left( P_i - \mathbb{P}_i \right) M^\mathsf{T} \left( \operatorname{diag}(\mathbb{A}_i^\mathsf{T}) - \mathbb{A}_i^\mathsf{T}\mathbb{A}_i \right)$$

Using the fact that $\mathbb{P}_i = \pi^\mathsf{T}$ and $\mathbb{A}_i = \pi^\mathsf{T}$ and substituting $M = \kappa_1^2 \frac{\pi}{\|\pi\|} \frac{\pi^\mathsf{T} - \frac{1}{d}\mathbf{1}^\mathsf{T}}{\|\pi - \frac{1}{d}\mathbf{1}\|}$ into the equation, we get

$$\begin{aligned} \frac{\partial \mathcal{L}}{\partial \Phi} &= -\kappa_1^2 \sum_i \pi_i e_i (P_i - \pi^\mathsf{T}) \frac{\pi - \frac{1}{d}\mathbf{1}}{\|\pi - \frac{1}{d}\mathbf{1}\|} \frac{\pi^\mathsf{T}}{\|\pi\|} (\operatorname{diag}(\pi) - \pi\pi^\mathsf{T}) \\ &= -\kappa_1^2 \left( \operatorname{diag}(\pi)P - \pi\pi^\mathsf{T} \right) \frac{\pi - \frac{1}{d}\mathbf{1}}{\|\pi - \frac{1}{d}\mathbf{1}\|} \frac{\pi^\mathsf{T}}{\|\pi\|} (\operatorname{diag}(\pi) - \pi\pi^\mathsf{T}) \end{aligned} \tag{131}$$

By the definition of $P$, we have

$$\begin{aligned} \operatorname{diag}(\pi)P - \pi\pi^\mathsf{T} &= \operatorname{diag}(\pi) \left( \lambda I + (1-\lambda)\mathbf{1}\pi^\mathsf{T} \right) - \pi\pi^\mathsf{T} \\ &= \lambda \left( \operatorname{diag}(\pi) - \pi\pi^\mathsf{T} \right) \end{aligned} \tag{132}$$

Combining Eqs (131) and (132), we get the expression of $\frac{\partial \mathcal{L}}{\partial \Phi}$ at $\theta_c^1$. Then, we consider the total differential of the gradient of the loss function with respect to $M$. Firstly, using Eq. (5) again, we get

$$\frac{\partial \mathcal{L}}{\partial M} = -\sum_i \pi_i \mathbb{A}_i^\mathsf{T}(P_i - \mathbb{P}_i)$$

By chain rule, we get

$$\mathrm{d}\frac{\partial \mathcal{L}}{\partial M} = -\sum_i \pi_i \mathrm{d}\mathbb{A}_i^\mathsf{T}(P_i - \mathbb{P}_i) - \sum_i \pi_i \mathbb{A}_i^\mathsf{T}(-\mathrm{d}\mathbb{P}_i). \tag{133}$$

Let's consider these two items separately. By the definition of $\mathbb{A}_i$, we find that

$$\begin{aligned} \mathrm{d}\mathbb{A}_{i,j} &= \mathbb{A}_{i,j}e_i^\mathsf{T}\mathrm{d}\Phi e_j - \mathbb{A}_{i,j}\sum_{j'}\mathbb{A}_{i,j'}e_i^\mathsf{T}\mathrm{d}\Phi e_{j'} \\ &= \mathbb{A}_{i,j}e_i^\mathsf{T}\mathrm{d}\Phi(e_j - \mathbb{A}_i^\mathsf{T}) \end{aligned} \tag{134}$$

As a result, it can be verified that $\mathrm{d}\mathbb{A}_i = e_i^\mathsf{T}\mathrm{d}\Phi\,(\mathrm{diag}(\mathbb{A}_i^\mathsf{T}) - \mathbb{A}_i^\mathsf{T}\mathbb{A}_i)$. However, this term will be zero because it contains the intersection terms $W_Q W_K^\mathsf{T}$ which will be zero by the chain rule and the condition $W_Q = 0$ and $W_K = 0$. Thus, we focus on the second term. Recall the definition of $\mathbb{P}_{i,j} = \frac{\exp(\mathbb{A}_i M e_j)}{\sum_{j'}\exp(\mathbb{A}_i M e_{j'})}$ and take the total differential of it:

$$
\begin{aligned}
\mathrm{d}\mathbb{P}_{i,j} &= \mathbb{P}_{i,j}\mathrm{d}(\mathbb{A}_i M)e_j - \mathbb{P}_{i,j}\sum_{j'}\mathbb{P}_{i,j'}\mathrm{d}(\mathbb{A}_i M)e_{j'} \\
&= \mathbb{P}_{i,j}\mathrm{d}(\mathbb{A}_i M)(e_j - \mathbb{P}_i^\mathsf{T}).
\end{aligned}
\tag{135}
$$

Similar to the derivation of $\mathrm{d}\mathbb{A}_i$, $\mathrm{d}\mathbb{P}_i$ can be reformulated as $\mathrm{d}(\mathbb{A}_i M)(\mathrm{diag}(\mathbb{P}_i^\mathsf{T}) - \mathbb{P}_i^\mathsf{T}\mathbb{P}_i)$. In particular, at this critical point,

$$
\mathrm{d}\mathbb{P}_i = \mathbb{A}_i\mathrm{d}M(\mathrm{diag}(\mathbb{P}_i^\mathsf{T}) - \mathbb{P}_i^\mathsf{T}\mathbb{P}_i) = \pi^\mathsf{T}\mathrm{d}M\,\mathrm{Var}(\pi).
\tag{136}
$$

Substitute this expression into Eq. (133), we get

$$
\mathrm{d}\frac{\partial\mathcal{L}}{\partial M} = \sum_i \pi_i\mathbb{A}_i^\mathsf{T}\pi^\mathsf{T}\mathrm{d}M\,\mathrm{Var}(\pi) = \pi\pi^\mathsf{T}\mathrm{d}M\,\mathrm{Var}(\pi).
\tag{137}
$$

$\square$

**Proof of Proposition 4.4**

*Proof.* We linearize the gradient flow (4) at $\theta_c^1$. Since $\frac{\partial\mathcal{L}}{\partial M}\big|_{\theta_c^1} = 0$ and $W_Q = W_K = 0$, the only first-order contribution in the $(W_0, W_1)$ subsystem comes from the first variation of $\frac{\partial\mathcal{L}}{\partial M}$, whereas the $(W_Q, W_K)$ subsystem is driven by the constant matrix $\frac{\partial\mathcal{L}}{\partial\Phi}\big|_{\theta_c^1}$. More specifically, the linearized subsystems with respect to $(W_0, W_1)$ and $(W_Q, W_K)$ are two decoupled systems which separately follow

$$
\begin{aligned}
\frac{\mathrm{d}\Delta W_0}{\mathrm{d}t} &= \mathrm{d}\frac{\partial\mathcal{L}}{\partial M}W_1^\mathsf{T} \\
\frac{\mathrm{d}\Delta W_1}{\mathrm{d}t} &= W_0^\mathsf{T}\mathrm{d}\frac{\partial\mathcal{L}}{\partial M}
\end{aligned}
\tag{138}
$$

and

$$
\begin{aligned}
\frac{\mathrm{d}\Delta W_Q}{\mathrm{d}t} &= -W_0^\mathsf{T}\frac{\partial\mathcal{L}}{\partial\Phi}W_0\Delta W_K \\
\frac{\mathrm{d}\Delta W_K}{\mathrm{d}t} &= -W_0^\mathsf{T}\left(\frac{\partial\mathcal{L}}{\partial\Phi}\right)^\mathsf{T}W_0\Delta W_Q
\end{aligned}
\tag{139}
$$

**Step 1: the $(W_0, W_1)$-subsystem is contracting along $\alpha_1$.** By Proposition B.7,

$$
\mathrm{d}\left(\frac{\partial\mathcal{L}}{\partial M}\right)\bigg|_{\theta_c^1} = \pi\pi^\mathsf{T}\,\mathrm{d}M\,\mathrm{Var}(\pi)
$$

At $\theta_c^1$, Proposition 4.3 gives the rank-one form

$$
W_{0,0} = \kappa_1 q\,\alpha_1^\mathsf{T}, \qquad W_{1,0} = \kappa_1\alpha_1\,u^\mathsf{T}, \qquad q := \frac{\pi}{\|\pi\|}, \qquad u := \frac{\pi - \frac{1}{|\mathcal{V}|}\mathbf{1}}{\|\pi - \frac{1}{|\mathcal{V}|}\mathbf{1}\|}.
\tag{140}
$$

A direct substitution into Eq. (138) shows that for any $v \perp \alpha_1$,

$$
\frac{\mathrm{d}}{\mathrm{d}t}\Delta W_0\,v = 0, \qquad \frac{\mathrm{d}}{\mathrm{d}t}v^\mathsf{T}\Delta W_1 = 0,
$$

i.e. the linearization is degenerate in the normal directions.

Therefore we focus on the $\alpha_1$-component and calculate the specific expansion:

$$
\begin{aligned}
\frac{\mathrm{d}\Delta W_0\alpha_1}{\mathrm{d}t} &= -\pi\pi^{\mathsf{T}}(\Delta W_0 W_1 + W_0\Delta W_1)\operatorname{Var}(\pi)W_1^{\mathsf{T}} \\
&= -\kappa_1^2\pi\pi^{\mathsf{T}}\Delta W_0\alpha_1\frac{\pi^{\mathsf{T}} - \frac{1}{d}\mathbf{1}^{\mathsf{T}}}{\|\pi - \frac{1}{d}\mathbf{1}\|}\operatorname{Var}(\pi)\frac{\pi - \frac{1}{d}\mathbf{1}}{\|\pi - \frac{1}{d}\mathbf{1}\|} \\
&\quad - \kappa_1^2\pi\pi^{\mathsf{T}}\frac{\pi}{\|\pi\|}\alpha_1^{\mathsf{T}}\Delta W_1\operatorname{Var}(\pi)\frac{\pi - \frac{1}{d}\mathbf{1}}{\|\pi - \frac{1}{d}\mathbf{1}\|}
\end{aligned}
\tag{141}
$$

and

$$
\begin{aligned}
\frac{\mathrm{d}\Delta W_1^{\mathsf{T}}\alpha_1}{\mathrm{d}t} &= -\operatorname{Var}(\pi)(W_1^{\mathsf{T}}\Delta W_0^{\mathsf{T}} + \Delta W_1^{\mathsf{T}}W_0^{\mathsf{T}})\pi\pi^{\mathsf{T}}W_0 \\
&= -\kappa_1^2\operatorname{Var}(\pi)\frac{\pi - \frac{1}{d}\mathbf{1}}{\|\pi - \frac{1}{d}\mathbf{1}\|}\alpha_1^{\mathsf{T}}\Delta W_0^{\mathsf{T}}\pi\pi^{\mathsf{T}}\frac{\pi}{\|\pi\|} \\
&\quad - \kappa_1^2\operatorname{Var}(\pi)\Delta W_1^{\mathsf{T}}\alpha_1\frac{\pi^{\mathsf{T}}}{\|\pi\|}\pi\pi^{\mathsf{T}}\frac{\pi}{\|\pi\|}
\end{aligned}
\tag{142}
$$

Finally, we have

$$
\mathrm{d}\begin{pmatrix}\Delta W_0\alpha_1 \\ \Delta W_1^{\mathsf{T}}\alpha_1\end{pmatrix} = -\kappa_1^2\begin{pmatrix} \frac{\pi^{\mathsf{T}} - \frac{1}{d}\mathbf{1}^{\mathsf{T}}}{\|\pi - \frac{1}{d}\mathbf{1}\|}\operatorname{Var}(\pi)\frac{\pi - \frac{1}{d}\mathbf{1}}{\|\pi - \frac{1}{d}\mathbf{1}\|}\pi\pi^{\mathsf{T}} & \pi^{\mathsf{T}}\frac{\pi}{\|\pi\|}\pi\frac{\pi^{\mathsf{T}} - \frac{1}{d}\mathbf{1}^{\mathsf{T}}}{\|\pi - \frac{1}{d}\mathbf{1}\|}\operatorname{Var}(\pi) \\ \pi^{\mathsf{T}}\frac{\pi}{\|\pi\|}\operatorname{Var}(\pi)\frac{\pi - \frac{1}{d}\mathbf{1}}{\|\pi - \frac{1}{d}\mathbf{1}\|}\pi^{\mathsf{T}} & \frac{\pi^{\mathsf{T}}}{\|\pi\|}\pi\pi^{\mathsf{T}}\frac{\pi}{\|\pi\|}\operatorname{Var}(\pi) \end{pmatrix}\begin{pmatrix}\Delta W_0\alpha_1 \\ \Delta W_1^{\mathsf{T}}\alpha_1\end{pmatrix}
\tag{143}
$$

We introduce the notations:

$$
x := \Delta W_0\alpha_1 \in \mathbb{R}^{|\mathcal{V}|}, \qquad y := \Delta W_1^{\mathsf{T}}\alpha_1 \in \mathbb{R}^{|\mathcal{V}|}.
$$

Eq. (143) can be rewritten in the following concise form

$$
\frac{\mathrm{d}}{\mathrm{d}t}\begin{pmatrix}x \\ y\end{pmatrix} = -\kappa_1^2 A\begin{pmatrix}x \\ y\end{pmatrix},
\tag{144}
$$

where

$$
A := \begin{pmatrix} a\,\pi\pi^{\mathsf{T}} & b\,\pi\,u^{\mathsf{T}}C \\ b\,Cu\,\pi^{\mathsf{T}} & b^2 C \end{pmatrix}, \qquad a := u^{\mathsf{T}}Cu, \quad b := \pi^{\mathsf{T}}q = \|\pi\|.
\tag{145}
$$

The matrix $A$ is symmetric by construction. Moreover, for arbitrary $x, y$ define $\alpha := \pi^{\mathsf{T}}x$ and $\beta := u^{\mathsf{T}}Cy$. Then the quadratic form is

$$
\begin{pmatrix}x^{\mathsf{T}} & y^{\mathsf{T}}\end{pmatrix} A \begin{pmatrix}x \\ y\end{pmatrix} = a\,\alpha^2 + 2b\,\alpha\beta + b^2\,y^{\mathsf{T}}Cy.
$$

Introducing the $C$-inner product $\langle v, w\rangle_C := v^{\mathsf{T}}Cw$ (with seminorm $\|v\|_C = \sqrt{v^{\mathsf{T}}Cv}$), we have $a = \|u\|_C^2 \geq 0$ and $|\beta| = |\langle u, y\rangle_C| \leq \|u\|_C\|y\|_C = \sqrt{a}\sqrt{y^{\mathsf{T}}Cy}$. Hence

$$
a\,\alpha^2 + 2b\,\alpha\beta + b^2\,y^{\mathsf{T}}Cy \geq \left(\sqrt{a}\,|\alpha| - b\sqrt{y^{\mathsf{T}}Cy}\right)^2 \geq 0,
$$

so $A \succeq 0$. Therefore all eigenvalues of $-\lambda^2 A$ in (144) are non-positive, and the $(x, y)$-subsystem is contracting (or neutrally stable in the degenerate directions).

**Step 2: effective coupling for $(W_Q, W_K)$.** Recall Eq. (139),

$$
\frac{\mathrm{d}}{\mathrm{d}t}\Delta W_Q = -W_0^{\mathsf{T}}\left.\frac{\partial\mathcal{L}}{\partial\Phi}\right|_{\theta_0}W_0\,\Delta W_K, \quad \frac{\mathrm{d}}{\mathrm{d}t}\Delta W_K = -W_0^{\mathsf{T}}\left(\frac{\partial\mathcal{L}}{\partial\Phi}\right)^{\mathsf{T}}\bigg|_{\theta_0}W_0\Delta W_Q
$$

Substitute the expression of $\frac{\partial\mathcal{L}}{\partial\Phi}$ into above equation, we take $\frac{\mathrm{d}\Delta W_Q}{\mathrm{d}t}$ as an example:

$$
\begin{aligned}
\frac{\mathrm{d}\Delta W_Q}{\mathrm{d}t} &= \kappa_1^2\alpha_1\frac{\pi^{\mathsf{T}}}{\|\pi\|}\lambda\kappa_1^2\Big(\operatorname{diag}(\pi) - \pi\pi^{\mathsf{T}}\Big)\frac{\pi - \frac{1}{|\mathcal{V}|}\mathbf{1}}{\|\pi - \frac{1}{|\mathcal{V}|}\mathbf{1}\|}\frac{\pi^{\mathsf{T}}}{\|\pi\|}\Big(\operatorname{diag}(\pi) - \pi\pi^{\mathsf{T}}\Big)\frac{\pi}{\|\pi\|}\alpha_1^{\mathsf{T}}\Delta W_K \\
&= c_1\alpha_1\alpha_1^{\mathsf{T}}\Delta W_K.
\end{aligned}
\tag{146}
$$

Left-multiplying by $\alpha_1^\mathsf{T}$ yields the results. Moreover, $c_1$ is positive by its definition. Thus, it is an unstable direction. It implies that the effective dynamics near the critical point is the subsystem about $W_Q$ and $W_K$

$\square$

### B.3. Theoretical details in Sec. 4.3

**Proof of Proposition 4.6** We complete the proof of Proposition 1 in two steps. First, we directly verify that a rank-one manifold is an invariant manifold. Then, we utilize data symmetry and permutation equivariance to prove the conservation of low-frequency tokens.

*Proof.* (i) Invariance of the rank-one form. Plug (18) into (4) and check that each right-hand side remains in the same rank-one span.

Since $W_1^\mathsf{T} = \beta\,\alpha_1^\mathsf{T}$,

$$-\frac{\partial\mathcal{L}}{\partial M}W_1^\mathsf{T} = -\Big(\frac{\partial\mathcal{L}}{\partial M}\beta\Big)\alpha_1^\mathsf{T},$$

which is of the form $\dot\gamma\,\alpha_1^\mathsf{T}$.

Next, using $\tilde\alpha_1^\mathsf{T}\tilde\alpha_1 = 1$,

$$W_K W_Q^\mathsf{T} = \lambda_K\lambda_Q\,\alpha_1(\tilde\alpha_1^\mathsf{T}\tilde\alpha_1)\alpha_1^\mathsf{T} = \eta\,\alpha_1\alpha_1^\mathsf{T}, \qquad W_0 W_K W_Q^\mathsf{T} = \eta\,\gamma\,(\alpha_1^\mathsf{T}\alpha_1)\alpha_1^\mathsf{T} = \eta\,\gamma\,\alpha_1^\mathsf{T}.$$

Therefore the $\Phi$-driven terms in $\dot W_0$ satisfy

$$-\frac{\partial\mathcal{L}}{\partial\Phi}W_0 W_K W_Q^\mathsf{T} = -\eta\Big(\frac{\partial\mathcal{L}}{\partial\Phi}\gamma\Big)\alpha_1^\mathsf{T}, \quad -\Big(\frac{\partial\mathcal{L}}{\partial\Phi}\Big)^\mathsf{T}W_0 W_Q W_K^\mathsf{T} = -\eta\Big(\Big(\frac{\partial\mathcal{L}}{\partial\Phi}\Big)^\mathsf{T}\gamma\Big)\alpha_1^\mathsf{T},$$

so $\dot W_0$ stays in the span of $\{\cdot\,\alpha_1^\mathsf{T}\}$ and hence $W_0(t) = \gamma(t)\alpha_1^\mathsf{T}$.

Similarly, since $W_0^\mathsf{T} = \alpha_1\gamma^\mathsf{T}$,

$$\dot W_1 = -W_0^\mathsf{T}\frac{\partial\mathcal{L}}{\partial M} = -\alpha_1\Big(\gamma^\mathsf{T}\frac{\partial\mathcal{L}}{\partial M}\Big),$$

which is of the form $\alpha_1\,\dot\beta^\mathsf{T}$.

Finally,

$$\dot W_Q = -W_0^\mathsf{T}\frac{\partial\mathcal{L}}{\partial\Phi}W_0 W_K = -\lambda_K\Big(\gamma^\mathsf{T}\frac{\partial\mathcal{L}}{\partial\Phi}\gamma\Big)\alpha_1\tilde\alpha_1^\mathsf{T},$$

so $W_Q$ remains in the form $\lambda_Q(t)\alpha_1\tilde\alpha_1^\mathsf{T}$. The argument for $W_K$ is identical. Thus the flow stays in $\mathcal{W}$.

(ii) Preservation of the low-frequency symmetry. Let

$$G := \{\sigma : \{1,\ldots,V\} \to \{1,\ldots,V\}\,|\,\sigma(1) = 1\}. \tag{147}$$

and let $\sigma \in G$ be the permutation matrix. Define the group action as

$$\rho_\sigma(\theta) := (\Pi_\sigma W_0, W_1\Pi_\sigma^\mathsf{T}, W_Q, W_K).$$

Under this action, ones check that $\mathbb{P}_{i,j}(\rho_\sigma(\theta)) = \mathbb{P}_{\sigma(i),\sigma(j)}(\theta)$. Since the loss function can be viewed as $\mathcal{L}(\theta) = -\sum_i \pi_i \sum_j P_{i,j}\log\mathbb{P}_{i,j}$, we find

$$\mathcal{L}(\rho_\sigma(\theta)) = -\sum_i \pi_i \sum_j P_{i,j}\log\mathbb{P}_{\sigma(i),\sigma(j)} = -\sum_i \pi_{\sigma^{-1}(i)}\sum_j P_{\sigma^{-1}(i),\sigma^{-1}(j)}\log\mathbb{P}_{i,j}. \tag{148}$$

Under the symmetry assumption on the data and the definition of the transition probability matrix $P$, $\mathcal{L}(\rho_\sigma(\theta)) = \mathcal{L}(\theta)$. Hence if $\theta(t)$ solves the gradient flow, so does $\rho_\sigma(\theta(t))$. If $\theta(t_0) = \rho_\sigma(\theta(t_0))$ for all $\sigma \in G$ which is equivalent to $\gamma_2 = \cdots = \gamma_d$ and $\beta_2 = \cdots = \beta_d$ at $t_0$, uniqueness of ODE solutions implies $\theta(t) = \rho_\sigma(\theta(t))$ for all $t \geq t_0$, which proves the symmetry is preserved. $\square$

**Proof of Theorem 4.7** We proceed with the proof of Theorem 4.7. First, we introduce some notation to show that the dynamics on a rank-one manifold will be further simplified in the case of low-frequency symmetry. Next, since we are still near the critical point described in Proposition 1, this means that we are also near the critical point for the dynamics on a rank-one manifold. Therefore, we continue using linearization methods to obtain the key conservation law results.

First, we find that the proxy attention matrix $\mathbb{A}$ has the form on $\mathcal{W}$ by direct computation,

$$
\mathbb{A} = \begin{pmatrix}
\xi_1 & \frac{1-\xi_1}{|\mathcal{V}|-1} & \cdots & \frac{1-\xi_1}{|\mathcal{V}|-1} \\
\xi_2 & \frac{1-\xi_2}{|\mathcal{V}|-1} & \cdots & \frac{1-\xi_2}{|\mathcal{V}|-1} \\
\vdots & \vdots & & \vdots \\
\xi_2 & \frac{1-\xi_2}{|\mathcal{V}|-1} & \cdots & \frac{1-\xi_2}{|\mathcal{V}|-1}
\end{pmatrix},
$$

where

$$
\xi_1 = \frac{\pi_1 \exp\left(\eta\gamma_1^2\right)}{\pi_1 \exp\left(\eta\gamma_1^2\right) + (1-\pi_1)\exp\left(\eta\gamma_1\gamma_{i\neq1}\right)}, \tag{149}
$$

$$
\xi_2 = \frac{\pi_1 \exp\left(\eta\gamma_1\gamma_{i\neq1}\right)}{\pi_1 \exp\left(\eta\gamma_1\gamma_{i\neq1}\right) + (1-\pi_1)\exp\left(\eta\gamma_{i\neq1}^2\right)}. \tag{150}
$$

Define the row-wise scalar projections

$$
m_1 := \mathbb{A}_1\gamma, \qquad m_2 := \mathbb{A}_2\gamma.
$$

Then

$$
m_1 = \gamma_{i\neq1} + \xi_1\Delta\gamma, \qquad m_2 = \gamma_{i\neq1} + \xi_2\Delta\gamma.
$$

Since $\mathbb{A}_i M = (\mathbb{A}_i\gamma)\beta^\intercal = m_i\beta^\intercal$, the model probability of predicting the first token is

$$
\hat{p}_i := \mathbb{P}_{i,1} = \frac{\exp(m_i\beta_1)}{\exp(m_i\beta_1) + (|\mathcal{V}|-1)\exp(m_i\beta_{i\neq1})} = \sigma\big(m_i\Delta\beta - \log(|\mathcal{V}|-1)\big), \qquad i \in \{1,2\}, \tag{151}
$$

where $\sigma$ is the sigmoid function. Here, we only consider the first and second probability because $\mathbb{P}_{i,1} = \mathbb{P}_{j,1}$ for $i,j \neq 1$. Moreover, there exists a key term $(P_i - \mathbb{P}_i)\beta$ in the following computation. By direct computation,

$$
\begin{aligned}
(P_i - \mathbb{P}_i)\beta &= (P_{i,1} - \mathbb{P}_{i,1})\beta_1 + \left((1 - P_{i,1} - (1 - \mathbb{P}_{i,1}))\right)\beta_{i\neq1} \\
&= (P_{i,1} - \mathbb{P}_{i,1})\Delta\beta.
\end{aligned} \tag{152}
$$

It implies that $(P_i - \mathbb{P}_i)\beta = (P_j - \mathbb{P}_j)\beta$ for $i,j \neq 1$. Let the residuals be

$$
r_i := P_{i,1} - \hat{p}_i, \qquad i \in \{1,2\}. \tag{153}
$$

For $i > 2$, we let $r_i = r_2$.

We now derive the explicit dynamics for $\gamma_1$ and $\gamma_{i\neq1}$ by expanding the two contributions in $\dot{\gamma}$ in (20).

(1). The $M$-driven term $-\frac{\partial\mathcal{L}}{\partial M}\beta$. By the definition of $\frac{\partial\mathcal{L}}{\partial M}$, we obtain

$$
-\frac{\partial\mathcal{L}}{\partial M}\beta = \sum_i \pi_i\mathbb{A}_i^\intercal(P_i - \mathbb{P}_i)\beta = \Delta\beta\left(\pi_1 r_1\mathbb{A}_1^\intercal + (1-\pi_1)r_2\mathbb{A}_2^\intercal\right).
$$

Taking the first coordinate and a generic low-token coordinate yields

$$
\dot{\gamma}_1\big|_M = \Delta\beta\left[\pi_1 r_1\xi_1 + (1-\pi_1)r_2\xi_2\right], \qquad \dot{\gamma}_{i\neq1}\big|_M = \frac{\Delta\beta}{|\mathcal{V}|-1}\left[\pi_1 r_1(1-\xi_1) + (1-\pi_1)r_2(1-\xi_2)\right]. \tag{154}
$$

(2). The $\Phi$-driven term $-\eta[(\partial\mathcal{L}/\partial\Phi) + (\partial\mathcal{L}/\partial\Phi)^\intercal]\gamma$. Using $\frac{\partial\mathcal{L}}{\partial\Phi} = -\sum_i \pi_i e_i(P_i - \mathbb{P}_i)M^\intercal \operatorname{Var}(\mathbb{A}_i)$ and $(P_i - \mathbb{P}_i)M^\intercal = \Delta\beta\, r_i\,\gamma^\intercal$, we get

$$
\frac{\partial\mathcal{L}}{\partial\Phi} = -\Delta\beta\sum_i \pi_i r_i\, e_i\,\gamma^\intercal \operatorname{Var}(\mathbb{A}_i).
$$

Thus, the $\Phi$-driven term is

$$-\eta\left[\left(\frac{\partial\mathcal{L}}{\partial\Phi}\right) + \left(\frac{\partial\mathcal{L}}{\partial\Phi}\right)^{\mathsf{T}}\right]\gamma = \eta\Delta\beta\sum_i \pi_i r_i\left(e_i\gamma^{\mathsf{T}}\operatorname{Var}(\mathbb{A}_i)\gamma + \gamma_i\operatorname{Var}(\mathbb{A}_i)\gamma\right)$$

By direct computation, we find that

$$\operatorname{Var}(\mathbb{A}_i)\gamma = \xi_i(1-\xi_i)\Delta\gamma\left(1, -\frac{1}{d-1}, \ldots, -\frac{1}{d-1}\right)^{\mathsf{T}},$$

$$\gamma^{\mathsf{T}}\operatorname{Var}(\mathbb{A}_i)\gamma = \xi_i(1-\xi_i)(\Delta\gamma)^2$$

Substituting into the equation, we get the $\Phi$- driven term:

$$\dot{\gamma}_1\big|_\Phi = \eta\Delta\beta\Big[\pi_1 r_1\,\xi_1(1-\xi_1)\big((\Delta\gamma)^2 + a\Delta\gamma\big) + (1-\pi_1)r_2\,b\,\xi_2(1-\xi_2)\Delta\gamma\Big], \tag{155}$$

$$\dot{\gamma}_{i\neq1}\big|_\Phi = \frac{\eta\Delta\beta}{|\mathcal{V}|-1}\Big[-\pi_1 r_1\,a\,\xi_1(1-\xi_1)\Delta\gamma + (1-\pi_1)r_2\,\xi_2(1-\xi_2)\big((\Delta\gamma)^2 - b\Delta\gamma\big)\Big]. \tag{156}$$

Combining (154)–(156) gives the closed ODEs for $(a, b)$ on the invariant manifold.

Plug the above equations into the dynamics of $\gamma_1, \gamma_{i\neq1}$

We now formally proceed with the proof of Theorem 4.7. We linearize the reduced system around the entry state of this phase and denote base values by superscript 0 and first-order variations by superscript 1.

*Proof.* The test for the critical point is the same as for Proposition 4.3, because the parameters are essentially located near the same minimum point.

We then linearize terms in (154)–(156) in a fixed order.

1. Linearization about $M$-driven term. We take the expansion up to the first order about $\xi_i$ and $\hat{p}_i$ and then substitute then into the expression of $M$-driven term.

   (1). Linearization of the proxy attention weights $\xi_1, \xi_2$. Take $\xi_1$ as an example,

   $$\xi_1 = \frac{1}{1 + \frac{1-\pi_1}{\pi_1}\exp\left(-\eta\gamma_1\Delta\gamma\right)} = \pi_1 + \pi_1(1-\pi_1)\eta^1\gamma_1^0\Delta\gamma^0 + \mathcal{O}(\|\theta\|^2)$$

   in which we use the fact that parameters locate near $\eta = 0$. Thus,

   $$\xi_1^1 = \pi_1(1-\pi_1)\eta^1\gamma_1^0\Delta\gamma^0, \qquad \xi_2^1 = \pi_1(1-\pi_1)\eta^1\gamma_{i\neq1}^0\Delta\gamma^0.$$

   (2). Linearization of the prediction probabilities $\hat{p}_i$ and residuals $r_i$. Recall $\hat{p}_i = \sigma(m_i\Delta\beta - \log(|\mathcal{V}| - 1))$ with $m_i = b + \xi_i\Delta\gamma$. Expanding $\hat{p}_i$ to first order gives (writing $m_i^0$ for the base value)

   $$\hat{p}_1^1 = \pi_1(1-\pi_1)\Big((b^1 + \pi_1\Delta\gamma^1)\Delta\beta^0 + \pi_1(1-\pi_1)\eta\,a^0\Delta\gamma^0\Delta\beta^0 + m_1^0\Delta\beta^1\Big),$$

   $$\hat{p}_2^1 = \pi_1(1-\pi_1)\Big((b^1 + \pi_1\Delta\gamma^1)\Delta\beta^0 + \pi_1(1-\pi_1)\eta\,b^0\Delta\gamma^0\Delta\beta^0 + m_2^0\Delta\beta^1\Big).$$

   Since $r_i = P_{i,1} - \hat{p}_i$, we have $r_i^1 = -\hat{p}_i^1$.

2. Linearization about $\Phi$-driven term. Using the fact that parameters locate near $\eta = 0$, the linearization of Eq (154)–(156) corresponds to the right-hand side except that eta takes a value at the initial point.

Substituting the above expansions into (154)–(156), and keeping only first-order terms, yields

$$\dot{\gamma}_1^1 = \Delta\beta^0\big(\pi_1^2(-\hat{p}_1^1) + \pi_1(1-\pi_1)(-\hat{p}_2^1)\big) + 3\lambda\,\eta^1\pi_1^2(1-\pi_1)^2\,\Delta\beta^0(\Delta\gamma^0)^2,$$

$$\dot{\gamma}_{i\neq1}^1 = \frac{\Delta\beta^0}{|\mathcal{V}|-1}\big(\pi_1(1-\pi_1)(-\hat{p}_1^1) + (1-\pi_1)^2(-\hat{p}_2^1)\big) - \frac{1}{|\mathcal{V}|-1}3\lambda\eta^1\pi_1^2(1-\pi_1)^2\,\Delta\beta^0(\Delta\gamma^0)^2.$$

Taking the linear combination $(1 - \pi_1)\dot{\gamma}_1^1 - (|\mathcal{V}| - 1)\pi_1\dot{\gamma}_{i\neq 1}^1$ cancels the $(-\hat{p}_i^1)$ terms and yields

$$(1 - \pi_1)\dot{\gamma}_1^1 - (|\mathcal{V}| - 1)\pi_1\dot{\gamma}_{i\neq 1}^1 = 3\lambda\pi_1^2(1 - \pi_1)^2\Delta\beta^0(\Delta\gamma^0)^2\eta^1. \tag{157}$$

Considering the linearized dynamics about $\eta$, there exists $c > 0$ such that

$$\dot{\eta}^1 = c\eta^1,$$

which indicating that $\eta^1$ admits a solution as

$$\eta^1(t) = \eta^1(t_0)\exp c(t - t_0). \tag{158}$$

Substituting the above equation into Eq. (157) and integrating both sides of the equation, we get

$$(1 - \pi_1)\gamma_1^1(t) - (|\mathcal{V}| - 1)\pi_1\gamma_{i\neq 1}^1(t) = c'\left(\exp(c(t - t_0)) - 1\right) \tag{159}$$

Here, we use the fact that

$$(1 - \pi_1)\gamma_1(t_0) - (d - 1)\pi_1\gamma_{i\neq 1}(t_0) = 0.$$

$\square$

# C. Theoretical details in Sec. 4.4

This appendix provides detailed proofs for Section 4.4. We focus on the minimal vocabulary size $d = 3$ to exhibit the separation between secondary high frequency and secondary low frequency. Throughout, we use the rank-one parametrization on the invariant manifold (cf. Proposition 4.6)

$$M = \gamma\beta^\mathsf{T}, \qquad \Phi = \eta\,\gamma\gamma^\mathsf{T}, \qquad \theta = (\gamma, \beta) \in \mathbb{R}^3 \times \mathbb{R}^3.$$

Here, we do not need to consider $\eta$, because calculations show that its derivatives up to the second order are zero, so it will not affect our analysis.

## C.1. A degenerate critical point on the rank-one manifold

We first formalize the "bad" critical point on the rank-one manifold under symmetric frequencies. This critical point is degenerate in the sense that the key driving terms $\partial\mathcal{L}/\partial M$ and $\partial\mathcal{L}/\partial\Phi$ vanish, hence linearization on the manifold cannot explain the escape to new embedding directions. The following is the proof of Proposition 4.8.

*Proof.* We construct a critical point on the rank-one manifold and show it is a local minimum for the linearized dynamics.

The critical point is constructed as follows. Take $\gamma_1\gamma_{i\neq 1} < 0$ as shown in Theorem 4.7. When $\eta$ is sufficiently large, the attention proxy satisfies $\mathbb{A}_1 \approx e_1$ and $\mathbb{A}_{i\neq 1} \approx \hat{e}_1 := (0, \frac{1}{2}, \frac{1}{2})$. Choose $\beta_1 > \beta_{i\neq 1}$ so that $\mathrm{softmax}(k\beta^\mathsf{T}) \to e_1$ as $k \to +\infty$ and $\mathrm{softmax}(k\beta^\mathsf{T}) \to \hat{e}_1$ as $k \to -\infty$. Thus we may choose $\gamma_1 > 0$ and $\gamma_{i\neq 1} < 0$ so that

$$\mathbb{P}_1 = P_1, \qquad \mathbb{P}_{i\neq 1} = \tfrac{1}{2}(P_2 + P_3).$$

By direct computation and symmetry of the data, we have $\frac{\partial\mathcal{L}}{\partial M} = 0$ and $\frac{\partial\mathcal{L}}{\partial\Phi} = 0$ at this point (refer to Lemma C.4). Substituting this fact into Eq. (4), it implies that our construction gives a critical point.

To verify local minimality for the linearized dynamics, we linearize the dynamics in Eq. (4). We compute $\mathrm{d}\left(\frac{\partial\mathcal{L}}{\partial M}\right)$ and $\mathrm{d}\left(\frac{\partial\mathcal{L}}{\partial\Phi}\right)$. At the constructed symmetric point, Lemma C.5 implies $\sum_i \pi_i\,\mathrm{d}\mathbb{A}_i^\mathsf{T}(P_i - \mathbb{P}_i) = 0$ and hence

$$\mathrm{d}\frac{\partial\mathcal{L}}{\partial M} = \sum_i \pi_i\mathbb{A}_i^\mathsf{T}\,\mathrm{d}\mathbb{P}_i \neq 0.$$

Moreover, Lemma C.5 shows that $\mathrm{d}\mathbb{P}_i = \mathbb{A}_i\,\mathrm{d}M\,\mathrm{Var}(\mathbb{P}_i)$. Since $\mathbb{A}_2 = \mathbb{A}_3$, it implies that $\mathrm{d}\mathbb{P}_2 = \mathrm{d}\mathbb{P}_3$. In addition, Lemma C.5 gives $\mathrm{d}(\partial\mathcal{L}/\partial\Phi) = 0$.

As a result, the linearized dynamics on $(\gamma, \beta, \eta)$ reduces to

$$\frac{\mathrm{d}\Delta\gamma}{\mathrm{d}t} = -\mathrm{d}\left(\frac{\partial\mathcal{L}}{\partial M}\right)\beta, \qquad \frac{\mathrm{d}\Delta\beta}{\mathrm{d}t} = -\mathrm{d}\left(\frac{\partial\mathcal{L}}{\partial M}\right)^{\mathsf{T}}\gamma, \qquad \frac{\mathrm{d}\Delta\eta}{\mathrm{d}t} = 0, \tag{160}$$

and the Jacobian $J_0 = -\nabla_\theta^2\mathcal{L}$ admits the explicit block form in Lemma C.6. In particular, $J_0$ is negative semidefinite with a nontrivial kernel. Hence, the critical point we constructed is a neutrally stable equilibrium for the linearized dynamics, which motivates the Lyapunov–Schmidt reduction in the main text. $\qquad\square$

### C.2. Breaking the degeneracy: frequency perturbation and Lyapunov–Schmidt reduction

To eliminate the degeneracy, we perturb the frequencies between the two low-frequency states:

$$\pi^{\mathsf{T}} = \left(c, \frac{1-c}{2}, \frac{1-c}{2}\right) \quad \Rightarrow \quad \tilde{\pi}^{\mathsf{T}} = \left(c, \frac{1-c}{2}+\delta, \frac{1-c}{2}-\delta\right). \tag{161}$$

The dynamics becomes

$$\dot{\theta} = -\nabla_\theta\mathcal{L}(\theta, \delta).$$

We study the perturbed critical point by solving

$$-\nabla_\theta\mathcal{L}(\theta, \delta) = 0 \tag{162}$$

near the degenerate minimum, which we shift to $\theta = 0$ for convenience.

We use the formal expansion (at $\theta = 0$):

$$-\nabla_\theta\mathcal{L}(\theta, \delta) = J_0\theta + \delta f_1 + \frac{1}{2}B(\theta, \theta) + \delta J_1\theta + \frac{1}{2}\delta^2 f_2 + \text{h.o.t.,} \tag{163}$$

where

$$J_0 = -\nabla_\theta^2\mathcal{L}, \quad f_1 = \partial_\delta(-\nabla_\theta\mathcal{L}), \quad B(\cdot, \cdot) = -\nabla_\theta^3\mathcal{L}, \quad J_1 = \partial_\delta J_0, \quad f_2 = \partial_\delta^2(-\nabla_\theta\mathcal{L}).$$

Since $J_0$ is singular, we apply Lyapunov–Schmidt reduction.

#### C.2.1. KERNEL/RANGE DECOMPOSITION OF $J_0$

**Proposition C.1** (Kernel and range bases). *Assume $\|\gamma\| = \|\beta\|$ and $\beta^{\mathsf{T}}\mathbf{1} = 0$ at the symmetric degenerate minimum. Then* $\dim\ker(J_0) = 3$ *and one convenient orthonormal basis is*

$$k_1 = \frac{1}{\sqrt{2}}\left((0, 1, -1), (0, 0, 0)\right),$$

$$k_2 = \frac{1}{\sqrt{3}}\left((0, 0, 0), (1, 1, 1)\right), \tag{164}$$

$$k_3 = \frac{1}{\sqrt{\|\gamma\|^2 + \|\beta\|^2}}\left(-\gamma, \beta\right).$$

*An orthonormal basis for* $\mathrm{Range}(J_0)$ *can be taken as*

$$q_1 = \frac{1}{\sqrt{4\gamma_2^2 + 2\gamma_1^2}}\left((-2\gamma_2, \gamma_1, \gamma_1), (0, 0, 0)\right),$$

$$q_2 = \frac{1}{\sqrt{2}}\left((0, 0, 0), (0, 1, -1)\right), \tag{165}$$

$$q_3 = \frac{1}{\sqrt{\|\gamma\|^2 + \|\beta\|^2}}\left(\gamma, \beta\right).$$

Let $Q_K = (k_1, k_2, k_3)$ and $Q_R = (q_1, q_2, q_3)$, and denote projections $P_K = Q_KQ_K^{\mathsf{T}}$, $P_R = Q_RQ_R^{\mathsf{T}}$. Write $\theta = Q_Kx + Q_Ry$.

C.2.2. SOLVING THE RANGE EQUATION

Recall the Lyapunov–Schmidt decomposition $\theta = Q_K x + Q_R y$ and define the range equation

$$F_R(x, y, \delta) := -Q_R^\mathsf{T} \nabla_\theta \mathcal{L}(Q_K x + Q_R y, \delta) = 0. \tag{166}$$

**Proposition C.2** (Range solution and first-order expansion). *Given a perturbation of the data parameterized by $\delta$, the range equation* (166) *admits a unique solution* $y = \zeta(x, \delta)$ *in a neighborhood of* $(x, \delta) = (0, 0)$. *Moreover, it satisfies the expansion*

$$\zeta(x, \delta) = \delta \begin{pmatrix} 0 \\ \sqrt{2} \dfrac{\lambda \gamma_{i \neq 1} + (1 - \lambda)(\pi_1 \gamma_1 + (1 - \pi_1)\gamma_{i \neq 1})}{\pi_1 \gamma_1^2 \mathbb{P}_{1,2} + (1 - \pi_1)\gamma_{i \neq 1}^2 \mathbb{P}_{i \neq 1, 2}} \\ 0 \end{pmatrix} + \mathcal{O}(\delta^2 + \|x\|^2), \tag{167}$$

*where the denominator*

$$c_1 := \pi_1 \gamma_1^2 \mathbb{P}_{1,2} + (1 - \pi_1)\gamma_{i \neq 1}^2 \mathbb{P}_{i \neq 1, 2}$$

*is strictly positive under our standing assumptions (in particular* $\gamma_1, \gamma_{i \neq 1} \neq 0$ *and* $\mathbb{P}_{1,2}, \mathbb{P}_{i \neq 1, 2} > 0$).

*Proof.* We expand $F_R$ around $(x, y, \delta) = (0, 0, 0)$. Writing $\theta = Q_K x + Q_R y$ and using

$$-\nabla_\theta \mathcal{L}(\theta, \delta) = J_0 \theta + \delta f_1 + \mathcal{O}(\|\theta\|^2 + \delta^2),$$

we obtain

$$F_R(x, y, \delta) = Q_R^\mathsf{T}(J_0 Q_R y + \delta f_1) + \mathcal{O}(\|\theta\|^2 + \delta^2) = \Lambda_R y + \delta Q_R^\mathsf{T} f_1 + \mathcal{O}(\|x\|^2 + \|y\|^2 + \delta^2), \tag{168}$$

where $\Lambda_R := Q_R^\mathsf{T} J_0 Q_R$.

By Lemma C.7 we have an explicit expression for $Q_R^\mathsf{T} f_1$, and by Lemma C.8 the matrix $\Lambda_R$ is invertible on the range coordinates; in particular, its $(2, 2)$-entry equals $-c_1 < 0$ and hence $(\Lambda_R^{-1})_{22} = -1/c_1$.

Therefore, $\partial_y F_R(0, 0, 0) = \Lambda_R$ is invertible, and the implicit function theorem yields a unique smooth function $y = \zeta(x, \delta)$ solving $F_R(x, \zeta(x, \delta), \delta) = 0$ locally, with

$$\zeta(x, \delta) = -\Lambda_R^{-1} \delta Q_R^\mathsf{T} f_1 + \mathcal{O}(\delta^2 + \|x\|^2). \tag{169}$$

Since $Q_R^\mathsf{T} f_1$ has only a nonzero second component (Lemma C.7), and $(\Lambda_R^{-1})_{22} = -1/c_1$ (Lemma C.8), the second coordinate of $\zeta$ equals

$$\zeta_2(x, \delta) = -\left(-\frac{1}{c_1}\right) \delta \cdot \sqrt{2}\big(\lambda \gamma_{i \neq 1} + (1 - \lambda)(\pi_1 \gamma_1 + (1 - \pi_1)\gamma_{i \neq 1})\big) + \mathcal{O}(\delta^2 + \|x\|^2),$$

which is exactly (167). This completes the proof. $\square$

C.2.3. REDUCED KERNEL EQUATION AND APPROXIMATE CRITICAL POINT

Plugging $y = \zeta(x, \delta)$ into the kernel equation gives

$$-Q_K^\mathsf{T} \nabla \mathcal{L}(Q_K x + Q_R \zeta(x, \delta), \delta) = 0.$$

Because $J_0 Q_K = 0$ and $Q_K^\mathsf{T} f_1 = 0$, the leading contributions are second order:

$$Q_K^\mathsf{T}\left(\frac{1}{2} B(\theta, \theta) + \delta J_1 \theta + \frac{1}{2}\delta^2 f_2\right) + \text{h.o.t.} = 0, \qquad \theta = Q_K x + Q_R \zeta(x, \delta).$$

**Theorem C.3** (Existence of an approximate critical point and its two-scale stability). *Let $\zeta(x, \delta)$ be given by Proposition C.2. Then $x = 0$ is an approximate solution of the reduced kernel equation up to second order, i.e.*

$$\big\|\nabla_\theta \mathcal{L}(Q_R \zeta(0, \delta), \delta)\big\| = \mathcal{O}(\delta^3).$$

*Moreover, the linear stability splits into two scales:*

1. **Slow manifold directions (within the rank-one manifold):** *any positive eigenvalues created from the kernel directions are at most $\mathcal{O}(\delta^2)$.*

2. **Fast transverse directions (escaping the manifold):** *Under condition in Lem. C.17, there exists a transverse positive eigenvalue of order $\Theta(\delta)$.*

*Proof.* The estimate $\|\nabla\mathcal{L}\| = \mathcal{O}(\delta^3)$ follows by inserting $\theta = Q_R\zeta(0, \delta)$ into the kernel expansion and using the explicit expressions:

(1). $\frac{1}{2}Q_K^\intercal B(q_2y_2, q_2y_2)$ (From Lem. C.13):

$$\frac{1}{2}Q_K^\intercal B(q_2y_2, q_2y_2) = \frac{1}{\sqrt{\|\gamma\|^2 + \|\beta\|^2}} \begin{pmatrix} 0 \\ 0 \\ \pi_1\gamma_1^2\mathbb{P}_{1,2} + (1 - \pi_1)\gamma_2^2\mathbb{P}_{i\neq1,2} \end{pmatrix} y_2^2.$$

(2). $\delta Q_K^\intercal J_1(q_2y_2)$ (From Lem. C.10):

$$\delta Q_K^\intercal J_1(q_2y_2) = -\delta\frac{\sqrt{2}}{\sqrt{\|\gamma\|^2 + \|\beta\|^2}} \begin{pmatrix} 0 \\ 0 \\ \pi_1\gamma_1(1 - \lambda) + \gamma_{i\neq1}(\lambda + (1 - \pi_1)(1 - \lambda)) \end{pmatrix} y_2.$$

(3). $\delta^2 f_2$ vanishes (From Lem. C.14).

Substitute $y_2 = \sqrt{2}\frac{\lambda\gamma_{i\neq1} + (1-\lambda)(\pi_1\gamma_1 + (1-\pi_1)\gamma_{i\neq1})}{\pi_1\gamma_1^2\mathbb{P}_{1,2} + (1-\pi_1)\gamma_{i\neq1}^2\mathbb{P}_{i\neq1,2}}\delta$. We found that the second-order terms cancel each other out automatically.

For stability, we write the perturbed Hessian at the approximate critical point as

$$\nabla_\theta^2\mathcal{L}(\theta, \delta) = J_0 + \delta H_1 + \mathcal{O}(\delta^2), \qquad H_1 := -\left(\begin{pmatrix} 0 & B \\ B^\intercal & 0 \end{pmatrix} + \begin{pmatrix} 0 & 0 \\ 0 & C \end{pmatrix}\right) + J_1,$$

with $(B, C)$ computed from the second-order kernel reduction (see Lem. C.15).

We take the basis as $Q = (Q_K, Q_R)$:

$$Q^\intercal JQ = \begin{pmatrix} 0 & 0 \\ 0 & \Lambda \end{pmatrix} + \delta\begin{pmatrix} G & E \\ E^\intercal & F \end{pmatrix} + \mathcal{O}(\delta^2) \tag{170}$$

where $\Lambda = Q_R^\intercal J_0 Q_R$, $G = Q_K^\intercal H_1 Q_K$, $E = Q_K^\intercal H_1 Q_R$, and $F = Q_R^\intercal H_1 Q_R$. Let $\lambda$ be a small eigenvalue and the corresponding eigenvector is $(x, y)$, the equation is

$$\begin{cases} \delta Gx + \delta Ey + \mathcal{O}(\delta^2) = \lambda x \\ \Lambda y + \delta E^\intercal x + \mathcal{O}(\delta)y = \lambda y \end{cases} \tag{171}$$

Since the new positive eigenvalue is small, we can solve $y$ as

$$y = -(\Lambda - \lambda I)^{-1}\delta E^\intercal x + \mathcal{O}(\delta^2) = -\delta\Lambda^{-1}E^\intercal x + \mathcal{O}(\delta^2) \tag{172}$$

Substitute this expression into the the first equation, we get

$$(\delta G - \delta^2 E\Lambda^{-1}E^\intercal)x = \lambda x + \mathcal{O}(\delta^3) \tag{173}$$

Hence $\lambda = \mathcal{O}(\delta^2)$ provided $G = Q_K^\intercal H_1 Q_K = 0$. This vanishing is proved in Lemma C.16.

Finally, we compute the eigenvalue of the normal directions. Recall the linearization of the whole dynamics is

$$
\begin{cases}
\dfrac{\mathrm{d}\Delta W_0}{\mathrm{d}t} = \Delta\left(-\dfrac{\partial\mathcal{L}}{\partial M}W_1^{\mathsf{T}} - \dfrac{\partial\mathcal{L}}{\partial\Phi}W_0 W_K W_Q^{\mathsf{T}} - \left(\dfrac{\partial\mathcal{L}}{\partial\Phi}\right)^{\mathsf{T}} W_0 W_Q W_K^{\mathsf{T}}\right) \\[3mm]
\dfrac{\mathrm{d}\Delta W_1}{\mathrm{d}t} = \Delta\left(-W_0^{\mathsf{T}}\dfrac{\partial\mathcal{L}}{\partial M}\right) \\[3mm]
\dfrac{\mathrm{d}\Delta W_Q}{\mathrm{d}t} = \Delta\left(-W_0^{\mathsf{T}}\dfrac{\partial\mathcal{L}}{\partial\Phi}W_0 W_K\right) \\[3mm]
\dfrac{\mathrm{d}\Delta W_K}{\mathrm{d}t} = \Delta\left(-W_0^{\mathsf{T}}\left(\dfrac{\partial\mathcal{L}}{\partial\Phi}\right)^{\mathsf{T}} W_0 W_Q\right)
\end{cases}
\tag{174}
$$

We find that

$$
\begin{cases}
\dfrac{\mathrm{d}\Delta W_0\alpha_{1,\perp}}{\mathrm{d}t} = -\dfrac{\partial\mathcal{L}}{\partial M}\Delta W_1^{\mathsf{T}}\alpha_{1,\perp} - \dfrac{\partial\mathcal{L}}{\partial\Phi}W_0 W_K \Delta W_Q^{\mathsf{T}}\alpha_{1,\perp} - \left(\dfrac{\partial\mathcal{L}}{\partial\Phi}\right)^{\mathsf{T}} W_0 W_Q \Delta W_K^{\mathsf{T}}\alpha_{1,\perp} \\[3mm]
\dfrac{\mathrm{d}\alpha_{1,\perp}^{\mathsf{T}}\Delta W_1}{\mathrm{d}t} = -\alpha_{1,\perp}^{\mathsf{T}}\Delta W_0^{\mathsf{T}}\dfrac{\partial\mathcal{L}}{\partial M} \\[3mm]
\dfrac{\mathrm{d}\alpha_{1,\perp}^{\mathsf{T}}\Delta W_Q}{\mathrm{d}t} = -\Delta W_0^{\mathsf{T}}\dfrac{\partial\mathcal{L}}{\partial\Phi}W_0 W_K \\[3mm]
\dfrac{\mathrm{d}\alpha_{1,\perp}^{\mathsf{T}}\Delta W_K}{\mathrm{d}t} = -\Delta W_0^{\mathsf{T}}\left(\dfrac{\partial\mathcal{L}}{\partial\Phi}\right)^{\mathsf{T}} W_0 W_Q
\end{cases}
\tag{175}
$$

Since $\partial_\delta\frac{\partial\mathcal{L}}{\partial\Phi} = 0$ and $\mathrm{d}\frac{\partial\mathcal{L}}{\partial\Phi} = 0$, we find that $\frac{\partial\mathcal{L}}{\partial\Phi} = \mathcal{O}(\delta^2)$. So the main term is

$$
\begin{cases}
\dfrac{\mathrm{d}\Delta W_0\alpha_{1,\perp}}{\mathrm{d}t} = -\dfrac{\partial\mathcal{L}}{\partial M}\Delta W_1^{\mathsf{T}}\alpha_{1,\perp} \\[3mm]
\dfrac{\mathrm{d}\alpha_{1,\perp}^{\mathsf{T}}\Delta W_1}{\mathrm{d}t} = -\alpha_{1,\perp}^{\mathsf{T}}\Delta W_0^{\mathsf{T}}\dfrac{\partial\mathcal{L}}{\partial M}
\end{cases}
\tag{176}
$$

From Lemma C.17, under some mild condition, We find that

$$
\frac{\partial\mathcal{L}}{\partial M} = \Theta(\delta)
\tag{177}
$$

Thus, there exists positive eigenvalue at least order $\Theta(\delta)$. $\qquad\square$

## C.3. Derivative toolbox

This section collects all derivative computations referenced in the proofs above.

### C.3.1. VANISHING OF GRADIENTS

**Lemma C.4** ($\partial\mathcal{L}/\partial M = 0$ and $\partial\mathcal{L}/\partial\Phi = 0$ at the constructed point). *At the symmetric degenerate minimum in Proposition 4.8, we have*

$$
\frac{\partial\mathcal{L}}{\partial M} = 0, \qquad \frac{\partial\mathcal{L}}{\partial\Phi} = 0.
$$

*Proof.* By direct computation, we get $\mathbb{P}_1 = P_1$ and $\mathbb{P}_2 = \mathbb{P}_3 = \frac{1}{2}(P_2 + P_3)$. Thus, the terms in the gradients cancel after summing with $\pi_2 = \pi_3$. $\qquad\square$

### C.3.2. FIRST-ORDER VARIATIONS

**Lemma C.5** (First-order variations with respect to parameters). *At the critical point in Proposition 4.8, the first-order variations have the following form:*

1. *The variation of the attention proxy satisfies* $\mathrm{d}\mathbb{A}_1 = 0$ *and* $\mathrm{d}\mathbb{A}_{i\neq 1} = \eta\gamma_{i\neq 1}\left(0, \frac{1}{4}(\mathrm{d}\gamma_2 - \mathrm{d}\gamma_3), -\frac{1}{4}(\mathrm{d}\gamma_2 - \mathrm{d}\gamma_3)\right)$ *for* $i \neq 1$.

2. *The variation of the output probability satisfies* $\mathrm{d}\mathbb{P}_i = \mathbb{A}_i \mathrm{d}M \, \mathrm{Var}(\mathbb{P}_i)$.

3. *The variation of* $\frac{\partial \mathcal{L}}{\partial M}$ *and* $\frac{\partial \mathcal{L}}{\partial \Phi}$ *admit the following expression:*

$$\mathrm{d}\frac{\partial \mathcal{L}}{\partial M} = \sum_i \pi_i \mathbb{A}_i^\mathsf{T} \mathrm{d}\mathbb{P}_i, \qquad \mathrm{d}\frac{\partial \mathcal{L}}{\partial \Phi} = 0.$$

*Proof.* We calculate the first-order variation in sequence.

1. At $\mathbb{A}_1 = e_1$, we have $\mathrm{diag}(e_1) - e_1 e_1^\mathsf{T} = 0$, hence $\mathrm{d}\mathbb{A}_1 = 0$. For $i \neq 1$, using $\mathbb{A}_i = \hat{e}_1$ and $\Phi = \eta \gamma \gamma^\mathsf{T}$,

$$e_i^\mathsf{T} \mathrm{d}\Phi = e_i^\mathsf{T} \mathrm{d}(\eta \gamma \gamma^\mathsf{T}) = \mathrm{d}(\eta \gamma_i \gamma^\mathsf{T}).$$

Since $\mathrm{Var}(\mathbb{A}_{i \neq 1}) = \mathrm{diag}(\hat{e}_1) - \hat{e}_1 \hat{e}_1^\mathsf{T}$ equals to

$$\begin{pmatrix} 0 & 0 & 0 \\ 0 & \frac{1}{4} & -\frac{1}{4} \\ 0 & -\frac{1}{4} & \frac{1}{4} \end{pmatrix},$$

we obtain the displayed vector form.

2. By definition,

$$\mathrm{d}\mathbb{P}_i = \mathrm{d}(\mathbb{A}_i M) \, \mathrm{Var}(\mathbb{P}_i) = (\mathrm{d}\mathbb{A}_i \, M + \mathbb{A}_i \, \mathrm{d}M) \, \mathrm{Var}(\mathbb{P}_i).$$

Under $M = \gamma \beta^\mathsf{T}$,

$$\mathrm{d}\mathbb{A}_i \, M = \mathrm{d}\mathbb{A}_i \, \gamma \beta^\mathsf{T} = (\mathrm{d}\mathbb{A}_i \, \gamma) \beta^\mathsf{T}.$$

For $i = 1$, $\mathrm{d}\mathbb{A}_1 = 0$, hence $\mathrm{d}\mathbb{A}_1 M = 0$. For $i \neq 1$, $\mathrm{d}\mathbb{A}_{i \neq 1} = \eta \gamma_{i \neq 1} \left(0, \frac{1}{4}(\mathrm{d}\gamma_2 - \mathrm{d}\gamma_3), -\frac{1}{4}(\mathrm{d}\gamma_2 - \mathrm{d}\gamma_3)\right)$. Thus,

$$\mathrm{d}\mathbb{A}_{i \neq 1} \, \gamma = \eta \gamma_{i \neq 1}(0, \tfrac{1}{4}(\mathrm{d}\gamma_2 - \mathrm{d}\gamma_3), -\tfrac{1}{4}(\mathrm{d}\gamma_2 - \mathrm{d}\gamma_3)) \cdot (\gamma_1, \gamma_2, \gamma_3) = \tfrac{1}{4}\eta \gamma_{i \neq 1}(\mathrm{d}\gamma_2 - \mathrm{d}\gamma_3)(\gamma_2 - \gamma_3) = 0,$$

since $\gamma_2 = \gamma_3$ at the symmetric point. Hence $\mathrm{d}\mathbb{A}_{i \neq 1} M = 0$. Therefore $\mathrm{d}\mathbb{P}_i = \mathbb{A}_i \, \mathrm{d}M \, \mathrm{Var}(\mathbb{P}_i)$.

Because $\mathbb{A}_2 = \mathbb{A}_3 = \hat{e}_1$ and $\mathrm{Var}(\mathbb{P}_2) = \mathrm{Var}(\mathbb{P}_3)$ under symmetry, we also have $\mathrm{d}\mathbb{P}_2 = \mathrm{d}\mathbb{P}_3$.

3. By definition of $\frac{\partial \mathcal{L}}{\partial M}$ and the chain rule,

$$\mathrm{d}\frac{\partial \mathcal{L}}{\partial M} = -\sum_i \pi_i \mathrm{d}\mathbb{A}_i^\mathsf{T}(P_i - \mathbb{P}_i) - \sum_i \pi_i \mathbb{A}_i^\mathsf{T} \mathrm{d}(-\mathbb{P}_i)$$

At the symmetric point, $P_1 - \mathbb{P}_1 = 0$ and $\sum_{i \neq 1}(P_i - \mathbb{P}_i) = 0$, while $\mathrm{d}\mathbb{A}_2 = \mathrm{d}\mathbb{A}_3$ and $\pi_2 = \pi_3$. Therefore the $i = 2, 3$ contributions cancel, giving $\sum_i \pi_i \, \mathrm{d}\mathbb{A}_i^\mathsf{T}(P_i - \mathbb{P}_i) = 0$. It yields the claimed form.

By the definition of $\frac{\partial \mathcal{L}}{\partial \Phi}$ and the chain rule,

$$\mathrm{d}\frac{\partial \mathcal{L}}{\partial \Phi} = -\mathrm{d}\Big(\sum_i \pi_i \, e_i \, (P_i - \mathbb{P}_i) \, M^\mathsf{T} \, \mathrm{Var}(\mathbb{A}_i)\Big)$$

$$= -\sum_i \pi_i e_i \mathrm{d}(-\mathbb{P}_i) M^\mathsf{T} \, \mathrm{Var}(\mathbb{A}_i) - \sum_i \pi_i e_i (P_i - \mathbb{P}_i) \mathrm{d}M^\mathsf{T} \, \mathrm{Var}(\mathbb{A}_i) - \sum_i \pi_i e_i (P_i - \mathbb{P}_i) M^\mathsf{T} \mathrm{d} \, \mathrm{Var}(\mathbb{A}_i).$$

Using the expression of $\mathrm{Var}(\mathbb{A}_i)$, we get $\gamma^\mathsf{T} \, \mathrm{Var}(\mathbb{A}_i) = 0$ since $\gamma_2 = \gamma_3$, which implies that the first term vanishes. Similarly, using the chain rule, we get $\mathrm{d}M^\mathsf{T} = \mathrm{d}\beta \gamma^\mathsf{T} + \beta \mathrm{d}\gamma^\mathsf{T}$. Combined with $(P_i - \mathbb{P}_i)\beta = 0$ and $\gamma^\mathsf{T} \, \mathrm{Var}(\mathbb{A}_i) = 0$, the second term vanishes. The third term vanishes due to the same reason.

$\square$

### C.3.3. HESSIAN MATRIX $J_0$

**Lemma C.6** (Computation of $J_0$ on the rank-one manifold). *At the critical point in Proposition 4.8, the linearization restricted to the rank-one manifold yields the Hessian $J_0 = -\nabla_\theta^2 \mathcal{L}(\theta, 0)$ in the matrix form:*

$$J_0 = - \begin{pmatrix} c_1 & 0 & 0 & v_1 \\ 0 & c_2 & c_2 & v_2 \\ 0 & c_2 & c_2 & v_2 \\ v_1^\mathsf{T} & v_2^\mathsf{T} & v_2^\mathsf{T} & C \end{pmatrix} \tag{178}$$

*where $c_1 := \pi_1 \|\beta\|_{\mathrm{Var}(\mathbb{P}_1)}^2$, $c_2 := \frac{1}{4}(1-\pi_1)\|\beta\|_{\mathrm{Var}(\mathbb{P}_{i\neq 1})}^2$, $v_1 := \pi_1 \gamma_1 \beta^\mathsf{T} \mathrm{Var}(\mathbb{P}_1)$, $v_2 := \frac{1}{2}(1-\pi_1)\gamma_{i\neq 1}\beta^\mathsf{T} \mathrm{Var}(\mathbb{P}_{i\neq 1})$, and $C := \pi_1 \gamma_1^2 \mathrm{Var}(\mathbb{P}_1) + (1-\pi_1)\gamma_{i\neq 1}^2 \mathrm{Var}(\mathbb{P}_{i\neq 1})$. Moreover, $J_0$ is negative semidefinite.*

*Proof.* As shown in Eq. (160), the linearized dynamics on the rank-one manifold can be written as

$$\frac{\mathrm{d}\Delta\gamma}{\mathrm{d}t} = -\left(\mathrm{d}\frac{\partial\mathcal{L}}{\partial M}\right)\beta,$$
$$\frac{\mathrm{d}\Delta\beta}{\mathrm{d}t} = -\left(\mathrm{d}\frac{\partial\mathcal{L}}{\partial M}\right)^\mathsf{T}\gamma,$$
$$\frac{\mathrm{d}\Delta\eta}{\mathrm{d}t} = 0,$$

where we used $\mathrm{d}(\partial\mathcal{L}/\partial\Phi) = 0$ at the symmetric point. We calculate the Jacobian corresponding to $\frac{\mathrm{d}\Delta\gamma}{\mathrm{d}t}$ and $\frac{\mathrm{d}\Delta\beta}{\mathrm{d}t}$ respectively.

1. The $\Delta\gamma$ equation. Using $\mathrm{d}\frac{\partial\mathcal{L}}{\partial M} = \pi_1 \mathbb{A}_1^\mathsf{T} \mathrm{d}\mathbb{P}_1 + (1-\pi_1)\mathbb{A}_{i\neq 1}^\mathsf{T} \mathrm{d}\mathbb{P}_{i\neq 1}$ and $\mathbb{A}_1 = e_1^\mathsf{T}$, $\mathbb{A}_{i\neq 1} = \hat{e}_1^\mathsf{T} = (0, \frac{1}{2}, \frac{1}{2})$, we obtain

$$\begin{aligned} \frac{\mathrm{d}\Delta\gamma}{\mathrm{d}t} &= -\pi_1 \mathbb{A}_1^\mathsf{T} \mathrm{d}\mathbb{P}_1 \beta - (1-\pi_1)\mathbb{A}_{i\neq 1}^\mathsf{T} \mathrm{d}\mathbb{P}_{i\neq 1}\beta \\ &= -\pi_1 \mathbb{A}_1^\mathsf{T}\left(\mathrm{d}\gamma_1 \beta^\mathsf{T} \mathrm{Var}(\mathbb{P}_1)\beta + \gamma_1 \mathrm{d}\beta^\mathsf{T} \mathrm{Var}(\mathbb{P}_1)\beta\right) \\ &\quad - (1-\pi_1)\mathbb{A}_{i\neq 1}^\mathsf{T}\left(\tfrac{1}{2}(\mathrm{d}\gamma_2 + \mathrm{d}\gamma_3)\beta^\mathsf{T} \mathrm{Var}(\mathbb{P}_{i\neq 1})\beta + \gamma_{i\neq 1}\mathrm{d}\beta^\mathsf{T} \mathrm{Var}(\mathbb{P}_{i\neq 1})\beta\right), \end{aligned} \tag{179}$$

where we used the rank-one identity $\mathrm{d}M = \mathrm{d}(\gamma\beta^\mathsf{T}) = (\mathrm{d}\gamma)\beta^\mathsf{T} + \gamma(\mathrm{d}\beta)^\mathsf{T}$ and $\mathrm{d}\mathbb{P}_i = \mathbb{A}_i \mathrm{d}M \mathrm{Var}(\mathbb{P}_i)$ at the symmetric point.

Let $\mathrm{d}\theta := (\mathrm{d}\gamma_1, \mathrm{d}\gamma_2, \mathrm{d}\gamma_3, \mathrm{d}\beta)$, where $\mathrm{d}\beta \in \mathbb{R}^3$. Collecting the coefficients in (179) gives the matrix form

$$\frac{\mathrm{d}\Delta\gamma}{\mathrm{d}t} = - \begin{pmatrix} c_1 & 0 & 0 & v_1 \\ 0 & c_2 & c_2 & v_2 \\ 0 & c_2 & c_2 & v_2 \end{pmatrix} \mathrm{d}\theta. \tag{180}$$

2. The $\Delta\beta$ equation. Similarly,

$$\frac{\mathrm{d}\Delta\beta}{\mathrm{d}t} = -\pi_1 \mathrm{d}\mathbb{P}_1^\mathsf{T} \mathbb{A}_1 \gamma - (1-\pi_1)\mathrm{d}\mathbb{P}_{i\neq 1}^\mathsf{T} \mathbb{A}_{i\neq 1}\gamma. \tag{181}$$

Using again $\mathrm{d}\mathbb{P}_i = \mathbb{A}_i \mathrm{d}M \mathrm{Var}(\mathbb{P}_i)$ and $\mathbb{A}_1 = e_1^\mathsf{T}$, $\mathbb{A}_{i\neq 1} = \hat{e}_1^\mathsf{T}$, we obtain the compact matrix form

$$\frac{\mathrm{d}\Delta\beta}{\mathrm{d}t} = - \begin{pmatrix} v_1^\mathsf{T} & v_2^\mathsf{T} & v_2^\mathsf{T} & C \end{pmatrix} \mathrm{d}\theta. \tag{182}$$

Combining (180) and (182), the linearization reads

$$\frac{\mathrm{d}}{\mathrm{d}t}\begin{pmatrix} \Delta\gamma \\ \Delta\beta \end{pmatrix} = J_0 \, \mathrm{d}\theta,$$

where $J_0$ is exactly the block matrix.

We now verify that $J_0$ is negative semidefinite. Let $d\theta = (d\gamma_1, d\gamma_2, d\gamma_3, d\beta)$ and define

$$d\gamma_+ := \tfrac{1}{2}(d\gamma_2 + d\gamma_3), \qquad d\gamma_- := \tfrac{1}{2}(d\gamma_2 - d\gamma_3).$$

A direct expansion of the quadratic form induced by (178) yields

$$d\theta^{\mathsf{T}} J_0 \, d\theta = -\pi_1 \left\| d\gamma_1 \, \beta + \gamma_1 \, d\beta \right\|_{\mathrm{Var}(\mathbb{P}_1)}^2 - (1 - \pi_1) \left\| d\gamma_+ \, \beta + \gamma_{i \neq 1} \, d\beta \right\|_{\mathrm{Var}(\mathbb{P}_{i \neq 1})}^2. \tag{183}$$

Indeed, for the $i = 1$ block one checks

$$-\pi_1 \Big( (d\gamma_1)^2 \beta^{\mathsf{T}} \mathrm{Var}(\mathbb{P}_1)\beta + 2\gamma_1 \, d\gamma_1 \, d\beta^{\mathsf{T}} \mathrm{Var}(\mathbb{P}_1)\beta + \gamma_1^2 \, d\beta^{\mathsf{T}} \mathrm{Var}(\mathbb{P}_1)d\beta \Big) = -\pi_1 \| d\gamma_1 \beta + \gamma_1 d\beta \|_{\mathrm{Var}(\mathbb{P}_1)}^2.$$

For the low-frequency block, the coefficients $\tfrac{1}{4}(1 - \pi_1)$ in the $(d\gamma_2, d\gamma_3)$-submatrix imply

$$-\frac{1}{4}(1 - \pi_1)\|\beta\|_{\mathrm{Var}(\mathbb{P}_{i \neq 1})}^2 \big(d\gamma_2 + d\gamma_3\big)^2 = -(1 - \pi_1)\|d\gamma_+\beta\|_{\mathrm{Var}(\mathbb{P}_{i \neq 1})}^2,$$

and the cross/$(d\beta, d\beta)$ terms match exactly the remaining pieces of $-(1 - \pi_1)\|d\gamma_+\beta + \gamma_{i \neq 1}d\beta\|_{\mathrm{Var}(\mathbb{P}_{i \neq 1})}^2$, giving (183).

Since $\mathrm{Var}(\mathbb{P}_1) \succeq 0$ and $\mathrm{Var}(\mathbb{P}_{i \neq 1}) \succeq 0$, the right-hand side of (183) is always non-positive, hence $J_0 \preceq 0$. Moreover, $d\gamma_-$ does not appear in (183), which already produces a nontrivial kernel direction; additional kernel directions arise from the scaling invariance $(d\gamma, d\beta) \propto (-\gamma, \beta)$ on the rank-one parametrization. Therefore, the equilibrium is a *degenerate* local minimum restricted to the rank-one manifold. $\qquad\square$

### C.3.4. COMPUTATION OF $f_1$ AND $Q_R^{\mathsf{T}} J_0 Q_R$ FOR THE RANGE EQUATION

**Lemma C.7** (Computation of $f_1$ and $Q_R^{\mathsf{T}} f_1$). *At the symmetric rank-one critical point, we have*

$$\partial_\delta \frac{\partial \mathcal{L}}{\partial \Phi} = 0, \qquad -\partial_\delta \left( \frac{\partial \mathcal{L}}{\partial M} \right) \beta = 0, \qquad -\left( \partial_\delta \left( \frac{\partial \mathcal{L}}{\partial M} \right) \right)^{\mathsf{T}} \gamma = \big( \lambda \gamma_{i \neq 1} + (1 - \lambda)(\pi_1 \gamma_1 + (1 - \pi_1)\gamma_{i \neq 1}) \big) (0, 1, -1)^{\mathsf{T}}.$$

*Consequently,*

$$f_1 = \big( \lambda \gamma_{i \neq 1} + (1 - \lambda)(\pi_1 \gamma_1 + (1 - \pi_1)\gamma_{i \neq 1}) \big) \begin{pmatrix} 0 \\ 0 \\ 0 \\ 0 \\ 1 \\ -1 \end{pmatrix}, \tag{184}$$

*and for the range basis $Q_R = (q_1, q_2, q_3)$ with $q_2 = \frac{1}{\sqrt{2}}(0, 0, 0, 0, 1, -1)$,*

$$Q_R^{\mathsf{T}} f_1 = \begin{pmatrix} 0 \\ \sqrt{2}\big( \lambda \gamma_{i \neq 1} + (1 - \lambda)(\pi_1 \gamma_1 + (1 - \pi_1)\gamma_{i \neq 1}) \big) \\ 0 \end{pmatrix}. \tag{185}$$

*Proof.* We differentiate the explicit gradient formula with respect to $\delta$. We calculate the partial derivatives of $\frac{\partial \mathcal{L}}{\partial M}$ and $\frac{\partial \mathcal{L}}{\partial \Phi}$ with respect to $\delta$, respectively.

1. Computation of $\frac{\partial}{\partial \delta} \frac{\partial \mathcal{L}}{\partial M}$. By definition,

$$\frac{\partial}{\partial \delta} \frac{\partial \mathcal{L}}{\partial M} = \frac{\partial}{\partial \delta} \left( -\sum_i \pi_i \mathbb{A}_i^{\mathsf{T}} (P_i - \mathbb{P}_i) \right)$$

$$= -\sum_i \partial_\delta \pi_i \mathbb{A}_i^{\mathsf{T}} (P_i - \mathbb{P}_i) - \sum_i \pi_i \partial_\delta \mathbb{A}_i^{\mathsf{T}} (P_i - \mathbb{P}_i) - \sum_i \pi_i \mathbb{A}_i^{\mathsf{T}} \partial_\delta (P_i - \mathbb{P}_i)$$

Using $P_i = \lambda e_i^\intercal + (1 - \lambda)\pi^\intercal$, the first term is computed as

$$-\sum_i \partial_\delta \pi_i \mathbb{A}_i^\intercal (P_i - \mathbb{P}_i) = -\mathbb{A}_{i\neq1}^\intercal ((P_2 - \mathbb{P}_2) - (P_3 - \mathbb{P}_3)) = -\mathbb{A}_{i\neq1}^\intercal (0, \lambda, -\lambda)$$

Since $\eta$ is sufficiently large and $\gamma_1 \gamma_{i\neq1} < 0$, we get

$$\partial_\delta \mathbb{A}_1 = (0, 0, 0), \quad \partial_\delta \mathbb{A}_{i\neq1} = \left(0, \frac{1}{1 - \pi_1}, -\frac{1}{1 - \pi_1}\right)$$

Using $\sum_{i\neq1}(P_i - \mathbb{P}_i) = 0$, the second term vanishes.

For the last term, we get

$$\partial_\delta P_i = (0, 1 - \lambda, -(1 - \lambda)), \quad \partial_\delta \mathbb{P}_i = 0$$

As a result,

$$\frac{\partial}{\partial \delta}\frac{\partial \mathcal{L}}{\partial M} = -\begin{pmatrix} 0 \\ \frac{1}{2} \\ \frac{1}{2} \end{pmatrix}(0, \lambda, -\lambda) - \begin{pmatrix} \pi_1 \\ \frac{1}{2}(1 - \pi_1) \\ \frac{1}{2}(1 - \pi_1) \end{pmatrix}(0, 1 - \lambda, -(1 - \lambda)). \tag{186}$$

2. Computation of $\frac{\partial}{\partial \delta}\frac{\partial \mathcal{L}}{\partial \Phi}$. By definition,

$$\frac{\partial}{\partial \delta}\frac{\partial \mathcal{L}}{\partial \Phi} = \frac{\partial}{\partial \delta}\left(-\sum_i \pi_i e_i (P_i - \mathbb{P}_i) M^\intercal \operatorname{Var}(\mathbb{A}_i)\right)$$

$$= -\sum_i \partial_\delta \pi_i e_i (P_i - \mathbb{P}_i) M^\intercal \operatorname{Var}(\mathbb{A}_i) - \sum_i \pi_i e_i \partial_\delta(P_i - \mathbb{P}_i) M^\intercal \operatorname{Var}(\mathbb{A}_i) - \sum_i \pi_i e_i (P_i - \mathbb{P}_i) M^\intercal \partial_\delta \operatorname{Var}(\mathbb{A}_i)$$

Similar to the computation about $\frac{\partial}{\partial \delta}\frac{\partial \mathcal{L}}{\partial M}$, ones can check that $\frac{\partial}{\partial \delta}\frac{\partial \mathcal{L}}{\partial \Phi}$ vanishes.

Multiplying (186) by $\beta$ on the right yields zero because it is proportional to $(0, 1, -1)$ and $\beta_2 = \beta_3$ at the symmetric point. Taking transpose and multiplying by $\gamma$ on the right yields a multiple of $(0, 1, -1)^\intercal$, with the scalar coefficient $\lambda \gamma_{i\neq1} + (1 - \lambda)(\pi_1 \gamma_1 + (1 - \pi_1)\gamma_{i\neq1})$, which gives the stated formula for $f_1$ in (184) under the definition of $f_1$ in the expansion of $-\nabla_\theta \mathcal{L}$.

Finally, (185) follows from $q_2^\intercal f_1 = \sqrt{2} \cdot (\text{scalar})$ and $q_1^\intercal f_1 = q_3^\intercal f_1 = 0$ by orthogonality. $\qquad\square$

**Lemma C.8** (Structure of $Q_R^\intercal J_0 Q_R$ on the range). *Let $\Lambda_R := Q_R^\intercal J_0 Q_R$. Then $\Lambda_R$ is nonsingular, and in particular,*

$$(\Lambda_R)_{22} = -c_1, \qquad c_1 := \pi_1 \gamma_1^2 \mathbb{P}_{1,2} + (1 - \pi_1)\gamma_{i\neq1}^2 \mathbb{P}_{i\neq1,2} > 0. \tag{187}$$

*Equivalently, $\Lambda_R$ has the block structure*

$$\Lambda_R = -\begin{pmatrix} * & 0 & * \\ 0 & c_1 & 0 \\ * & 0 & * \end{pmatrix},$$

*where the starred entries are finite constants determined by the symmetric point, and are not needed in Proposition C.2.*

*Proof.* This follows by substituting the explicit expression of $J_0$ (computed from the linearization on the rank-one manifold) into the orthonormal basis $Q_R = (q_1, q_2, q_3)$.

The key point is the $q_2$ direction. Recall $q_2 = \frac{1}{\sqrt{2}}(0, 0, 0, 0, 1, -1)$, i.e., it lies purely in the $\beta$-difference direction. At the symmetric point, the $\beta$-block of $J_0$ equals

$$J_{0,\beta\beta} = -\left(\pi_1 \gamma_1^2 \operatorname{Var}(\mathbb{P}_1) + (1 - \pi_1)\gamma_{i\neq1}^2 \operatorname{Var}(\mathbb{P}_{i\neq1})\right).$$

A direct computation gives

$$q_2^\intercal J_0 q_2 = -\left(\pi_1 \gamma_1^2 q_2^\intercal \operatorname{Var}(\mathbb{P}_1)q_2 + (1 - \pi_1)\gamma_{i\neq1}^2 q_2^\intercal \operatorname{Var}(\mathbb{P}_{i\neq1})q_2\right) = -\left(\pi_1 \gamma_1^2 \mathbb{P}_{1,2} + (1 - \pi_1)\gamma_{i\neq1}^2 \mathbb{P}_{i\neq1,2}\right),$$

where we used $q_2^\intercal \operatorname{Var}(\mathbb{P}_i)q_2 = \mathbb{P}_{i,2}$ under the symmetric specialization $\mathbb{P}_{i,2} = \mathbb{P}_{i,3}$ (hence $\operatorname{Var}(\mathbb{P}_i)$ acts diagonally on the $(2, -3)$ difference). This proves (187). The remaining entries are obtained similarly and yield the stated block structure, implying $\Lambda_R$ is invertible on the range. $\qquad\square$

C.3.5. COMPUTATION OF CROSS TERM $J_1$

**Lemma C.9** (Derivation of the mixed operator $J_1$). *Write $\theta = (\gamma, \beta) \in \mathbb{R}^3 \times \mathbb{R}^3$, and view $J_1$ as a $2 \times 2$ block operator with respect to the $(\gamma, \beta)$-splitting. Then*

$$J_1 = -\begin{pmatrix} J_{1,\gamma\gamma} & J_{1,\gamma\beta} \\ J_{1,\beta\gamma} & 0 \end{pmatrix} + \begin{pmatrix} 0 & A \\ A^\intercal & 0 \end{pmatrix}, \tag{188}$$

*where*

$$J_{1,\gamma\gamma} = \begin{pmatrix} 0 & 0 & 0 \\ 0 & \beta^\intercal \operatorname{Var}(\mathbb{P}_{i\neq1})\beta & 0 \\ 0 & 0 & -\beta^\intercal \operatorname{Var}(\mathbb{P}_{i\neq1})\beta \end{pmatrix}, \qquad J_{1,\gamma\beta} = \begin{pmatrix} 0 \\ \gamma_{i\neq1}\,\beta^\intercal \operatorname{Var}(\mathbb{P}_{i\neq1}) \\ -\gamma_{i\neq1}\,\beta^\intercal \operatorname{Var}(\mathbb{P}_{i\neq1}) \end{pmatrix}, \tag{189}$$

$$J_{1,\beta\gamma} = \Big( 0, \; \gamma_{i\neq1} \operatorname{Var}(\mathbb{P}_{i\neq1})\beta, \; -\gamma_{i\neq1} \operatorname{Var}(\mathbb{P}_{i\neq1})\beta \Big),$$

*and*

$$A = \begin{pmatrix} 0 & \pi_1(1-\lambda) & -\pi_1(1-\lambda) \\ 0 & \frac{1}{2}\big(\lambda + (1-\pi_1)(1-\lambda)\big) & -\frac{1}{2}\big(\lambda + (1-\pi_1)(1-\lambda)\big) \\ 0 & \frac{1}{2}\big(\lambda + (1-\pi_1)(1-\lambda)\big) & -\frac{1}{2}\big(\lambda + (1-\pi_1)(1-\lambda)\big) \end{pmatrix}. \tag{190}$$

*Proof.* We compute the mixed differential

$$J_1 \;=\; \partial_\delta \Big( \nabla_\theta \big[ -\nabla_\theta \mathcal{L}(\theta, \delta) \big] \Big) \Big|_{(\theta,\delta)=(\theta_*, 0)}.$$

On the rank-one manifold, the $(\gamma, \beta)$-dynamics involve the two components

$$-\frac{\partial \mathcal{L}}{\partial M}\beta, \qquad -\Big(\frac{\partial \mathcal{L}}{\partial M}\Big)^\intercal \gamma,$$

while the $\Phi$-part does not contribute to $J_1$ at the symmetric point (see Step 2 below). Therefore it suffices to compute

$$\partial_\delta \nabla_\theta \Big( -\frac{\partial \mathcal{L}}{\partial M}\beta \Big), \qquad \partial_\delta \nabla_\theta \Big( -\Big(\frac{\partial \mathcal{L}}{\partial M}\Big)^\intercal \gamma \Big).$$

We follow the same route as in the derivation of $J_0$: we first compute $\mathrm{d}(\partial \mathcal{L}/\partial M)$ and $\mathrm{d}(\partial \mathcal{L}/\partial \Phi)$, then take $\partial_\delta$ and finally reassemble the induced variation of the rank-one gradients.

**Step 1: computing $\partial_\delta \nabla_\theta (\partial \mathcal{L}/\partial M)$.** Recall

$$\frac{\partial \mathcal{L}}{\partial M} = -\sum_i \pi_i \mathbb{A}_i^\intercal (P_i - \mathbb{P}_i).$$

Taking $\theta$-differential gives

$$\mathrm{d}\frac{\partial \mathcal{L}}{\partial M} = -\sum_i \pi_i (\mathrm{d}\mathbb{A}_i^\intercal)(P_i - \mathbb{P}_i) - \sum_i \pi_i \mathbb{A}_i^\intercal \mathrm{d}(P_i - \mathbb{P}_i),$$

and since $\mathrm{d}P_i = 0$, we have $\mathrm{d}(P_i - \mathbb{P}_i) = -\mathrm{d}\mathbb{P}_i$. Differentiating w.r.t. $\delta$ and using the product rule yields

$$\begin{aligned}
\partial_\delta \mathrm{d}\frac{\partial \mathcal{L}}{\partial M} = &-\sum_i (\partial_\delta \pi_i)\,(\mathrm{d}\mathbb{A}_i^\intercal)(P_i - \mathbb{P}_i) - \sum_i \pi_i\,\partial_\delta \mathrm{d}\mathbb{A}_i^\intercal (P_i - \mathbb{P}_i) - \sum_i \pi_i\,(\mathrm{d}\mathbb{A}_i^\intercal)\,\partial_\delta(P_i - \mathbb{P}_i) \\
&-\sum_i (\partial_\delta \pi_i)\,\mathbb{A}_i^\intercal(-\mathrm{d}\mathbb{P}_i) - \sum_i \pi_i\,(\partial_\delta \mathbb{A}_i^\intercal)(-\mathrm{d}\mathbb{P}_i) - \sum_i \pi_i\,\mathbb{A}_i^\intercal \partial_\delta(-\mathrm{d}\mathbb{P}_i).
\end{aligned} \tag{191}$$

We now analyze each term. (All computations are evaluated at the symmetric point.)

1. The term $-\sum_i (\partial_\delta \pi_i)(\mathrm{d}\mathbb{A}_i^\intercal)(P_i - \mathbb{P}_i)$: using the structure of $\mathrm{d}\mathbb{A}_i$ and $(P_i - \mathbb{P}_i)\beta = 0$, its contribution vanishes when paired with $\beta$ and with $\gamma$, i.e.,

$$\Big( -\sum_i (\partial_\delta \pi_i)(\mathrm{d}\mathbb{A}_i^\intercal)(P_i - \mathbb{P}_i)\Big)\beta = 0, \qquad \Big( \cdot \Big)^\intercal \gamma = 0.$$

2. The term $-\sum_i \pi_i \partial_\delta \mathrm{d}\mathbb{A}_i^\intercal (P_i - \mathbb{P}_i)$: Since $\mathrm{d}\mathbb{A}_i = e_i^\intercal \Phi \operatorname{Var}(\mathbb{A}_i)$, we get $\partial_\delta \mathrm{d}\mathbb{A}_i = e_i^\intercal \Phi (\partial_\delta \operatorname{Var}(\mathbb{A}_i))$. For $i = 1$, we have $\partial_\delta \operatorname{Var}(\mathbb{A}_1) = 0$ since $\mathbb{A}_1 = e_1^\intercal$. For $i \neq 1$,

$$\partial_\delta \operatorname{Var}(\mathbb{A}_i) = (1 - \pi_1)\left[ \begin{pmatrix} 0 & 0 & 0 \\ 0 & 1 & 0 \\ 0 & 0 & -1 \end{pmatrix} - \begin{pmatrix} 0 \\ 1 \\ -1 \end{pmatrix}\left(0, \frac{1}{2}, \frac{1}{2}\right) - \begin{pmatrix} 0 \\ \frac{1}{2} \\ \frac{1}{2} \end{pmatrix}(0, 1, -1) \right] = 0$$

Hence this term is zero.

3. The term $-\sum_i \pi_i (\mathrm{d}\mathbb{A}_i^\intercal)\partial_\delta(P_i - \mathbb{P}_i)$: since $\partial_\delta(P_i - \mathbb{P}_i) = \partial_\delta P_i$, we obtain

$$-\sum_i \pi_i (\mathrm{d}\mathbb{A}_i^\intercal)\partial_\delta(P_i - \mathbb{P}_i) = \sum_i \pi_i(\mathrm{d}\mathbb{A}_i^\intercal)(0, 1 - \lambda, -(1 - \lambda)).$$

By the symmetric specialization $\beta_2 = \beta_3$ and $\gamma_2 = \gamma_3$, this term also satisfies

$$\Big( \cdot \Big)\beta = 0, \qquad \Big( \cdot \Big)^\intercal \gamma = 0.$$

4. The term $-\sum_i(\partial_\delta \pi_i)\mathbb{A}_i^\intercal(-\mathrm{d}\mathbb{P}_i)$: using $\partial_\delta \pi_2 = -\partial_\delta \pi_3$ and $\mathbb{A}_2 = \mathbb{A}_3$ at $\delta = 0$, we get

$$-\sum_i(\partial_\delta \pi_i)\mathbb{A}_i^\intercal(-\mathrm{d}\mathbb{P}_i) = -\mathbb{A}_2^\intercal(-\mathrm{d}\mathbb{P}_2) + \mathbb{A}_3^\intercal(-\mathrm{d}\mathbb{P}_3) = 0.$$

5. The term $-\sum_i \pi_i \partial_\delta \mathbb{A}_i^\intercal(-\mathrm{d}\mathbb{P}_i)$: We have

$$-\sum_i \pi_i \partial_\delta \mathbb{A}_i^\intercal(-\mathrm{d}\mathbb{P}_i) = \begin{pmatrix} 0 \\ 1 \\ -1 \end{pmatrix}\mathbb{A}_{i\neq1}\mathrm{d}M \operatorname{Var}(\mathbb{P}_i) \tag{192}$$

6. The term $-\sum_i \pi_i \mathbb{A}_i^\intercal \partial_\delta(-\mathrm{d}\mathbb{P}_i)$:

$$-\sum_i \pi_i \mathbb{A}_i^\intercal \partial_\delta(-\mathrm{d}\mathbb{P}_i) = \sum_i \pi_i \mathbb{A}_i^\intercal \partial_\delta (\mathrm{d}\mathbb{A}_i M \operatorname{Var}(\mathbb{P}_i) + \mathbb{A}_i \mathrm{d}M \operatorname{Var}(\mathbb{P}_i))$$

Similar to the previous computation, we have

$$-\sum_i \pi_i \mathbb{A}_i^\intercal \partial_\delta(-\mathrm{d}\mathbb{P}_i) = \mathbb{A}_{i\neq1}^\intercal(0, 1, -1)(\mathrm{d}\gamma)\beta^\intercal \operatorname{Var}(\mathbb{P}_{i\neq1}). \tag{193}$$

The last two terms, $-\sum_i \pi_i (\partial_\delta \mathbb{A}_i^\intercal)(-\mathrm{d}\mathbb{P}_i)$ and $-\sum_i \pi_i \mathbb{A}_i^\intercal \partial_\delta(-\mathrm{d}\mathbb{P}_i)$, produce the only nonzero contribution to $\partial_\delta \mathrm{d}(\partial \mathcal{L}/\partial M)\beta$ along the $(2, -3)$ antisymmetric direction. Collecting them gives

$$\partial_\delta \mathrm{d}\Big(\frac{\partial \mathcal{L}}{\partial M}\Big)\beta = \left( \begin{array}{ccc|c} 0 & 0 & 0 & 0 \\ 0 & \beta^\intercal \operatorname{Var}(\mathbb{P}_{i\neq1})\beta & 0 & \gamma_{i\neq1}\beta^\intercal \operatorname{Var}(\mathbb{P}_{i\neq1}) \\ 0 & 0 & -\beta^\intercal \operatorname{Var}(\mathbb{P}_{i\neq1})\beta & -\gamma_{i\neq1}\beta^\intercal \operatorname{Var}(\mathbb{P}_{i\neq1}) \end{array} \right) \mathrm{d}(\gamma, \beta), \tag{194}$$

which exactly corresponds to the $J_{1,\gamma\gamma}$ and $J_{1,\gamma\beta}$ blocks in (189).

**Step 2: $\partial_\delta \mathrm{d}(\partial \mathcal{L}/\partial \Phi) = 0$.** We differentiate

$$\frac{\partial \mathcal{L}}{\partial \Phi} = -\sum_i \pi_i e_i(P_i - \mathbb{P}_i)M^\intercal \operatorname{Var}(\mathbb{A}_i),$$

and check term by term (product rule) that every contribution vanishes at the symmetric point: the $\partial_\delta \pi_i$-terms cancel by symmetry and the $\partial_\delta$-dependence of $\operatorname{Var}(\mathbb{A}_i)$ does not contribute at $\delta = 0$. Hence $\partial_\delta \mathrm{d}(\partial \mathcal{L}/\partial \Phi) = 0$.

**Step 3: contribution from $\partial_\delta(\partial\mathcal{L}/\partial M)\,\mathrm{d}\beta$.** Using the explicit formula of $\partial_\delta(\partial\mathcal{L}/\partial M)$ (computed previously), we obtain the linear map acting on $\mathrm{d}\beta$:

$$\partial_\delta\!\left(\frac{\partial\mathcal{L}}{\partial M}\right)\mathrm{d}\beta = -\left(\begin{array}{c|cc} 0 & \pi_1(1-\lambda) & -\pi_1(1-\lambda) \\ 0 & \frac{1}{2}(\lambda+(1-\pi_1)(1-\lambda)) & -\frac{1}{2}(\lambda+(1-\pi_1)(1-\lambda)) \\ 0 & \frac{1}{2}(\lambda+(1-\pi_1)(1-\lambda)) & -\frac{1}{2}(\lambda+(1-\pi_1)(1-\lambda)) \end{array}\right)\mathrm{d}\beta, \tag{195}$$

which is exactly the $A$ block in (190) (placed in the $(\gamma,\beta)$ off-diagonal).

**Step 4: assembling $J_1$ from the two dynamics components.** By the chain rule,

$$\partial_\delta \mathrm{d}\!\left(-\frac{\partial\mathcal{L}}{\partial M}\beta\right) = -\left(\partial_\delta\mathrm{d}\frac{\partial\mathcal{L}}{\partial M}\right)\beta - \left(\partial_\delta\frac{\partial\mathcal{L}}{\partial M}\right)\mathrm{d}\beta,$$

so combining (194) and (195) yields the $\gamma$-equation blocks in (188).

Similarly,

$$\partial_\delta \mathrm{d}\!\left(-\left(\frac{\partial\mathcal{L}}{\partial M}\right)^{\mathsf{T}}\gamma\right) = -\left(\partial_\delta\mathrm{d}\frac{\partial\mathcal{L}}{\partial M}\right)^{\mathsf{T}}\gamma - \left(\partial_\delta\frac{\partial\mathcal{L}}{\partial M}\right)^{\mathsf{T}}\mathrm{d}\gamma,$$

which gives the $(\beta,\gamma)$ block $J_{1,\beta\gamma}$ together with the transpose $A^{\mathsf{T}}$ in (188). $\qquad\square$

Next, we will calculate the identity needed in Theorem C.3.

**Lemma C.10.** *Let $\theta = q_2y_2$ be the leading-order reduction (since $\zeta(0,\delta)\sim\delta q_2$). Then*

$$Q_K^{\mathsf{T}}J_1(q_2y_2) = -\frac{\sqrt{2}}{\sqrt{\|\gamma\|^2+\|\beta\|^2}}\left(\begin{array}{c} 0 \\ 0 \\ \pi_1\gamma_1(1-\lambda)+\gamma_{i\neq 1}(\lambda+(1-\pi_1)(1-\lambda)) \end{array}\right)y_2.$$

*Proof.* This can be verified using the expression in Lem. C.9 and by direct calculation. $\qquad\square$

C.3.6. COMPUTATION OF THE BILINEAR FORM $B(\cdot,\cdot)$

Before proceeding with the specific calculations, let's review the following lemma.

**Lemma C.11** (Second differential). *Let $f:\mathbb{R}^d\to\mathbb{R}$ be $C^2$. Then for any $h\in\mathbb{R}^d$,*

$$f(\theta+h) = f(\theta) + \mathrm{d}f(\theta)[h] + \frac{1}{2}\,\mathrm{d}^2 f(\theta)[h,h] + \mathcal{O}(\|h\|^3), \tag{196}$$

*where $\mathrm{d}^2 f(\theta)[u,v] = u^{\mathsf{T}}\nabla^2 f(\theta)\,v$ is the (symmetric) bilinear form induced by the Hessian. The same expansion applies componentwise to vector-valued maps; in particular, for the gradient map $g(\theta) = \nabla f(\theta)$,*

$$g(\theta+h) = g(\theta) + Dg(\theta)\,h + \frac{1}{2}\,D^2 g(\theta)[h,h] + \mathcal{O}(\|h\|^3). \tag{197}$$

Also, we have the following lemma which simplifies the computation.

**Lemma C.12** (A useful identity: $\mathrm{d}\,\mathrm{Var}(\mathbb{A}_{i\neq 1}) = 0$ for the antisymmetric direction). *At $\mathbb{A}_{i\neq 1} = \hat{e}_1 = (0,\frac{1}{2},\frac{1}{2})$, if $\mathrm{d}\mathbb{A}_{i\neq 1} = (0,a,-a)$ for some $a$, then $\mathrm{d}\,\mathrm{Var}(\mathbb{A}_{i\neq 1}) = 0$.*

*Proof.* By definition, $\mathrm{d}\,\mathrm{Var}(\mathbb{A}) = \mathrm{diag}(\mathrm{d}\mathbb{A}) - (\mathrm{d}\mathbb{A})\mathbb{A}^{\mathsf{T}} - \mathbb{A}(\mathrm{d}\mathbb{A})^{\mathsf{T}}$. Substituting $\mathbb{A} = \hat{e}_1$ and $\mathrm{d}\mathbb{A} = (0,a,-a)$ gives exact cancellation of all entries. $\qquad\square$

Consequently, in our regime the second differential of $\mathbb{A}_i$ simplifies to

$$\mathrm{d}^2\mathbb{A}_i = \mathrm{d}^2(\eta\,\gamma_i\gamma^{\mathsf{T}})\,\mathrm{Var}(\mathbb{A}_i), \tag{198}$$

because the potentially present term $\mathrm{d}(\eta\gamma_i\gamma^{\mathsf{T}})\,\mathrm{d}\,\mathrm{Var}(\mathbb{A}_i)$ vanishes (identically for $i=1$ since $\mathrm{d}\mathbb{A}_1 = 0$, and by Lemma C.12 for $i\neq 1$).

Now we will begin the calculation of the bilinear term $B$.

**Second differential of $\frac{\partial \mathcal{L}}{\partial M}$.** Recall

$$\frac{\partial \mathcal{L}}{\partial M} = -\sum_i \pi_i \, \mathbb{A}_i^\intercal (P_i - \mathbb{P}_i). \tag{199}$$

Differentiating once (with $\pi_i$ fixed) yields

$$\mathrm{d}\frac{\partial \mathcal{L}}{\partial M} = -\sum_i \pi_i \, \mathrm{d}\mathbb{A}_i^\intercal (P_i - \mathbb{P}_i) + \sum_i \pi_i \, \mathbb{A}_i^\intercal \, \mathrm{d}\mathbb{P}_i, \tag{200}$$

since $\mathrm{d}(P_i - \mathbb{P}_i) = -\mathrm{d}\mathbb{P}_i$. Differentiating again gives the decomposition

$$\mathrm{d}^2 \frac{\partial \mathcal{L}}{\partial M} = -\sum_i \pi_i \, \mathrm{d}^2\mathbb{A}_i^\intercal (P_i - \mathbb{P}_i) + 2\sum_i \pi_i \, \mathrm{d}\mathbb{A}_i^\intercal \, \mathrm{d}\mathbb{P}_i + \sum_i \pi_i \, \mathbb{A}_i^\intercal \, \mathrm{d}^2\mathbb{P}_i. \tag{201}$$

The coefficient 2 in the middle term is the standard product-rule contribution: it comes once from differentiating $-\sum \pi_i \, \mathrm{d}\mathbb{A}_i^\intercal (P_i - \mathbb{P}_i)$ and once from differentiating $+\sum \pi_i \, \mathbb{A}_i^\intercal \, \mathrm{d}\mathbb{P}_i$.

**On $\mathrm{d}^2\mathbb{P}_i$.** Using $\mathrm{d}\mathbb{P}_i = \mathrm{d}(\mathbb{A}_i M) \operatorname{Var}(\mathbb{P}_i)$, we have

$$\mathrm{d}^2\mathbb{P}_i = \mathrm{d}^2(\mathbb{A}_i M) \operatorname{Var}(\mathbb{P}_i) + \mathrm{d}(\mathbb{A}_i M) \, \mathrm{d}\operatorname{Var}(\mathbb{P}_i). \tag{202}$$

Moreover,

$$\mathrm{d}^2(\mathbb{A}_i M) = \mathrm{d}^2\mathbb{A}_i \, M + 2 \, \mathrm{d}\mathbb{A}_i \, \mathrm{d}M + \mathbb{A}_i \, \mathrm{d}^2 M. \tag{203}$$

**From $\mathrm{d}^2 \frac{\partial \mathcal{L}}{\partial M}$ to the quadratic term in the vector field** In the rank-one dynamics, the $\gamma$-component contains the factor $(\frac{\partial \mathcal{L}}{\partial M})\beta$. At the critical point, $\frac{\partial \mathcal{L}}{\partial M} = 0$, hence

$$\mathrm{d}^2\left(\frac{\partial \mathcal{L}}{\partial M}\beta\right) = \left(\mathrm{d}^2 \frac{\partial \mathcal{L}}{\partial M}\right)\beta + 2\left(\mathrm{d}\frac{\partial \mathcal{L}}{\partial M}\right)\mathrm{d}\beta. \tag{204}$$

An analogous identity holds for $\mathrm{d}^2\big((\frac{\partial \mathcal{L}}{\partial M})^\intercal \gamma\big)$.

**Decomposition into explicit matrix blocks.** We decompose the resulting bilinear form $B(\cdot, \cdot)$ into contributions coming from the different terms in (201)–(202) and from $\mathrm{d}^2 \frac{\partial \mathcal{L}}{\partial \Phi}$. Concretely, for each output coordinate $k$,

$$B_k(\cdot, \cdot) = \sum_\ell B_k^{(\ell)}(\cdot, \cdot), \tag{205}$$

where $B^{(1)}$–$B^{(5)}$ come from the $M$-part and $B^{(6)}$ comes from the $\Phi$-part.

We calculate bilinear term for $1 \le k \le 3$ and $4 \le k \le 6$ respectively.

1. The computation of $B_k$ for $1 \le k \le 3$. We take the second differential of $-\frac{\partial \mathcal{L}}{\partial M}\beta - \eta\left(\frac{\partial \mathcal{L}}{\partial \Phi} + \left(\frac{\partial \mathcal{L}}{\partial \Phi}\right)^\intercal\right)\gamma$,

$$-\mathrm{d}^2\left(\frac{\partial \mathcal{L}}{\partial M}\beta\right) - \mathrm{d}^2\left[\eta\left(\frac{\partial \mathcal{L}}{\partial \Phi} + \left(\frac{\partial \mathcal{L}}{\partial \Phi}\right)^\intercal\right)\gamma\right]$$

The computation of $M$-term and $\Phi$-term is computed as follows.

**$M$-term.**

(a) Contribution from $2(\mathrm{d}(\partial \mathcal{L}/\partial M))\,\mathrm{d}\beta$. This produces the blocks denoted by $B_k^{(1)}$:

$$B_1^{(1)}(\cdot, \cdot) = -\pi_1 \begin{pmatrix} 0 & 0 & 0 & \beta^\intercal \operatorname{Var}(\mathbb{P}_1) \\ 0 & 0 & 0 & 0 \\ 0 & 0 & 0 & 0 \\ \operatorname{Var}(\mathbb{P}_1)\beta & 0 & 0 & 2\gamma_1 \operatorname{Var}(\mathbb{P}_1) \end{pmatrix},$$

$$B_2^{(1)}(\cdot, \cdot) = B_3^{(1)}(\cdot, \cdot) = -(1 - \pi_1) \begin{pmatrix} 0 & 0 & 0 & 0 \\ 0 & 0 & 0 & \frac{1}{4}\beta^\intercal \operatorname{Var}(\mathbb{P}_{i\neq 1}) \\ 0 & 0 & 0 & \frac{1}{4}\beta^\intercal \operatorname{Var}(\mathbb{P}_{i\neq 1}) \\ 0 & \frac{1}{4}\operatorname{Var}(\mathbb{P}_{i\neq 1})\beta & \frac{1}{4}\operatorname{Var}(\mathbb{P}_{i\neq 1})\beta & (1 - \pi_1)\gamma_{i\neq 1} \operatorname{Var}(\mathbb{P}_{i\neq 1}) \end{pmatrix}. \tag{206}$$

(b) Contribution from $\sum_i \mathrm{d}^2 \mathbb{A}_i^\intercal (P_i - \mathbb{P}_i)\beta$. This term vanishes due to $(P_i - \mathbb{P}_i)\beta = 0$ for each $i$.

(c) Contribution from $-2\sum_i \pi_i \mathrm{d}\mathbb{A}_i^\intercal \mathrm{d}\mathbb{P}_i \beta$. Using the expression of $\mathrm{d}\mathbb{A}_i$ and $\mathrm{d}\mathbb{P}_i$,

$$-2\sum_i \pi_i \mathrm{d}\mathbb{A}_i^\intercal \mathrm{d}\mathbb{P}_i \beta = -2(1-\pi_1)\eta\gamma_{i\neq 1} \begin{pmatrix} 0 \\ \frac{1}{4}(\mathrm{d}\gamma_2 - \mathrm{d}\gamma_3) \\ -\frac{1}{4}(\mathrm{d}\gamma_2 - \mathrm{d}\gamma_3) \end{pmatrix} \mathrm{d}(\mathbb{A}_i M)\operatorname{Var}(\mathbb{P}_i)\beta$$

$$= -\frac{1}{2}(1-\pi_1)\eta\gamma_{i\neq 1} \begin{pmatrix} 0 \\ \mathrm{d}\gamma_2 - \mathrm{d}\gamma_3 \\ -(\mathrm{d}\gamma_2 - \mathrm{d}\gamma_3) \end{pmatrix} \mathbb{A}_i \mathrm{d}M \operatorname{Var}(\mathbb{P}_i)\beta$$

This produces the blocks denoted by $B_k^{(2)}$:

$$B_1^{(2)}(\cdot,\cdot) = 0,$$

$$B_2^{(2)}(\cdot,\cdot) = -\frac{1}{2}(1-\pi_1)\eta\gamma_{i\neq 1} \begin{pmatrix} 0 & 0 & 0 & 0 \\ 0 & \frac{1}{2}\|\beta\|^2_{\operatorname{Var}(\mathbb{P}_{i\neq 1})} & 0 & \frac{1}{2}\gamma_{i\neq 1}\beta^\intercal \operatorname{Var}(\mathbb{P}_{i\neq 1}) \\ 0 & 0 & -\frac{1}{2}\|\beta\|^2_{\operatorname{Var}(\mathbb{P}_{i\neq 1})} & -\frac{1}{2}\gamma_{i\neq 1}\beta^\intercal \operatorname{Var}(\mathbb{P}_{i\neq 1}) \\ 0 & \frac{1}{2}\gamma_{i\neq 1}\operatorname{Var}(\mathbb{P}_{i\neq 1})\beta & -\frac{1}{2}\gamma_{i\neq 1}\operatorname{Var}(\mathbb{P}_{i\neq 1})\beta & 0 \end{pmatrix},$$

$$B_3^{(2)}(\cdot,\cdot) = -B_2^{(2)}(\cdot,\cdot).$$

$$(207)$$

(d) Contribution from $-\sum_i \pi_i \mathbb{A}_i^\intercal \mathrm{d}^2\mathbb{P}_i \beta$. We further split into:

• Terms contributed by $-2\sum_i \pi_i \mathbb{A}_i^\intercal \mathrm{d}\mathbb{A}_i \mathrm{d}M \operatorname{Var}(\mathbb{P}_i)\beta$:

$$B_1^{(3)}(\cdot,\cdot) = 0$$

$$B_2^{(3)}(\cdot,\cdot) = -\frac{1}{4}(1-\pi_1)\eta\gamma_{i\neq 1} \begin{pmatrix} 0 & 0 & 0 & 0 \\ 0 & \beta^\intercal \operatorname{Var}(\mathbb{P}_{i\neq 1})\beta & -\beta^\intercal \operatorname{Var}(\mathbb{P}_{i\neq 1})\beta & 0 \\ 0 & -\beta^\intercal \operatorname{Var}(\mathbb{P}_{i\neq 1})\beta & \beta^\intercal \operatorname{Var}(\mathbb{P}_{i\neq 1})\beta & 0 \\ 0 & 0 & 0 & 0 \end{pmatrix}$$

$$(208)$$

$$B_3^{(3)}(\cdot,\cdot) = B_2^{(3)}(\cdot,\cdot)$$

• Terms contributed by $-\sum_i \pi_i \mathbb{A}_i^\intercal \mathbb{A}_i \mathrm{d}^2 M \operatorname{Var}(\mathbb{P}_i)\beta$. The matrix form is

$$B_1^{(4)} = -\pi_1 \begin{pmatrix} 0 & 0 & 0 & \beta^\intercal \operatorname{Var}(\mathbb{P}_1) \\ 0 & 0 & 0 & 0 \\ 0 & 0 & 0 & 0 \\ \operatorname{Var}(\mathbb{P}_1)\beta & 0 & 0 & 0 \end{pmatrix}$$

$$(209)$$

$$B_2^{(4)} = B_3^{(4)} = -(1-\pi_1) \begin{pmatrix} 0 & 0 & 0 & 0 \\ 0 & 0 & 0 & \frac{1}{4}\beta^\intercal \operatorname{Var}(\mathbb{P}_{i\neq 1}) \\ 0 & 0 & 0 & \frac{1}{4}\beta^\intercal \operatorname{Var}(\mathbb{P}_{i\neq 1}) \\ 0 & \frac{1}{4}\operatorname{Var}(\mathbb{P}_{i\neq 1})\beta & \frac{1}{4}\operatorname{Var}(\mathbb{P}_{i\neq 1})\beta & 0 \end{pmatrix}$$

• Terms contributed by $-\sum_i \pi_i \mathbb{A}_i^\intercal \mathrm{d}(\mathbb{A}_i M)\mathrm{d}\operatorname{Var}(\mathbb{P}_i)\beta$. The matrix form is of the shape The matrix form is of the shape

$$B_1^{(5)} = -\pi_1 \begin{pmatrix} c_1 & 0 & 0 & c_2 \\ 0 & 0 & 0 & 0 \\ 0 & 0 & 0 & 0 \\ c_2^\intercal & 0 & 0 & c_3 \end{pmatrix}$$

$$(210)$$

where

$$c_1 = \beta^\intercal \operatorname{diag}(\operatorname{Var}(\mathbb{P}_1))\beta - 2(\mathbb{P}_1\beta)\|\beta\|^2_{\operatorname{Var}(\mathbb{P}_1)}$$

$$c_2 = \frac{1}{2}\gamma_1 \left( \beta^{\odot 2,\intercal}\operatorname{Var}(\mathbb{P}_1) + \beta^\intercal \odot \beta^\intercal \operatorname{Var}(\mathbb{P}_1) - 3(\mathbb{P}_1\beta)\beta^\intercal \operatorname{Var}(\mathbb{P}_1) - \beta^\intercal \operatorname{Var}(\mathbb{P}_1)\beta\mathbb{P}_1 \right)$$

$$(211)$$

And

$$c_3(\mathrm{d}\beta, \mathrm{d}\beta) = \gamma_1^2 \left( \mathrm{d}\beta^{\mathsf{T}} \operatorname{diag}(\mathrm{d}\beta^{\mathsf{T}} \operatorname{Var}(\mathbb{P}_1))\beta - \mathrm{d}\beta^{\mathsf{T}}\mathbb{P}_1^{\mathsf{T}}\mathrm{d}\beta^{\mathsf{T}} \operatorname{Var} \mathbb{P}_1\beta - \mathrm{d}\beta^{\mathsf{T}} \operatorname{Var}(P_1)\mathrm{d}\beta\mathbb{P}_1\beta \right).$$

Writing into the matrix, we get

$$c_3 = \gamma_1^2 \left( \operatorname{diag}(\mathbb{P}_1 \odot \beta) - \frac{1}{2} \left( (\mathbb{P}_1^{\mathsf{T}}\mathbb{P}_1 \odot \beta) + (\mathbb{P}_1^{\mathsf{T}}\mathbb{P}_1 \odot \beta)^{\mathsf{T}} \right) \right.$$
$$\left. - \frac{1}{2} \left( (\mathbb{P}_1^{\mathsf{T}}\beta^{\mathsf{T}} \operatorname{Var}(\mathbb{P}_1)) + (\mathbb{P}_1^{\mathsf{T}}\beta^{\mathsf{T}} \operatorname{Var}(\mathbb{P}_1))^{\mathsf{T}} \right) - (\mathbb{P}_1\beta) \operatorname{Var}(\mathbb{P}_1) \right) \tag{212}$$

And

$$B_2^{(5)} = B_3^{(5)} = -\frac{1}{2}(1 - \pi_1) \begin{pmatrix} 0 & 0 & 0 & 0 \\ 0 & c_4 & c_4 & c_5 \\ 0 & c_4 & c_4 & c_5 \\ 0 & c_5^{\mathsf{T}} & c_5^{\mathsf{T}} & c_6 \end{pmatrix} \tag{213}$$

where $c_6(\mathrm{d}\beta, \mathrm{d}\beta) = \gamma_{i\neq 1}^2 \left( \mathrm{d}\beta^{\mathsf{T}} \operatorname{diag}(\mathrm{d}\beta^{\mathsf{T}} \operatorname{Var}(\mathbb{P}_{i\neq 1}))\beta - \mathrm{d}\beta^{\mathsf{T}}\mathbb{P}_{i\neq 1}^{\mathsf{T}}\mathrm{d}\beta^{\mathsf{T}} \operatorname{Var} \mathbb{P}_{i\neq 1}\beta - \mathrm{d}\beta^{\mathsf{T}} \operatorname{Var}(P_1)\mathrm{d}\beta\mathbb{P}_{i\neq 1}\beta \right)$
and the matrix form is:

$$c_6 = \operatorname{diag}(\mathbb{P}_{i\neq 1} \odot \beta) - \frac{1}{2} \left( (\mathbb{P}_{i\neq 1}^{\mathsf{T}}\mathbb{P}_{i\neq 1} \odot \beta) + (\mathbb{P}_{i\neq 1}^{\mathsf{T}}\mathbb{P}_{i\neq 1} \odot \beta)^{\mathsf{T}} \right)$$
$$- \frac{1}{2} \left( (\mathbb{P}_{i\neq 1}^{\mathsf{T}}\beta^{\mathsf{T}} \operatorname{Var}(\mathbb{P}_{i\neq 1})) + (\mathbb{P}_{i\neq 1}^{\mathsf{T}}\beta^{\mathsf{T}} \operatorname{Var}(\mathbb{P}_{i\neq 1}))^{\mathsf{T}} \right) - (\mathbb{P}_{i\neq 1}\beta) \operatorname{Var}(\mathbb{P}_{i\neq 1}) \tag{214}$$

$\Phi$-**term.** Using the fact that $\frac{\partial \mathcal{L}}{\partial \Phi} = 0$ and $\mathrm{d}\frac{\partial \mathcal{L}}{\partial \Phi} = 0$, we get

$$-\mathrm{d}^2 \left[ \eta \left( \frac{\partial \mathcal{L}}{\partial \Phi} + \left( \frac{\partial \mathcal{L}}{\partial \Phi} \right)^{\mathsf{T}} \right) \gamma \right] = -\eta \, \mathrm{d}^2 \left( \frac{\partial \mathcal{L}}{\partial \Phi} + \left( \frac{\partial \mathcal{L}}{\partial \Phi} \right)^{\mathsf{T}} \right) \gamma$$

Differentiating twice $\frac{\partial \mathcal{L}}{\partial \Phi} = -\sum_i \pi_i e_i (P_i - \mathbb{P}_i)M^{\mathsf{T}} \operatorname{Var}(\mathbb{A}_i)$ gives

$$\mathrm{d}^2 \frac{\partial \mathcal{L}}{\partial \Phi} = 2\sum_i \pi_i e_i \, \mathrm{d}\mathbb{P}_i \, \mathrm{d}M^{\mathsf{T}} \operatorname{Var}(\mathbb{A}_i) - \sum_i \pi_i e_i (P_i - \mathbb{P}_i) \, \mathrm{d}^2 M^{\mathsf{T}} \operatorname{Var}(\mathbb{A}_i), \tag{215}$$

where the second term vanishes after contraction with $\gamma$ at the critical point by the same symmetry used in the first-order analysis. The first term yields $B^{(6)}$ blocks:

$$B_1^{(6)}(\cdot, \cdot) = 0,$$

$$B_2^{(6)}(\cdot, \cdot) = -B_3^{(6)}(\cdot, \cdot) = -\frac{1}{2} \eta \gamma_{i\neq 1}(1 - \pi_1) \begin{pmatrix} 0 & 0 & 0 & 0 \\ 0 & \frac{1}{2}\|\beta\|^2_{\operatorname{Var}(\mathbb{P}_{i\neq 1})} & 0 & \frac{1}{2}\beta^{\mathsf{T}} \operatorname{Var}(\mathbb{P}_{i\neq 1}) \\ 0 & 0 & -\frac{1}{2}\|\beta\|^2_{\operatorname{Var}(\mathbb{P}_{i\neq 1})} & -\frac{1}{2}\beta^{\mathsf{T}} \operatorname{Var}(\mathbb{P}_{i\neq 1}) \\ 0 & \frac{1}{2} \operatorname{Var}(\mathbb{P}_{i\neq 1})\beta & -\frac{1}{2} \operatorname{Var}(\mathbb{P}_{i\neq 1})\beta & 0 \end{pmatrix}. \tag{216}$$

2. The computation of $B_k$ for $4 \leq k \leq 6$. Similarly, we compute the $\mathrm{d}^2 \left( \left( \frac{\partial \mathcal{L}}{\partial M} \right)^{\mathsf{T}} \gamma \right)$

$$\mathrm{d}^2 \left( \left( \frac{\partial \mathcal{L}}{\partial M} \right)^{\mathsf{T}} \gamma \right) = \mathrm{d}^2 \left( \frac{\partial \mathcal{L}}{\partial M} \right)^{\mathsf{T}} \gamma + 2\mathrm{d} \left( \frac{\partial \mathcal{L}}{\partial M} \right)^{\mathsf{T}} \mathrm{d}\gamma$$
$$= \left( 2\sum_i \pi_i \mathrm{d}\mathbb{A}_i^{\mathsf{T}}\mathrm{d}\mathbb{P}_i + \sum_i \pi_i \mathbb{A}_i^{\mathsf{T}}\mathrm{d}^2\mathbb{P}_i \right)^{\mathsf{T}} \gamma + 2 \left( \sum_i \pi_i \mathbb{A}_i^{\mathsf{T}}\mathbb{A}_i \mathrm{d}M \operatorname{Var}(\mathbb{P}_i) \right)^{\mathsf{T}} \mathrm{d}\gamma \tag{217}$$
$$= \sum_i \pi_i \mathrm{d}^2\mathbb{P}_i^{\mathsf{T}}\mathbb{A}_i\gamma + 2\sum_i \pi_i \operatorname{Var}(\mathbb{P}_i)\mathrm{d}M^{\mathsf{T}}\mathbb{A}_i^{\mathsf{T}}\mathbb{A}_i\mathrm{d}\gamma$$

For the term $2\sum_i \pi_i \operatorname{Var}(\mathbb{P}_i)\mathrm{d}M^\intercal\mathbb{A}_i^\intercal\mathbb{A}_i\mathrm{d}\gamma$, we get

$$2\sum_i \pi_i \operatorname{Var}(\mathbb{P}_i)\mathrm{d}M^\intercal\mathbb{A}_i^\intercal\mathbb{A}_i\mathrm{d}\gamma = 2\pi_1 \operatorname{Var}(\mathbb{P}_1)\mathrm{d}(\beta\gamma_1)\mathrm{d}\gamma_1 + 2(1-\pi_1)\operatorname{Var}(\mathbb{P}_{i\neq1})\mathrm{d}(\frac{1}{2}(\gamma_2+\gamma_3)\beta)\mathrm{d}(\frac{1}{2}(\gamma_2+\gamma_3)) \tag{218}$$

Writing into the matrix form, we get

$$\begin{aligned}
B_k^{(1)} = &-2\begin{pmatrix}
\pi_1 \operatorname{Var}(\mathbb{P}_1)_{k-3}\beta & 0 & 0 & \frac{1}{2}\pi_1\gamma_1 \operatorname{Var}(\mathbb{P}_1)_{k-3} \\
0 & 0 & 0 & 0 \\
0 & 0 & 0 & 0 \\
\frac{1}{2}\pi_1\gamma_1 \operatorname{Var}(\mathbb{P}_1)_{k-3}^\intercal & 0 & 0 & 0
\end{pmatrix} \\
&-2\begin{pmatrix}
0 & 0 & 0 & 0 \\
0 & \frac{1}{4}(1-\pi_1)\operatorname{Var}(\mathbb{P}_{i\neq1})_{k-3}\beta & \frac{1}{4}(1-\pi_1)\operatorname{Var}(\mathbb{P}_{i\neq1})_{k-3}\beta & \frac{1}{4}\operatorname{Var}(\mathbb{P}_{i\neq1})_{k-3} \\
0 & \frac{1}{4}(1-\pi_1)\operatorname{Var}(\mathbb{P}_{i\neq1})_{k-3}\beta & \frac{1}{4}(1-\pi_1)\operatorname{Var}(\mathbb{P}_{i\neq1})_{k-3}\beta & \frac{1}{4}\operatorname{Var}(\mathbb{P}_{i\neq1})_{k-3} \\
0 & \frac{1}{4}\operatorname{Var}(\mathbb{P}_{i\neq1})_{k-3}^\intercal & \frac{1}{4}\operatorname{Var}(\mathbb{P}_{i\neq1})_{k-3}^\intercal & 0
\end{pmatrix}
\end{aligned} \tag{219}$$

Recall that $\mathrm{d}^2\mathbb{P}_i = 2\mathrm{d}\mathbb{A}_i\mathrm{d}M\operatorname{Var}(\mathbb{P}_i) + \mathbb{A}_i\mathrm{d}^2M\operatorname{Var}(\mathbb{P}_i) + \mathrm{d}(\mathbb{A}_iM)\mathrm{d}\operatorname{Var}(\mathbb{P}_i)$, we have

$$\sum_i \pi_i\mathrm{d}^2\mathbb{P}_i^\intercal\mathbb{A}_i\gamma = \sum_i \pi_i(2\mathrm{d}\mathbb{A}_i\mathrm{d}M\operatorname{Var}(\mathbb{P}_i) + \mathbb{A}_i\mathrm{d}^2M\operatorname{Var}(\mathbb{P}_i) + \mathrm{d}(\mathbb{A}_iM)\mathrm{d}\operatorname{Var}(\mathbb{P}_i))^\intercal\mathbb{A}_i\gamma \tag{220}$$

The first term contributes to

$$B_k^{(2)} = -\frac{1}{2}(1-\pi_1)\eta\gamma_{i\neq1}^2\begin{pmatrix}
0 & 0 & 0 & 0 \\
0 & \operatorname{Var}(\mathbb{P}_{i\neq1})_{k-3}\beta & -\operatorname{Var}(\mathbb{P}_{i\neq1})_{k-3}\beta & 0 \\
0 & -\operatorname{Var}(\mathbb{P}_{i\neq1})_{k-3}\beta & \operatorname{Var}(\mathbb{P}_{i\neq1})_{k-3}\beta & 0 \\
0 & 0 & 0 & 0
\end{pmatrix} \tag{221}$$

The second term contributes to

$$\begin{aligned}
B_k^{(3)} = &-2\pi_1\gamma_1\begin{pmatrix}
0 & 0 & 0 & \frac{1}{2}\operatorname{Var}(\mathbb{P}_1)_{k-3} \\
0 & 0 & 0 & 0 \\
0 & 0 & 0 & 0 \\
\frac{1}{2}\operatorname{Var}(\mathbb{P}_1)_{k-3}^\intercal & 0 & 0 & 0
\end{pmatrix} \\
&-2(1-\pi_1)\gamma_{i\neq1}\begin{pmatrix}
0 & 0 & 0 & 0 \\
0 & 0 & 0 & \frac{1}{4}\operatorname{Var}(\mathbb{P}_{i\neq1})_{k-3} \\
0 & 0 & 0 & \frac{1}{4}\operatorname{Var}(\mathbb{P}_{i\neq1})_{k-3} \\
0 & \frac{1}{4}\operatorname{Var}(\mathbb{P}_{i\neq1})_{k-3}^\intercal & \frac{1}{4}\operatorname{Var}(\mathbb{P}_{i\neq1})_{k-3}^\intercal & 0
\end{pmatrix}
\end{aligned} \tag{222}$$

We consider the action of the last term on $\mathrm{d}\beta^2$:

$$\begin{aligned}
\begin{pmatrix} B_4^{(4)} \\ B_5^{(4)} \\ B_6^{(4)} \end{pmatrix}(\mathrm{d}\beta,\mathrm{d}\beta) = &-\pi_1\gamma_1^3\left(\operatorname{diag}(\mathrm{d}\beta^\intercal\operatorname{Var}(\mathbb{P}_1)) - \mathbb{P}_1^\intercal(\mathrm{d}\beta^\intercal\operatorname{Var}(\mathbb{P}_1)) - \operatorname{Var}(\mathbb{P}_1)\mathrm{d}\beta\mathbb{P}_1\right)\mathrm{d}\beta \\
&-(1-\pi_1)\gamma_{i\neq1}^3\left(\operatorname{diag}(\mathrm{d}\beta^\intercal\operatorname{Var}(\mathbb{P}_{i\neq1})) - \mathbb{P}_{i\neq1}^\intercal(\mathrm{d}\beta^\intercal\operatorname{Var}(\mathbb{P}_{i\neq1})) - \operatorname{Var}(\mathbb{P}_{i\neq1})\mathrm{d}\beta\mathbb{P}_{i\neq1}\right)\mathrm{d}\beta
\end{aligned} \tag{223}$$

**Lemma C.13** (Kernel-equation identities used in Theorem C.3). *Let $\theta = q_2 y_2$ be the leading-order reduction (since $\zeta(0,\delta) \sim \delta q_2$). Then*

$$\frac{1}{2}Q_K^\intercal B(q_2 y_2, q_2 y_2) = \frac{1}{\sqrt{\|\gamma\|^2 + \|\beta\|^2}}\begin{pmatrix} 0 \\ 0 \\ \pi_1\gamma_1^2\mathbb{P}_{1,2} + (1-\pi_1)\gamma_2^2\mathbb{P}_{i\neq1,2} \end{pmatrix}y_2^2.$$

*Proof.* We first calculate $B(q_2 y_2, q_2 y_2)$, and then calculate its projection onto the kernel basis. We calculate the cases where $1 \leq k \leq 3$ and $4 \leq k \leq 6$ respectively, and then combine them into the form we want.

1. The computation of the cases where $1 \leq k \leq 3$. From Eq. (206), we get

$$
\begin{pmatrix} B_1^{(1)}(q_2 y_2, q_2 y_2) \\ B_2^{(1)}(q_2 y_2, q_2 y_2) \\ B_3^{(1)}(q_2 y_2, q_2 y_2) \end{pmatrix} = -\frac{1}{2} \begin{pmatrix} 4\pi_1 \gamma_1 \mathbb{P}_{1,2} \\ 2(1 - \pi_1)\gamma_{i\neq 1}\mathbb{P}_{i\neq 1,2} \\ 2(1 - \pi_1)\gamma_{i\neq 1}\mathbb{P}_{i\neq 1,2} \end{pmatrix} y_2^2
\tag{224}
$$

For $l$ from 2 to 4 and $l = 6$, their contribution vanishes. For $l = 5$, the contribution is

$$
\begin{pmatrix} B_1^{(5)}(q_2 y_2, q_2 y_2) \\ B_2^{(5)}(q_2 y_2, q_2 y_2) \\ B_3^{(5)}(q_2 y_2, q_2 y_2) \end{pmatrix} = -\frac{1}{2} \begin{pmatrix} 2\pi_1 \gamma_1^2 (\mathbb{P}_{1,2}\beta_2 - \mathbb{P}_{1,2}\mathbb{P}_1\beta) \\ (1 - \pi_1)\gamma_{i\neq 1}^2 (\mathbb{P}_{i\neq 1,2}\beta_2 - \mathbb{P}_{i\neq 1,2}\mathbb{P}_{i\neq 1}\beta) \\ (1 - \pi_1)\gamma_{i\neq 1}^2 (\mathbb{P}_{i\neq 1,2}\beta_2 - \mathbb{P}_{i\neq 1,2}\mathbb{P}_{i\neq 1}\beta) \end{pmatrix} y_2^2
\tag{225}
$$

2. The computation of the cases where $4 \leq k \leq 6$. For $l = 1, 2, 3$, their contribution vanishes. Then contribution of $l = 4$ case is

$$
-\frac{1}{2}\pi_1 \gamma_1^3 \begin{pmatrix} 0 \\ 0 \\ 0 \\ \begin{pmatrix} 0 \\ \mathbb{P}_{1,2} \\ \mathbb{P}_{1,2} \end{pmatrix} - 2\mathbb{P}_{1,2}\mathbb{P}_1^\mathsf{T} \end{pmatrix} - \frac{1}{2}(1 - \pi_1)\gamma_{i\neq 1}^3 \begin{pmatrix} 0 \\ 0 \\ 0 \\ \begin{pmatrix} 0 \\ \mathbb{P}_{i\neq 1,2} \\ \mathbb{P}_{i\neq 1,2} \end{pmatrix} - 2\mathbb{P}_{i\neq 1,2}\mathbb{P}_{i\neq 1}^\mathsf{T} \end{pmatrix}
\tag{226}
$$

Sum them up, we get $B(q_2 y_2, q_2 y_2)$. Then by direct computation, we get the projection onto the kernel directions.

$$
\frac{1}{2}Q_K^\mathsf{T} B(q_2 y_2, q_2 y_2) = \frac{1}{4\sqrt{\|\gamma\|^2 + \|\beta\|^2}} \begin{pmatrix} 0 \\ 0 \\ 4\pi_1\gamma_1^2\mathbb{P}_{1,2} + 4(1 - \pi_1)\gamma_2^2\mathbb{P}_{i\neq 1,2} \end{pmatrix} y_2^2.
\tag{227}
$$

$\square$

### C.3.7. COMPUTATION OF SECOND ORDER DERIVATIVE $f_2$

The calculation about $f_2 = \partial_\delta^2(-\nabla\mathcal{L})$ is summarized by following lemma.

**Lemma C.14.** *The second derivative of* $-\nabla\mathcal{L}$ *with respect to perturbation parameter* $\delta$ *vanishes, i.e.* $f_2 = 0$.

*Proof.* We compute $\partial_\delta^2 \frac{\partial\mathcal{L}}{\partial M}$ and $\partial_\delta^2 \frac{\partial\mathcal{L}}{\partial\Phi}$ as follows.

1. The computation of $\partial_\delta^2 \frac{\partial\mathcal{L}}{\partial M}$. By definition,

$$
\begin{aligned}
\partial_\delta^2 \frac{\partial\mathcal{L}}{\partial M} &= \partial_\delta^2 \left( -\sum_i \pi_i \mathbb{A}_i^\mathsf{T}(P_i - \mathbb{P}_i) \right) \\
&= \partial_\delta \left( -\sum_i \partial_\delta\pi_i \mathbb{A}_i^\mathsf{T}(P_i - \mathbb{P}_i) - \sum_i \pi_i \partial_\delta\mathbb{A}_i^\mathsf{T}(P_i - \mathbb{P}_i) - \sum_i \pi_i \mathbb{A}_i^\mathsf{T}\partial_\delta(P_i - \mathbb{P}_i) \right) \\
&= -\sum_i \partial_\delta^2\pi_i \mathbb{A}_i^\mathsf{T}(P_i - \mathbb{P}_i) - 2\sum_i \partial_\delta\pi_i \partial_\delta\mathbb{A}_i^\mathsf{T}(P_i - \mathbb{P}_i) - 2\sum_i \partial_\delta\pi_i \mathbb{A}_i^\mathsf{T}\partial_\delta(P_i - \mathbb{P}_i) \\
&\quad - \sum_i \pi_i \partial_\delta^2\mathbb{A}_i^\mathsf{T}(P_i - \mathbb{P}_i) - 2\sum_i \pi_i \partial_\delta\mathbb{A}_i^\mathsf{T}\partial_\delta(P_i - \mathbb{P}_i) - \sum_i \pi_i \mathbb{A}_i^\mathsf{T}\partial_\delta^2(P_i - \mathbb{P}_i) \\
&= -2\sum_i \partial_\delta\pi_i \partial_\delta\mathbb{A}_i^\mathsf{T}(P_i - \mathbb{P}_i) - 2\sum_i \partial_\delta\pi_i \mathbb{A}_i^\mathsf{T}\partial_\delta(P_i - \mathbb{P}_i) - 2\sum_i \pi_i \partial_\delta\mathbb{A}_i^\mathsf{T}\partial_\delta(P_i - \mathbb{P}_i)
\end{aligned}
$$

The last equality uses the second order derivative of $\pi_i$, $\mathbb{A}_i$, and $P_i - \mathbb{P}_i$ with respect to $\delta$ vanishes. Using the fact that $\partial_\delta\mathbb{A}_i$ is of the shape like $(0, a, -a)$ and $(P_i - \mathbb{P}_i)\beta = 0$. The contribution of the first term vanishes. Similarly, the third term vanishes. For the second term,

$$
-2\sum_i \partial_\delta\pi_i \mathbb{A}_i^\mathsf{T}\partial_\delta(P_i - \mathbb{P}_i) = -2\mathbb{A}_{i\neq 1}^\mathsf{T}(\partial_\delta P_2 - \partial_\delta P_3) = 0
$$

Thus, this term makes no contribution.

2. The computation of $\partial_\delta^2 \frac{\partial \mathcal{L}}{\partial \Phi}$. By definition, $\partial_\delta^2 \frac{\partial \mathcal{L}}{\partial \Phi} = \partial_\delta^2 \left( -\sum_i \pi_i\, e_i \left( P_i - \mathbb{P}_i \right) M^\intercal \operatorname{Var}(\mathbb{A}_i) \right)$. In particular,

$$\partial_\delta \left( -\sum_i \partial_\delta \pi_i\, e_i \left( P_i - \mathbb{P}_i \right) M^\intercal \operatorname{Var}(\mathbb{A}_i) - \sum_i \pi_i\, e_i \partial_\delta \left( P_i - \mathbb{P}_i \right) M^\intercal \operatorname{Var}(\mathbb{A}_i) - \sum_i \pi_i\, e_i \left( P_i - \mathbb{P}_i \right) M^\intercal \partial_\delta \operatorname{Var}(\mathbb{A}_i) \right)$$

$$= -\sum_i \partial_\delta^2 \pi_i\, e_i \left( P_i - \mathbb{P}_i \right) M^\intercal \operatorname{Var}(\mathbb{A}_i) - 2\sum_i \partial_\delta \pi_i\, e_i \partial_\delta \left( P_i - \mathbb{P}_i \right) M^\intercal \operatorname{Var}(\mathbb{A}_i) - 2\sum_i \partial_\delta \pi_i\, e_i \left( P_i - \mathbb{P}_i \right) M^\intercal \partial_\delta \operatorname{Var}(\mathbb{A}_i)$$

$$- \sum_i \pi_i\, e_i \partial_\delta^2 \left( P_i - \mathbb{P}_i \right) M^\intercal \operatorname{Var}(\mathbb{A}_i) - 2\sum_i \pi_i\, e_i \partial_\delta \left( P_i - \mathbb{P}_i \right) M^\intercal \partial_\delta \operatorname{Var}(\mathbb{A}_i) - \sum_i \pi_i\, e_i \left( P_i - \mathbb{P}_i \right) M^\intercal \partial_\delta^2 \operatorname{Var}(\mathbb{A}_i)$$

By direct computation, this term is zero.

$\square$

### C.3.8. STABILITY ON THE KERNEL DIRECTIONS

To account for stability on the manifold, we need to calculate the perturbed Hessian matrix. By the expansion of $-\nabla \mathcal{L}$, we get

$$-\nabla_\theta^2 \mathcal{L}(\theta, \delta) = J_0 + \frac{1}{2}\nabla_\theta B(\theta, \theta) + \delta J_1 + \mathcal{O}(\delta^2)$$

Substitute $\theta = q_2 y_2$ into the expression, we get the perturbed hessian matrix

**Lemma C.15** (Perturbed hessian matrix). *The expression of the perturbed hessian matrix is*

$$J_{pert} = J_0 + H_1 + \mathcal{O}(\delta^2), \tag{228}$$

*where*

$$H_1 = -c\delta \left( \begin{pmatrix} 0 & B \\ B^\intercal & 0 \end{pmatrix} + \begin{pmatrix} 0 & 0 \\ 0 & C \end{pmatrix} \right) + \delta J_1, \tag{229}$$

*in which* $c = \frac{\lambda \gamma_{i\neq 1} + (1-\lambda)(\pi_1 \gamma_1 + (1-\pi_1)\gamma_{i\neq 1})}{\pi_1 \gamma_1^2 \mathbb{P}_{1,2} + (1-\pi_1)\gamma_{i\neq 1}^2 \mathbb{P}_{i\neq 1,2}}$,

$$B = \begin{pmatrix} 0 & 2\pi_1\gamma_1 \mathbb{P}_{1,2} & -2\pi_1\gamma_1 \mathbb{P}_{1,2} \\ 0 & (1-\pi_1)\gamma_{i\neq 1}\mathbb{P}_{i\neq 1,2} & -(1-\pi_1)\gamma_{i\neq 1}\mathbb{P}_{i\neq 1,2} \\ 0 & (1-\pi_1)\gamma_{i\neq 1}\mathbb{P}_{i\neq 1,2} & -(1-\pi_1)\gamma_{i\neq 1}\mathbb{P}_{i\neq 1,2} \end{pmatrix}$$
$$+ \begin{pmatrix} 0 & \pi_1\gamma_1^2(\mathbb{P}_{1,2}\beta_2 - \mathbb{P}_{1,2}(\mathbb{P}_1\beta)) & -\pi_1\gamma_1^2(\mathbb{P}_{1,2}\beta_2 - \mathbb{P}_{1,2}(\mathbb{P}_1\beta)) \\ 0 & \frac{1}{2}(1-\pi_1)\gamma_{i\neq 1}^2(\mathbb{P}_{i\neq 1,2}\beta_2 - \mathbb{P}_{i\neq 1,2}(\mathbb{P}_{i\neq 1}\beta)) & -\frac{1}{2}(1-\pi_1)\gamma_{i\neq 1}^2(\mathbb{P}_{i\neq 1,2}\beta_2 - \mathbb{P}_{i\neq 1,2}(\mathbb{P}_{i\neq 1}\beta)) \\ 0 & \frac{1}{2}(1-\pi_1)\gamma_{i\neq 1}^2(\mathbb{P}_{i\neq 1,2}\beta_2 - \mathbb{P}_{i\neq 1,2}(\mathbb{P}_{i\neq 1}\beta)) & -\frac{1}{2}(1-\pi_1)\gamma_{i\neq 1}^2(\mathbb{P}_{i\neq 1,2}\beta_2 - \mathbb{P}_{i\neq 1,2}(\mathbb{P}_{i\neq 1}\beta)) \end{pmatrix},$$

*and*

$$C = \pi_1\gamma_1^3 \begin{pmatrix} 0 & 0 \\ 0 & \begin{pmatrix} 0 & -\mathbb{P}_{1,1}\mathbb{P}_{1,2} & \mathbb{P}_{1,1}\mathbb{P}_{1,2} \\ -\mathbb{P}_{1,1}\mathbb{P}_{1,2} & \mathbb{P}_{1,2} - 2\mathbb{P}_{1,2}^2 & 0 \\ \mathbb{P}_{1,1}\mathbb{P}_{1,2} & 0 & -(\mathbb{P}_{1,2} - 2\mathbb{P}_{1,2}^2) \end{pmatrix} \end{pmatrix}$$
$$+ (1-\pi_1)\gamma_{i\neq 1}^3 \begin{pmatrix} 0 & 0 \\ 0 & \begin{pmatrix} 0 & -\mathbb{P}_{i\neq 1,1}\mathbb{P}_{i\neq 1,2} & \mathbb{P}_{i\neq 1,1}\mathbb{P}_{i\neq 1,2} \\ -\mathbb{P}_{i\neq 1,1}\mathbb{P}_{i\neq 1,2} & \mathbb{P}_{i\neq 1,2} - 2\mathbb{P}_{i\neq 1,2}^2 & 0 \\ \mathbb{P}_{i\neq 1,1}\mathbb{P}_{i\neq 1,2} & 0 & -(\mathbb{P}_{i\neq 1,2} - 2\mathbb{P}_{i\neq 1,2}^2) \end{pmatrix} \end{pmatrix}$$

*Proof.* We just need to compute $B(q_2 y_2)$ and substitute $y_2$ as the solution of the range equation. Similar to previous computation, we divide into the cases of $1 \leq k \leq 3$ and $4 \leq k \leq 6$.

**The cases of $1 \leq k \leq 3$.**

1. The contribution from $l = 1$. Using the matrix form defined in Eq. (206), we get

$$B_1^{(1)}(q_2y_2) = -\frac{y_2}{\sqrt{2}}(0,0,0,0,2\pi_1\gamma_1\mathbb{P}_{1,2}, -2\pi_1\gamma_1\mathbb{P}_{1,2})$$

$$B_2^{(1)}(q_2y_2) = -\frac{y_2}{\sqrt{2}}(0,0,0,0,(1-\pi_1)\gamma_{i\neq1}\mathbb{P}_{i\neq1,2}, -(1-\pi_1)\gamma_{i\neq1}\mathbb{P}_{i\neq1,2})$$

$$B_3^{(1)}(q_2y_2) = B_2^{(1)}(q_2y_2)$$

2. The contributions from $l = 2, 3, 4, 6$ vanish.

3. The contribution from $l = 5$. Using the matrix defined in Eq. (212), we get

$$c_3\begin{pmatrix} 0 \\ 1 \\ -1 \end{pmatrix} = -\pi_1\gamma_1^2(0, \mathbb{P}_{1,2}\beta_2 - \mathbb{P}_{1,2}(\mathbb{P}_1\beta), -\mathbb{P}_{1,2}\beta_2 - \mathbb{P}_{1,2}(\mathbb{P}_1\beta)),$$

which implies

$$B_1^{(5)}(q_2y_2) = -\frac{y_2}{\sqrt{2}}\pi_1\gamma_1^2(0,0,0,0,\mathbb{P}_{1,2}\beta_2 - \mathbb{P}_{1,2}(\mathbb{P}_1\beta), -(\mathbb{P}_{1,2}\beta_2 - \mathbb{P}_{1,2}(\mathbb{P}_1\beta)))$$

Similarly,

$$B_2^{(5)}(q_2y_2) = B_3^{(5)}(q_2y_2) = -\frac{y_2}{\sqrt{2}}\frac{1}{2}(1-\pi_1)\gamma_{i\neq1}^2(0,0,0,0,\mathbb{P}_{i\neq1,2}\beta_2 - \mathbb{P}_{i\neq1,2}(\mathbb{P}_{i\neq1}\beta), -\mathbb{P}_{i\neq1,2}\beta_2 - \mathbb{P}_{i\neq1,2}(\mathbb{P}_{i\neq1}\beta))$$

**The cases of $4 \leq k \leq 6$.**

1. The contribution from $l = 1$. By direct computation,

$$B_4^{(1)}(q_2y_2) = (0,0,0,0,0,0)$$

$$B_5^{(1)}(q_2y_2) = -\frac{y_2}{\sqrt{2}}(\pi_1\gamma_1\mathbb{P}_{1,2}, \frac{1}{2}(1-\pi_1)\gamma_{i\neq1}\mathbb{P}_{i\neq1,2}, \frac{1}{2}(1-\pi_1)\gamma_{i\neq1}\mathbb{P}_{i\neq1,2}, 0,0,0)$$

$$B_6^{(1)}(q_2y_2) = -B_5^{(1)}$$

2. The contribution from $l = 2$. This term vanishes.

3. The contribution from $l = 3$. This term makes the same contribution as the first term.

4. The contribution from $l = 4$. By definition,

$$-\sum_i \pi_i(\mathrm{d}(\mathbb{A}_iM)\mathrm{d}\,\mathrm{Var}(\mathbb{P}_i))^\mathsf{T}\mathbb{A}_i\gamma = -\pi_1\gamma_1(\mathbb{A}_1\mathrm{d}M\mathrm{d}\,\mathrm{Var}(\mathbb{P}_1))^\mathsf{T} - (1-\pi_1)\gamma_{i\neq1}(\mathbb{A}_{i\neq1}\mathrm{d}M\mathrm{d}\,\mathrm{Var}(\mathbb{P}_{i\neq1}))^\mathsf{T}$$

Take the first term as an example, the cross term in $(\mathbb{A}_1\mathrm{d}M\mathrm{d}\,\mathrm{Var}(\mathbb{P}_1))^\mathsf{T}$ is

$$\gamma_1\mathrm{d}\gamma_1\left(\mathrm{diag}(\beta^\mathsf{T}\,\mathrm{Var}(\mathbb{P}_1)) - \mathbb{P}_1^\mathsf{T}\beta^\mathsf{T}\,\mathrm{Var}(\mathbb{P}_1) - \mathrm{Var}(\mathbb{P}_1)\beta\mathbb{P}_1\right)\mathrm{d}\beta$$
$$+ \gamma_1\mathrm{d}\gamma_1\left(\mathrm{diag}(\mathrm{d}\beta^\mathsf{T}\,\mathrm{Var}(\mathbb{P}_1)) - \mathbb{P}_1^\mathsf{T}\mathrm{d}\beta^\mathsf{T}\,\mathrm{Var}(\mathbb{P}_1) - \mathrm{Var}(\mathbb{P}_1)\mathrm{d}\beta\mathbb{P}_1\right)\beta$$

Write in entry form, for $4 \leq k \leq 6$, we get

$$\gamma_1\mathrm{d}\gamma_1\left((\beta^\mathsf{T}\,\mathrm{Var}(\mathbb{P}_1))_k\mathrm{d}\beta_k - \mathbb{P}_{1,k}\beta^\mathsf{T}\,\mathrm{Var}(\mathbb{P}_1)\mathrm{d}\beta - (\mathrm{Var}(\mathbb{P}_1)\beta)_k\mathbb{P}_1\mathrm{d}\beta\right)$$
$$+ \gamma_1\mathrm{d}\gamma_1\left(\mathrm{d}\beta^\mathsf{T}\,\mathrm{Var}(\mathbb{P}_1)_k\beta_k - \mathbb{P}_{1,k}\mathrm{d}\beta^\mathsf{T}\,\mathrm{Var}(\mathbb{P}_1)\beta - \mathrm{Var}(\mathbb{P}_1)_k\mathrm{d}\beta\mathbb{P}_1\beta\right)$$

Writing into the matrix form, we get

$$
\begin{pmatrix} 0 & \frac{1}{2}(\beta^\mathsf{T}\,\mathrm{Var}(\mathbb{P}_1))_k E_{1,k} \\ \frac{1}{2}(\beta^\mathsf{T}\,\mathrm{Var}(\mathbb{P}_1))_k E_{1,k}^\mathsf{T} & 0 \end{pmatrix} + \begin{pmatrix} 0 & \frac{1}{2}\beta_k e_k\,\mathrm{Var}(\mathbb{P}_1)_k \\ \frac{1}{2}\beta_k\,\mathrm{Var}(\mathbb{P}_1)_k^\mathsf{T} e_k^\mathsf{T} & 0 \end{pmatrix}
$$

$$
- \mathbb{P}_{1,k}\begin{pmatrix} 0 & e_1\beta^\mathsf{T}\,\mathrm{Var}(\mathbb{P}_1) \\ \mathrm{Var}(\mathbb{P}_1)\beta e_1^\mathsf{T} & 0 \end{pmatrix} - \begin{pmatrix} 0 & \frac{1}{2}(\mathrm{Var}(\mathbb{P}_1)\beta)_k e_1 \mathbb{P}_1 \\ \frac{1}{2}(\mathrm{Var}(\mathbb{P}_1)\beta)_k \mathbb{P}_1^\mathsf{T} e_1^\mathsf{T} & 0 \end{pmatrix}
$$

$$
- \begin{pmatrix} 0 & \frac{1}{2}(\mathbb{P}_1\beta)e_1\,\mathrm{Var}(\mathbb{P}_1)_k \\ \frac{1}{2}(\mathbb{P}_1\beta)\,\mathrm{Var}(\mathbb{P}_1)_k^\mathsf{T} e_1^\mathsf{T} & 0 \end{pmatrix}
$$

Multiplying the matrix form by $(0,0,0,0,1,-1)$ on the right, we get

$$
B_4(q_2 y_2) = 0
$$
$$
B_5(q_2 y_2) = -\frac{y_2}{\sqrt{2}}\pi_1\gamma_1^2\big(\frac{1}{2}(\beta^\mathsf{T}\,\mathrm{Var}(\mathbb{P}_1))_2 + \frac{1}{2}\beta_2 \mathbb{P}_{1,2} - \frac{1}{2}(\mathbb{P}_1\beta)\mathbb{P}_{1,2}, 0, \ldots, 0\big)
$$
$$
= -\frac{y_2}{\sqrt{2}}\pi_1\gamma_1^2(\beta_2 \mathbb{P}_{1,2} - (\mathbb{P}_1\beta)\mathbb{P}_{1,2}, 0, \ldots, 0)
$$
$$
B_6(q_2 y_2) = -B_5(q_2 y_2)
$$

Similarly, the cross term in $-(1-\pi_1)\gamma_{i\neq1}(\mathbb{A}_{i\neq1}\mathrm{d}M\mathrm{d}\,\mathrm{Var}(\mathbb{P}_{i\neq1}))^\mathsf{T}$ contributes to

$$
B_4(q_2 y_2) = 0
$$
$$
B_5(q_2 y_2) = -\frac{y_2}{\sqrt{2}}\frac{1}{2}(1-\pi_1)\gamma_{i\neq1}^2(0, \beta_2\mathbb{P}_{i\neq1,2} - (\mathbb{P}_{i\neq1}\beta)\mathbb{P}_{i\neq1,2}, \beta_2\mathbb{P}_{i\neq1,2} - (\mathbb{P}_{i\neq1}\beta)\mathbb{P}_{i\neq1,2}, 0, \ldots, 0)
$$
$$
B_6(q_2 y_2) = -B_5(q_2 y_2)
$$

Finally, we compute the contribution from quadratic form. Take the first term as an example,

$$
\gamma_1^2\big(\mathrm{diag}(\mathrm{d}\beta^\mathsf{T}\,\mathrm{Var}(\mathbb{P}_1)) - \mathbb{P}_1^\mathsf{T}\,\mathrm{Var}\,\mathbb{P}_1 - \mathrm{Var}(\mathbb{P}_1)\mathrm{d}\beta\mathbb{P}_1\big)\,\mathrm{d}\beta
$$

Writing into the matrix form, we get

$$
B_k = \begin{pmatrix} 0 & 0 \\ 0 & \frac{1}{2}\big(e_k\,\mathrm{Var}(\mathbb{P}_1)_k + \mathrm{Var}(\mathbb{P}_1)_k^\mathsf{T} e_k^\mathsf{T}\big) \end{pmatrix} - \begin{pmatrix} 0 & 0 \\ 0 & \mathbb{P}_{1,k}\,\mathrm{Var}(\mathbb{P}_1) \end{pmatrix} - \begin{pmatrix} 0 & 0 \\ 0 & \frac{1}{2}\big(\mathbb{P}_1^\mathsf{T}\,\mathrm{Var}(\mathbb{P}_1)_k + \mathrm{Var}(\mathbb{P}_1)_k^\mathsf{T}\mathbb{P}_1\big) \end{pmatrix}
$$

By direct computation, we get the contribution to the perturbed hessian is

$$
-\frac{y_2}{\sqrt{2}}\pi_1\gamma_1^3\begin{pmatrix} 0 & 0 \\ 0 & \begin{pmatrix} 0 & -\mathbb{P}_{1,1}\mathbb{P}_{1,2} & \mathbb{P}_{1,1}\mathbb{P}_{1,2} \\ -\mathbb{P}_{1,1}\mathbb{P}_{1,2} & \mathbb{P}_{1,2} - 2\mathbb{P}_{1,2}^2 & 0 \\ \mathbb{P}_{1,1}\mathbb{P}_{1,2} & 0 & -(\mathbb{P}_{1,2} - 2\mathbb{P}_{1,2}^2) \end{pmatrix} \end{pmatrix}
$$

$$
-\frac{y_2}{\sqrt{2}}(1-\pi_1)\gamma_{i\neq1}^3\begin{pmatrix} 0 & 0 \\ 0 & \begin{pmatrix} 0 & -\mathbb{P}_{i\neq1,1}\mathbb{P}_{i\neq1,2} & \mathbb{P}_{i\neq1,1}\mathbb{P}_{i\neq1,2} \\ -\mathbb{P}_{i\neq1,1}\mathbb{P}_{i\neq1,2} & \mathbb{P}_{i\neq1,2} - 2\mathbb{P}_{i\neq1,2}^2 & 0 \\ \mathbb{P}_{i\neq1,1}\mathbb{P}_{i\neq1,2} & 0 & -(\mathbb{P}_{i\neq1,2} - 2\mathbb{P}_{i\neq1,2}^2) \end{pmatrix} \end{pmatrix}
$$

We obtain the result by summing all non-zero terms. $\qquad\square$

**Lemma C.16** (Vanishing of the first-order perturbation on the kernel). *On the symmetric rank-one manifold ($\gamma_2 = \gamma_3$ and $\beta_2 = \beta_3$), we have $Q_K^\mathsf{T} H_1 Q_K = 0$.*

*Proof.* We write any $z \in \mathbb{R}^6$ as $z = (z_\gamma, z_\beta)$ with $z_\gamma, z_\beta \in \mathbb{R}^3$. Let

$$
s := (0, 1, -1)^\mathsf{T}, \qquad u := (1, 1, 1)^\mathsf{T}.
$$

Then $k_{1,\gamma} = \frac{1}{\sqrt{2}}s$, $k_{1,\beta} = 0$; $k_{2,\gamma} = 0$, $k_{2,\beta} = \frac{1}{\sqrt{3}}u$; and $k_{3,\gamma} \propto -\gamma$, $k_{3,\beta} \propto \beta$.

**Step 1: the off-diagonal block with respect to $B$**    For any $x = (x_\gamma, x_\beta)$ and $y = (y_\gamma, y_\beta)$,

$$x^\mathsf{T} \begin{pmatrix} 0 & B \\ B^\mathsf{T} & 0 \end{pmatrix} y = x_\gamma^\mathsf{T} B y_\beta + x_\beta^\mathsf{T} B^\mathsf{T} y_\gamma.$$

From Eq. (171), $B$ has the structural identities

$$B_{\cdot,1} = 0, \qquad B_{\cdot,3} = -B_{\cdot,2}, \qquad B_{2,\cdot} = B_{3,\cdot}.$$

Hence

$$Bu = B(e_1 + e_2 + e_3) = 0, \qquad B\beta = \beta_1 B_{\cdot,1} + \beta_2 (B_{\cdot,2} + B_{\cdot,3}) = 0 \;\; (\text{since } \beta_2 = \beta_3),$$

and

$$s^\mathsf{T} B = (0, 1, -1) B = B_{2,\cdot} - B_{3,\cdot} = 0.$$

Combining these, every matrix element $k_i^\mathsf{T} \begin{pmatrix} 0 & B \\ B^\mathsf{T} & 0 \end{pmatrix} k_j$ vanishes: either a factor $k_{1,\beta} = 0$ appears, or a factor $s^\mathsf{T} B = 0$ appears, or a factor $Bu = 0$ / $B\beta = 0$ appears.

**Step 2: the lower-right block $C$.**    Here

$$x^\mathsf{T} \begin{pmatrix} 0 & 0 \\ 0 & C \end{pmatrix} y = x_\beta^\mathsf{T} C y_\beta.$$

Each summand of $C$ in Eq. (172) has the pattern

$$C^{(\ell)} = \begin{pmatrix} 0 & -p_\ell & p_\ell \\ -p_\ell & a_\ell & 0 \\ p_\ell & 0 & -a_\ell \end{pmatrix} \quad \text{for some } (p_\ell, a_\ell), \qquad C = \sum_\ell \omega_\ell C^{(\ell)}.$$

A direct expansion gives, for any $x, y \in \mathbb{R}^3$,

$$x^\mathsf{T} C^{(\ell)} y = p_\ell (x_3 - x_2) y_1 + p_\ell x_1 (y_3 - y_2) + a_\ell (x_2 y_2 - x_3 y_3).$$

Therefore, if $x_2 = x_3$ and $y_2 = y_3$, then $x^\mathsf{T} C^{(\ell)} y = 0$ and hence $x^\mathsf{T} C y = 0$. On the symmetric manifold we have $(k_{2,\beta})_2 = (k_{2,\beta})_3$ and $(k_{3,\beta})_2 = (k_{3,\beta})_3$ (since $\beta_2 = \beta_3$), while $k_{1,\beta} = 0$. Thus, $k_i^\mathsf{T} \begin{pmatrix} 0 & 0 \\ 0 & C \end{pmatrix} k_j = 0$ for all $i, j$.

**Step 3: the $J_1$ term.**    By Proposition 7.4,

$$J_1 = \widetilde{J}_1 + \begin{pmatrix} 0 & A \\ A^\mathsf{T} & 0 \end{pmatrix},$$

where $A$ in Eq. (109) satisfies the same cancellation identities as $B$:

$$A_{\cdot,1} = 0, \qquad A_{\cdot,3} = -A_{\cdot,2}, \qquad A_{2,\cdot} = A_{3,\cdot}.$$

Hence $Au = 0$, $A\beta = 0$ (since $\beta_2 = \beta_3$), and $s^\mathsf{T} A = 0$. Repeating Step 1 with $B$ replaced by $A$, we get

$$Q_K^\mathsf{T} \begin{pmatrix} 0 & A \\ A^\mathsf{T} & 0 \end{pmatrix} Q_K = 0.$$

For the remaining part $\widetilde{J}_1$, the explicit computation shows that its action on the kernel directions has no kernel component, i.e. $Q_K^\mathsf{T} \widetilde{J}_1 Q_K = 0$. Therefore $Q_K^\mathsf{T} J_1 Q_K = 0$.

**Conclusion.**    Combining Step 1–3 yields $Q_K^\mathsf{T} H_1 Q_K = 0$.    □

C.3.9. FAST TRANSVERSE INSTABILITY: $\Theta(\delta)$ EIGENVALUE

**Lemma C.17** (A transverse eigenvalue of order $\Theta(\delta)$). *Let* $c = \frac{\lambda\gamma_{i\neq 1}+(1-\lambda)(\pi_1\gamma_1+(1-\pi_1)\gamma_{i\neq 1})}{\pi_1\gamma_1^2\mathbb{P}_{1,2}+(1-\pi_1)\gamma_{i\neq 1}^2\mathbb{P}_{i\neq 1,2}}$ *and assume that*

$$c(1-\pi_1)\gamma_{i\neq 1}\mathbb{P}_{i\neq 1,2} - (\lambda + (1-\pi_1)(1-\lambda)) \neq 0. \tag{230}$$

*At the perturbed point,* $\partial\mathcal{L}/\partial\Phi = \mathcal{O}(\delta^2)$ *while* $\partial\mathcal{L}/\partial M = \Theta(\delta)$. *Consequently, linearizing the full dynamics (Eq.* (174)*) yields a transverse positive eigenvalue of size* $\Theta(\delta)$.

*Proof.* Since $\mathrm{d}\frac{\partial\mathcal{L}}{\partial\Phi} = 0$ and $\partial_\delta\frac{\partial\mathcal{L}}{\partial\Phi} = 0$, we get $\frac{\partial\mathcal{L}}{\partial\Phi} = \mathcal{O}(\delta^2)$ after perturbation.

Next we compute perturbed $\frac{\partial\mathcal{L}}{\partial M}$. By Lemma C.5, we get the new term is

$$c\pi_1\gamma_1 \begin{pmatrix} 1 \\ 0 \\ 0 \end{pmatrix} (0, \mathbb{P}_{1,2}, -\mathbb{P}_{1,2}) + c(1-\pi_1)\gamma_{i\neq 1} \begin{pmatrix} 0 \\ \frac{1}{2} \\ \frac{1}{2} \end{pmatrix} (0, \mathbb{P}_{i\neq 1,2}, -\mathbb{P}_{i\neq 1,2})$$

where $c = \frac{\lambda\gamma_{i\neq 1}+(1-\lambda)(\pi_1\gamma_1+(1-\pi_1)\gamma_{i\neq 1})}{\pi_1\gamma_1^2\mathbb{P}_{1,2}+(1-\pi_1)\gamma_{i\neq 1}^2\mathbb{P}_{i\neq 1,2}}\delta$.

However, from Eq. (186), we find that

$$\frac{\partial}{\partial\delta}\frac{\partial\mathcal{L}}{\partial M} = -\begin{pmatrix} 0 \\ \frac{1}{2} \\ \frac{1}{2} \end{pmatrix} (0, \lambda, -\lambda) - \begin{pmatrix} \pi_1 \\ \frac{1}{2}(1-\pi_1) \\ \frac{1}{2}(1-\pi_1) \end{pmatrix} (0, 1-\lambda, -(1-\lambda)).$$

The two terms cannot cancel each other out by our assumption (A parameter that does not meet the condition is a zero test set), thus resulting in an $\Theta(\delta)$ term. $\square$

# D. Detailed Experiment Setup

## D.1. Detailed Synthetic Experiment Setup

**Dataset** To better induce an exponential decay in the stationary distribution, and to more clearly illustrate the phase transition, we adopt an exponentially decaying form

$$\pi_0 = \left(1/2,\ 1/4,\ \ldots,\ 1/2^d\right), \qquad \pi = \pi_0/\|\pi_0\|.$$

Following (Makkuva et al., 2025), diagonal dominant transition matrices are unfavorable local minima during optimization, we set $\lambda = 0.8$ to ensure diagonal dominance. For a fixed sequence length of 20, we sample 100,000 sequences $\{X_i\}_{i=1}^{100,000}$ from the resulting Markov chain, and use the last token $X_i[-1]$ as the training label. By the Markov property, this label is completely determined by the second-to-last token $X_i[-2]$. Accordingly, we group both the training and test sets by the value of $X_i[-2]$, denoting the group indexed by state $k$ as $S_k$.

**Model** We follow exactly the model specification in Def. 3.2, with embedding dimension $m = 256$. Since our theoretical analysis is derived under small initialization, we adopt the initialization scheme of (Zhang et al., 2024b; 2025b), initializing each weight independently as $\mathcal{N}(0, 1/m^2)$.

**Training** We train the model using the Adam optimizer with a fixed learning rate of $1.5 \times 10^{-4}$ and do not use any learning-rate scheduler.

## D.2. Analysis Tools

**Condensation Heatmap** To quantify parameter condensation, we compute the pairwise cosine similarity between the input-weight vectors of neurons in the weight matrix $W$. Specifically, for the $i$-th and $j$-th neurons, we define

$$C(i, j) = \frac{W[i, :] \cdot W[j, :]}{\|W[i, :]\|_2 \|W[j, :]\|_2}.$$

For clearer visualization, we permute the rows and columns of the similarity matrix $C$ and display the reordered matrix in Fig. 2(A).

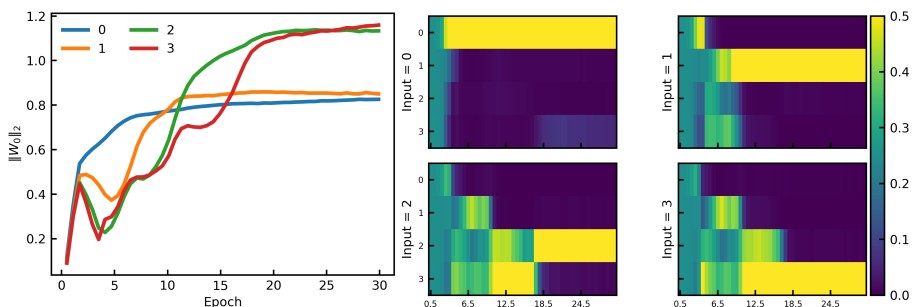

*Figure 4.* The experiment result with residual connections, similarly with the Fig. 2(D,C) . The multi-stage focus-dilution structure remains clearly visible, showing that residual connections do not qualitatively alter the mechanism studied in this paper.

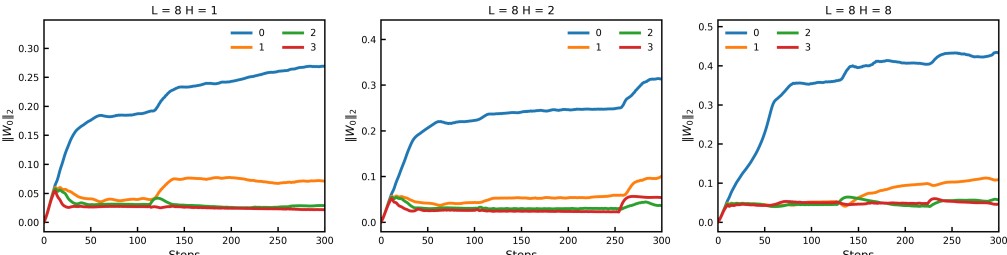

*Figure 5.* The embedding norm dynamics on multi-layer and multi-head transformer. The 8L1H model preserves the distinct norm growth-and-retraction phenomenon remarkably well but the phenomenon becomes less visually pronounced with head num increase.

**Embedding Visualization**    Let $W_{0,t}$ denote the embedding parameters at training epoch $t$ for $t = 0, \ldots, T$. We form the collection of embedding snapshots $\{W_{0,0}, \ldots, W_{0,T}\}$ and apply principal component analysis (PCA) to obtain the leading eigen-directions $\hat{e}_1$ and $\hat{e}_2$. We then project the embedding vectors onto $\hat{e}_1$ and $\hat{e}_2$ to produce the two-dimensional visualization shown in Fig. 2(B).

### D.3. Additional Experiments

In this section, we present additional experimental results for attention mechanisms with residual connections and Transformers with multiple attention layers. The experimental setup is identical to that described in the preceding sections. For the model with residual connections, we modify the architecture as follows:

$$f_\theta(X) = \left( \text{Attn}(E_X W_0) + E_X W_0 \right) W_1. \tag{231}$$

For multilayer and multi-head attention models, we further stack multiple attention layers on top of the architecture with residual connections. The corresponding experimental results are reported in Fig. 4 and Fig. 5.

