# OpenReview forum: "Focus and Dilution: The Multi-stage Learning Process of Attention"
_ICML.cc/2026/Conference — ICML 2026 spotlight_

### Official Review · Reviewer_rvXr · 2026-02-17

**Soundness:** 2
**Presentation:** 3
**Significance:** 3
**Originality:** 3
**Overall Recommendation:** 5
**Confidence:** 3

**Summary:**

This work studies the training dynamics of attention layers in one-layer transformer models. Developing a minimal testbed, the paper assumes Markovian data with one high frequency token. Through analyzing the linearization of the gradient flow at critical points, the paper theoretically predicts a cyclic focus-dilution dynamic in the attention layer. The paper then validates this finding sampling and training on a 4-token dataset. Training the same model on small natural language datasets, the paper provides some initial evidence by which this dynamic could generalize to the early training stages of more realistic settings.

**Compliance With Llm Reviewing Policy:**

Affirmed.

**Final Justification:**

This work establishes a novel and potentially very impactful step towards explaining the training dynamics of Transformer models. My concerns regarding the scope of the experiments have been addressed in the rebuttal, though more efforts will be required to study the proposed focus-dilution attention learning in more realistic settings in future work. My other concerns have also been addressed adequately, especially regarding framing and discussion of the results.

**Therefore, I increase my overall assessment to 5 and recommend this for publication at ICML.**

**Key Questions For Authors:**

**Questions:**

(Q1) Would it be feasible to study the proposed phenomenon not just training on language data, but also through architectures that are closer to the “real world”? To avoid expensive training there, one could follow [1]’s approach and analyze existing checkpoints from a Pythia model, though those checkpoints might not be “early” and fine-grained enough to actually observe focus-dilution.

(Q2) Will you release the code and data implementing your experiments?

(Q3) What questions you point at in your Abstract and Introduction remain unanswered by your work? How could this paper function as a basis for future work attempting to explain attention learning dynamics in the Transformer?

**Questions to help the Reviewer’s understanding:**

(Q4) Could you please elaborate on the following result described in lines 378-380:

> Fig. 2(D) further illustrates the later stages of the training process, revealing a distinct periodicity in the embedding norms.

I am only able to observe said periodicity for token 3, but might have missed something here?

(Q5) Could you also please elaborate on

> According to our theory, the attention mechanism evolves such that high-frequency tokens are gradually focused by the remaining tokens. This phenomenon is clearly visualized in Fig. 2(C).

(Lines 323 to 326)?

**Limitations:**

A Limitations section is needed but missing (see weaknesses). Impact: Yes.

**Strengths And Weaknesses:**

**Strengths:**

(S1) The paper successfully isolates the training dynamics into a minimal, yet interesting setting.

(S2) The mathematical analysis of this model looks rigorous and convincingly predicts the cyclic focus-dilution training dynamic (note: I come from a more applied background, so, despite an extensive mathematical education, it is likely some details might have been missed. Proofs in the appendix have also not been checked.)

(S3) In the testbed setting, the empirical validation of the dynamic supports the theoretical analysis

(S4) The paper is well written and presents its findings in a clear and digestible manner

(S5) The results of training on natural language scenarios suggests the phenomenon might generalize to realistic settings (note W1 though)

**Weaknesses:**

(W1) While training on natural language is attempted, I’d be cautious to claim these as “Real-world Experiments”. For example, previous work (e.g. [1] or [2]), has established that layer dynamics play an important role in generalization in Transformer models. By the nature of the setting studied, this is not captured in this work. Given that this work is focused on theoretical analysis, discussing this in a limitations section would be an acceptable minimal remedy for this.

(W2) The current submission lacks the code for reproducing the experiment part of the work. In addition. the experiment settings are underspecified in the Appendix. E.g., for the Adam optimization, only the learning rate is reported. For the natural language settings, no additional information about the data preprocessing etc. is provided. Providing a well-documented implementation of the reported experiments is strictly necessary for improving Reviewer’s current overall impression of the work.

(W3) Given the scope of scenarios studied, this work is missing a Limitations and perhaps also a Future Work section. See, e.g., W1. What questions remain unanswered by your work? How could your work function as a basis for future work attempting to explain attention learning dynamics in the Transformer?


**Minor weaknesses:**

(MW1) I disagree with the decision to relegate the entire Related Work section to the appendix. While the introduction discusses some background, this work seems to follow contributions from an extensive body of previous work that should be recognized in the main paper.

(MW2) The presentation and discussion of the empirical results could be improved/extended. In particular:

(MW2.1) While I share the qualitative interpretation in paragraph "Stage I: Initial Condensation" (from line 306), it would be better to actually calculate the rank of the matrix and put that into a Figure over time.

(MW2.2) Similarly, for "Stage II: Growth of Attention" a trajectory over the parameter changes would be nice.

(MW3) Only a single model instance is analyzed in the experiments. Is it feasible/sensible to repeat the experiments of multiple seeds?

**Suggestions/typos:**

(SG1) The font size in parts of Figure 1 and Figure 2, as well as the tokens on the axes of Figure 3 could be larger. Currently, one has to zoom in a lot to comfortably read all of the text there.

(SG2) In line 429, “As illustrated in the figure” should be a reference to Figure 3, i.e. “As illustrated in Figure 3”.

(SG3) In line 047, should it be “technical condition[s]”?

---

> ### Author Rebuttal · Authors · 2026-03-27
>
> We sincerely thank for the constructive feedback. We have carefully addressed your questions below:
> ### W1
> We agree that the term "Real-world Experiments" should be used cautiously. To bridge the gap toward more realistic settings, we have conducted additional numerical experiments on more complex architectures, such as an 8-layer Transformer. The results are provided in https://ibb.co/9HVCkgP7.
>
> Furthermore, as suggested, we will explicitly state in the Limitations section that our current theoretical guarantees are constrained to the 1-layer architecture. We are also very eager to discuss the references [1] and [2] you mentioned regarding layer dynamics and generalization; however, it appears the specific citations were inadvertently omitted from the review text. If the reviewer could provide these references during the discussion phase, we would gladly incorporate them into our revision.
> ### W2 & Q2
> We sincerely apologize for this oversight. Providing reproducible experimental details is indeed crucial. We will include a comprehensive experimental setup section in the revised appendix, and we will also release the fully documented code to ensure reproducibility.
> Specifically, our experimental settings are as follows:
> - Sequence Length: 20
> - Batch Size: 2000
> - Dimensions: $m$ = 256
> - Data Generation: Generated via a Markov process. We sample the first token based on the stationary distribution. Subsequent tokens are sampled according to the transition probabilities.
> - Initialization: $\epsilon$ = 2e-4
> - Random Seed: 3 runs and the phenomenon is robust.
> Importantly, our ablation studies reveal that the observed focus-dilution phenomenon is robust and not highly sensitive to variations in sequence length, $m$, or whether the loss is calculated specifically on the last token / all tokens.
> ### W3 & Q3
> We thank the reviewer for pointing this out. We will add a dedicated Limitations and Future Work section in the revised manuscript. In particular, we will highlight two key unanswered questions: (1) extending the analysis to include multi-layer Transformers with residual connections, and (2) investigating the specific impact of layerNorm on learning dynamics.
> ### MW1
> We agree with the reviewer. Relegating the related work to the appendix was a difficult decision forced strictly by the initial page limit. In the revised scipt, we will properly integrate the key related works back into the main text right after the Introduction.
> ### MW2.1
> We are happy to accommodate this request. We will calculate the matrix rank and present its evolution over time in the revised appendix. We also note that similar rank behaviors have been discussed in detail in arXiv:2510.06954.
> ### MW2.2
> We agree that tracking parameter trajectories would be insightful. However, since the attention weights form a high-dimensional matrix, directly observing and visualizing the internal parameter changes over time is challenging. As an effective alternative, we have plotted the trajectory of the parameter norm (https://ibb.co/1twjGK70). This parameter norm curve clearly corroborates the theoretical behaviors we described for Stage I~III.
> ### MW3
> We find the results to be stable for different random seeds; please see W2.
>
> ### Suggestions/typos:
> We thank the reviewer's advice and will make these revisions in version.
>
> ### Q1
> Your intuition is exactly correct, and we deeply share this thought. As you pointed out, the focus-dilution cycle happens very early and rapidly during training, and checkpoints that are simply not saved or released by open-source projects.
> ### Q4
> We thank the reviewer for this careful observation. The reason the periodicity is most prominent for token 3 is rooted in the sequential learning dynamics of the network. Under our setting, the network learns the tokens in a strict frequency-based order: token 0 → 1 → 2 → 3. To successfully learn each token, the network must traverse one complete "focus-dilution" cycle.
>
> A clear way to visualize this is by tracking the embedding norm trajectory in **Fig. 2(D2)**. Because of the sequential learning order, token 0 - 3 experiences 0 - 3 retractions respectively. This cumulative effect is why the periodicity is most pronounced and easily observable for token 3. To provide a finer-grained view of this process, **Fig. 2(B1-B3)** specifically illustrates the detailed embedding changes during the network's initial phase of learning token 0.
> ### Q5
> We apologize if the description in lines 323-326 was too brief. This phenomenon consists of two distinct continuous phases: The "Focus" Phase: At the very beginning of training, the attention weights of all tokens (including the remaining lower-frequency tokens) gradually concentrate on the most frequent token (token 0). The "Dilution" Phase: Subsequently, all tokens except token 0 itself gradually lose this intense focus on token 0. Their attention weights retract and enter a near-uniform distribution state.

---

> > ### Author Rebuttal · Reviewer_rvXr · 2026-04-01
> >
> > I seem to have used the wrong button for my follow-up. Please find my further questions in the comment above.
> >
> > Edit: I paste my initial response below.
> >
> > Thank you for your response. I'll need some further clarifications as follows:
> >
> > (W1) Could you please elaborate on what you did in these experiments? The way you present them right now, I am not able to follow what you did, nor what this shows. Please also explain, how they address my concern.
> >
> > (W2) My concerns have been addressed sufficiently.
> >
> > (W3) Please feel free to use the space of the additional reply to elaborate on this. Optimally, show the new paragraphs you are putting into the paper.
> >
> > Minor weaknesses: Most of my key concerns there have been addressed sufficiently, for some I don't think further discussion would help.
> >
> > Here are the references that got lost copy pasting my review:
> >
> > [1] Skean, O., Arefin, M. R., Zhao, D., Patel, N., Naghiyev, J., LeCun, Y., & Shwartz-Ziv, R. (2025). Layer by Layer: Uncovering Hidden Representations in Language Models (arXiv:2502.02013). https://doi.org/10.48550/arXiv.2502.02013
> >
> > [2] Tenney, I., Das, D., & Pavlick, E. (2019). BERT Rediscovers the Classical NLP Pipeline. In A. Korhonen, D. Traum, & L. Màrquez (Eds.), Proceedings of the 57th Annual Meeting of the Association for Computational Linguistics (pp. 4593–4601). Association for Computational Linguistics. https://doi.org/10.18653/v1/P19-1452
> >
> > Revisiting this, I think this one is also relevant:
> >
> > [3] Lad, V., Gurnee, W., & Tegmark, M. (2024). The Remarkable Robustness of LLMs: Stages of Inference? (arXiv:2406.19384). https://doi.org/10.48550/arXiv.2406.19384

---

> > > ### Author Response · Authors · 2026-04-02
> > >
> > > We thank you again for your thorough review and for supporting our submission!
> > >
> > > ### Further Response to W1
> > >
> > > Thank you for raising this point. To address the concern that the phenomenon  may be affected by layer dynamics, we conducted additional experiments on the same controlled Markov data using deeper real transformers. Concretely, we only replace the 1L1H model by 8L1H, 8L2H, and 8L8H architectures. During training, we track the norm trajectory of each token’s embedding **(the same as Fig 2(D 2))**. We use embedding norms as a practical proxy for the multi-stage dynamics because their growth and retraction closely track the underlying change in attention.
> > >
> > > Interestingly, we observed the following: The 8L1H model preserves the distinct norm growth-and-retraction phenomenon remarkably well. We attribute this to the residual connections (ResNet structure), which help propagate the sequential learning dynamics across layers. As the number of heads increases, the phenomenon becomes less visually pronounced. This aligns with theoretical intuition: the rank-one condensation is tightly tied to the single-head structure. Multi-head attention intrinsically increases the rank of the attention matrix, meaning the dynamics must now be analyzed on a high-rank manifold rather than a simple rank-one trajectory.
> > >
> > > These experiments show that the multi-stage phenomenon predicted by our single-layer analysis is not merely an artifact of the 1L1H model. At the same time, we agree that they should not be interpreted as full real-world experiments, nor do they resolve the broader issues raised in [1–3], such as layer-wise representation dynamics and their role in generalization in realistic language models. We will add this point explicitly to the limitations discussion, as shown below.
> > >
> > > ### Further Response to W3
> > >
> > > We thank the reviewer for this suggestion. In the revision, we will add the following paragraph to the limitations discussion:
> > >
> > > > We provides a analytical framework for studying the coupled evolution of embeddings and attention, and identifies the focus–dilution cycle as a concrete mechanism. Experiments on synthetic data and small-scale natural language data provide preliminary support for this mechanism. Owing to the limitations of the theoretical framework, our analysis mainly focuses on the training dynamics of single-layer attention, and thus does not yet capture the effects of layer interactions or multi-head structure. Extending the analysis to multi-layer and multi-head architectures and realistic data, and studying how LayerNorm affects the dynamics, are important directions for future work.
> > >
> > > [1] Skean, O., Arefin, M. R., Zhao, D., Patel, N., Naghiyev, J., LeCun, Y., & Shwartz-Ziv, R. (2025). Layer by Layer: Uncovering Hidden Representations in Language Models (arXiv:2502.02013). https://doi.org/10.48550/arXiv.2502.02013
> > >
> > > [2] Tenney, I., Das, D., & Pavlick, E. (2019). BERT Rediscovers the Classical NLP Pipeline. In A. Korhonen, D. Traum, & L. Màrquez (Eds.), Proceedings of the 57th Annual Meeting of the Association for Computational Linguistics (pp. 4593–4601). Association for Computational Linguistics. https://doi.org/10.18653/v1/P19-1452
> > >
> > > [3] Lad, V., Gurnee, W., & Tegmark, M. (2024). The Remarkable Robustness of LLMs: Stages of Inference? (arXiv:2406.19384). https://doi.org/10.48550/arXiv.2406.19384

---

### Official Review · Reviewer_aCoV · 2026-03-08

**Soundness:** 4
**Presentation:** 3
**Significance:** 4
**Originality:** 4
**Overall Recommendation:** 6
**Confidence:** 4

**Summary:**

This paper studies the population gradient flow of a one-layer Transformer with trainable
embedding, key/query, and output weights, trained with cross-entropy on next-token
prediction over Markovian data. The stationary distribution $\pi$ has a single dominant
token and a group of approximately symmetric low-frequency ones. The
analysis reveals four dynamical stages, each governed by linearization near successive
critical points.

The first stage is the **condensation** of **Chen & Luo (2025)**: embedding and output
matrices collapse to a rank-one structure aligned with $\pi$, while key/query matrices
stay still. The second stage, **focus**, begins at a saddle point on the condensation direction
whose unique unstable eigendirection is in the attention subspace; $W_Q, W_K$ grow
exponentially and attention concentrates on the dominant token. In the third
stage, **dilution**, the growing attention feeds back into embeddings via a
mass-redistribution mechanism on a rank-one invariant manifold, forcing a divergence
between dominant and remaining token embeddings that progressively weakens the focus. The
fourth stage, **new direction emergence**, addresses a degenerate critical point where
learning stalls under perfect symmetry; a Lyapunov–Schmidt reduction shows that a small
asymmetry among low-frequency tokens creates a fast transverse instability that drives the
trajectory off the rank-one manifold, initiating the next cycle.

Experiments on synthetic Markov data, WikiText, and TinyStories corroborate the predicted
stages and cyclical dynamics.

**Compliance With Llm Reviewing Policy:**

Affirmed.

**Final Justification:**

This paper is a direct successor to **Chen & Luo (2025, oral at NeurIPS)**, which
introduced the condensation phase, dynamic separation between outer and attention
parameters, and the notion of rank collapse for key/query matrices. The present
work goes substantially further: it gives condensation and rank collapse an explicit
data-driven interpretation (the role of the stationary distribution $\pi$ ), and then finds
two entirely novel dynamical regimes: dilution via a mass-redistribution mechanism on
an invariant manifold, and emergence of new directions via Lyapunov–Schmidt
bifurcation analysis. The sequential tracking of four saddle-to-saddle transitions within a single theoretical framework,
culminating with the identification of a recurring cycle, represents an unusual density
of contributions for a single paper. I consider this work to be among the strongest
submissions I have reviewed in this cycle and recommend it for highlighted
presentation.

**Key Questions For Authors:**

## Questions

- In connection to the first weakness listed above: what happens in the synthetic
  experiments when randomness is increased (e.g. fewer training samples or a
  larger learning rate)? At what point does the saddle-to-saddle picture break down?
  Are some stages more robust than others?

- The model has no positional encodings. With positional encodings, the network
  could in principle learn to attend directly to position $s$ (the last token),
  bypassing the frequency-driven mechanism entirely. Would this shortcut eliminate
  the focus–dilution cycle, or would the cycle still appear during early training
  before positional attention is learned?

## Final Considerations

This paper is a direct successor to **Chen & Luo (2025, oral at NeurIPS)**, which
introduced the condensation phase, dynamic separation between outer and attention
parameters, and the notion of rank collapse for key/query matrices. The present
work goes substantially further: it gives condensation and rank collapse an explicit
data-driven interpretation (the role of the stationary distribution $\pi$ ), and then finds
two entirely novel dynamical regimes: dilution via a mass-redistribution mechanism on
an invariant manifold, and emergence of new directions via Lyapunov–Schmidt
bifurcation analysis. The sequential tracking of four saddle-to-saddle transitions within a single theoretical framework,
culminating with the identification of a recurring cycle, represents an unusual density
of contributions for a single paper. I consider this work to be among the strongest
submissions I have reviewed in this cycle and recommend it for highlighted
presentation.

**Limitations:**

yes

**Strengths And Weaknesses:**

## Strengths

- The mathematical analysis concretely connects properties of the data distribution
  and the architecture during optimization, by characterizing each successive saddle
  point in terms of the stationary distribution $\pi$, the embedding/output matrices,
  and the attention parameters.

- The paper traces the population gradient flow through four qualitatively distinct
  critical points, each governed by a different aspect of the data–architecture
  interaction. Analyzing even a single saddle-to-saddle transition is typically
  considered a significant contribution in this area. The sequential tracking across
  all four stages is especially notable: usually, once the trajectory exits the
  neighborhood where a local expansion is valid, precise control is lost. Here, the
  combination of rank-one condensation and spectral gaps at each saddle creates a
 single dominant unstable direction,  that pins down the
  entry point of the next stage and makes the full trajectory predictable.

- Each critical point is qualitatively different, not a repetition of the same
  pattern. The four stages exhibit a genuine feedback loop: embedding condensation
  shapes the attention landscape (Stage I $\to$ II), and in turn, attention growth
  feeds back into embedding redistribution (Stage II $\to$ III $\to$ IV). This
  bidirectional coupling is made analytically precise, which is rare in the
  literature.

- The Lyapunov–Schmidt reduction in Stage IV is an original methodological choice
  for the study of training dynamics. It is a  principled way to handle the
  degenerate critical point that arises from low-frequency token symmetry, and
  separates fast transverse instabilities ($\Theta(\delta)$) from slow
  tangential ones ($O(\delta^2)$).

- The dilution mechanism and subsequent emergence of new directions provide insight
  into how transformers can iteratively discover structure in data: the model first
  learns to attend to the most frequent tokens, then redistributes attention toward
  less frequent ones, in a process reminiscent of the simplicity bias observed in other architectures.

- The figures are detailed and informative. In particular, Figure 2 effectively
  decomposes the four stages through complementary views (loss curves, cosine
  similarity heatmaps, PCA trajectories, attention maps, and embedding norms),
  making the theoretical predictions visually accessible.

- Experiments on WikiText and TinyStories (Figure 3) show that the key qualitative
  predictions of the analysis (attention initially concentrating on high-frequency tokens such as
  whitespace and punctuation, followed by dilution and retraction of embedding norms) persist on real corpora with a one-layer Transformer, suggesting that the
  focus–dilution phenomenon is not an artifact of the synthetic setting chosen for the theoretical analysis.

## Weaknesses

- Only population gradient flow is analyzed, so the results are silent on sample
  complexity, the effect of mini-batch noise, and learning-rate discretization. It
  is unclear whether the sharp stage boundaries predicted by the continuous-time,
  infinite-data analysis would survive under stochastic gradient descent with finite
  samples. That said, the synthetic experiments do use finite samples and a discrete
  optimizer (Adam), and still exhibit the predicted four-stage pattern, which
  provides indirect evidence of robustness to these effects.

- The data model (first-order Markov chain) and architecture (one-layer Transformer
  without MLP, residual connections, or positional encodings) are quite restrictive.
  There is no formal guarantee that the four-stage structure extends to the
  multi-layer Transformers used in practice. That said, this level of simplification
  is in line with comparable theoretical works, and appears necessary for
  mathematical tractability.

- There is an inconsistency in token indexing between the theory and experiments:
  the theoretical sections use 1-indexing (token 1 is the most frequent, e.g.
  equation (17) gives $\lim A_i = e_1^\top$), while the experimental figures use
  0-indexing (token 0 is the most frequent, as seen from the $S_0$ loss curve in
  Figure 1). This needs to be fixed.

- The real-data experiments (Figure 3) confirm only a subset of the predicted
  phenomenology: primarily the attention focus/dilution pattern and embedding norm
  retraction as mentioned above; but do not clearly exhibit the full four-stage cycle or the
  emergence of new directions. This is partly mitigated by the fact that several
  accepted works in this area provide no real-data validation at all (e.g., Makkuva
  et al, (2025) uses only binary synthetic Markov data).

- No loss curves are shown for the real-data experiments,
  making it difficult to verify whether the characteristic multi-stage loss profile
  (sharp drop, plateau, secondary drop) observed in the synthetic setting also
  appears on real corpora.

### Minor points

- The block notation in equation (13) is confusing: the $2 \times 2$ block matrix
  multiplies a 4-entry vector, but the grouping of entries into sub-blocks is not
  indicated. Adding explicit parentheses around the outer and attention sub-vectors
  would clarify the intended block structure.

---

> ### Author Rebuttal · Authors · 2026-03-27
>
> We express our deepest gratitude to the reviewer for the outstanding assessment. We highly appreciate your careful reading and your exact recognition of the contributions we aimed to provide, particularly the discovery of the focus-dilution regime within a single theoretical framework. To further improve the manuscript based on your insightful feedback, we have addressed your concerns point-by-point below:
>
> ### W1
> We sincerely thank the reviewer for this insightful comment. As you correctly pointed out, our synthetic experiments already utilize finite samples and a discrete optimizer (Adam), which provides indirect evidence of robustness. To further address your concern, we have conducted additional experiments using SGD during the rebuttal period. As shown in (https://ibb.co/67679BLk), the predicted four-stage pattern remains highly robust and closely matches the theoretical predictions.
>
> ### W2
> We appreciate the reviewer's understanding that architectural simplification is necessary for mathematical tractability. However, to bridge the gap between our theory and practical applications, we have added new discussions and experiments evaluating 8-layer Transformers (also discussed in Reviewer eWfc W1). As illustrated in https://ibb.co/9HVCkgP7 similar to Fig. 2(D), the embedding norm confirms that multi-stage phenomenon still clearly emerges in these deeper architectures.
>
> ### W3 & Minor Points
> We apologize for these confusing presentation issues. The token-indexing discrepancy was indeed a typo in our manuscript, and we will correct the theoretical sections in the revised version to consistently use token 0 as the most frequent token. We will also revise Eq. (13) by adding explicit parentheses to clearly separate the outer and attention parameters, so that the block structure is unambiguous.
>
> ### W4
> We agree that the real-data experiments validate only part of the full phenomenology. The mismatch mainly comes from two factors: first, real-world NLP tasks cannot be perfectly modeled by a simple Markov chain; second, computational constraints prevent the use of extremely large batch sizes to approximate full-batch training, creating a mismatch compared to the clean Markov setting. As suggested, we will explicitly emphasize these mismatches and discuss them in detail in the Limitations section of our revised manuscript.
>
> ### W5
> We thank the reviewer for this valuable suggestion. In the revised appendix, we will include the loss curves for the real-data (WikiText) training process https://ibb.co/1GZMG6Fw. As shown in the provided link, the WikiText training trajectory indeed prominently exhibits the characteristic multi-stage loss curve (sharp drop, plateau, secondary drop).
>
> ### Q1
> We thank the reviewer for this insightful question. Following your suggestion, we increased the stochasticity in our synthetic experiments by reducing the batch size while keeping the total number of training steps constant. The corresponding results are shown in https://ibb.co/3YpwSDCw.
>
> We find that the saddle-to-saddle picture gradually degrades as randomness increases. Specifically, at a moderately reduced batch size (e.g., bs=256), the pattern remains highly robust. However, when the batch size is drastically reduced (e.g., bs=32), the high mini-batch noise begins to disrupt the dynamics.  **The early stages (Stages I to IV) exhibit strong robustness** and still clearly emerge even at bs=32, whereas the later cycles become increasingly noisy and less clearly separated. We have added these ablation results and discussions to the revised appendix.
>
> ### Q2
> We completely understand your concern.  Reviewer wfbF (Weakness 1.1) raised a very similar point, highlighting the importance of this discussion. In the first-order Markov setting, positional embeddings indeed provide a strong shortcut: the model can directly identify the last position, which may weaken the frequency-driven focus-dilution pattern in attention. To examine this, we conduct new numerical experiments incorporating absolute positional embedding and relative positional embedding https://ibb.co/79LYbTZ.
>
> Empirically, we find that positional embeddings do affect the dynamics quantitatively and make the attention-side focus-dilution cycle less pronounced. However, this does not mean that the mechanism identified in our paper is an artifact. Rather, it suggests that focus-dilution is a natural learning dynamics arising from the coupling between attention and embeddings when no positional shortcut is available. Indeed, even with positional encodings, we still observe a clear multi-stage evolution and pronounced retraction phenomenon in the embeddings.

---

> > ### Author Rebuttal · Reviewer_aCoV · 2026-04-01
> >
> > All my questions and remarks have been addressed with additional simulations that corroborate further my positive evaluation of the submission. I confirm my assessment.

---

### Official Review · Reviewer_eWfc · 2026-03-15

**Soundness:** 3
**Presentation:** 3
**Significance:** 3
**Originality:** 4
**Overall Recommendation:** 4
**Confidence:** 2

**Summary:**

the paper studies how attention evolves during training of a one-layer transformer on markov data. the authors find a repeating four-stage cycle: (1) embeddings collapse to rank-one structure while attention stays frozen, (2) attention grows and focuses on high-frequency tokens, (3) embedding redistribution dilutes this focus, (4) asymmetry among rare tokens breaks degeneracy and starts the next cycle. theory is based on linearization around successive saddle points. experiments on synthetic data, wikitext, and tinystories confirm the predictions.

**Compliance With Llm Reviewing Policy:**

Affirmed.

**Key Questions For Authors:**

1/ does the focus-dilution cycle survive with multi-head attention? the rank-one condensation seems tied to single-head structure. any evidence, even empirical?
2/ have you tried observing these stages in a standard multi-layer transformer (e.g. small gpt-2), not just your simplified architecture on real data?
3/ how sensitive are the stage transitions to the choice of optimizer? theory uses gradient flow but experiments use adam. does vanilla sgd give the same four stages?
4/ is there a quantitative relationship between vocabulary size d and the number of focus-dilution cycles before convergence?

**Limitations:**

yes, adequately discussed.

**Strengths And Weaknesses:**

the core finding - that attention training is cyclical rather than monotonic - is new and well-supported. previous work explained how attention becomes sparse but not how it then diffuses. the mass-redistribution mechanism (theorem 3.7) fills this gap.
the technical execution is strong. the lyapunov-schmidt reduction for the degenerate critical point in section 3.4 is nontrivial and gives a clean explanation of how new embedding directions emerge. the chain of saddle-to-saddle transitions is a natural framework for multi-stage dynamics.

the submission addresses a major problem in understanding transformer optimization.
experiments match theory well.


weaknesses:
the setting is narrow: one layer, one head, markov data, small init, gradient flow (not sgd). the gap to real transformers is large. the real-data experiments help but still use the simplified architecture.
population-level analysis ignores stochastic noise, which could matter near saddle points in practice.

the paper does not discuss whether the cycle count or cycle duration relates to vocabulary size or data complexity in any quantitative way.

---

> ### Author Rebuttal · Authors · 2026-03-27
>
> We are grateful for the insightful review. We particularly appreciate your thought-provoking questions regarding the generalization of our findings to more complex architectures:
>
> ### Weakness 1, Q1 and Q2
> We deeply understand the reviewer's valid concerns regarding multi-layer and multi-head architectures. Theoretically analyzing a deep, multi-head model is notoriously challenging, and empirically, it is difficult to isolate and trace a specific attention head in a multi-layer setting. However, we can still reliably track the overall emergence and retraction of new directions by observing the embedding norms (similar to Figure 2D).
>
> To test this, we conducted new experiments on no-layernorm transformer  (8L1H, 8L2H, 8L8H). The results are shown in (https://ibb.co/77QY5Yf). Interestingly, we observed the following: The 8L1H model preserves the distinct norm growth-and-retraction phenomenon remarkably well. We attribute this to the residual connections (ResNet structure), which help propagate the sequential learning dynamics across layers. As the number of heads increases, the phenomenon becomes less visually pronounced. This aligns with theoretical intuition: the rank-one condensation is tightly tied to the single-head structure. Multi-head attention intrinsically increases the rank of the attention matrix, meaning the dynamics must now be analyzed on a high-rank manifold rather than a simple rank-one trajectory.
>
> ### Q3
> We primarily used Adam in our main experiments to accelerate training, as the focus-dilution can take a significant amount of time to fully unfold. However, to address your question regarding the vanilla SGD optimizer (which is closer to our gradient flow theory), we have provided the corresponding experimental results in https://ibb.co/r2V4wJyg  (similar to Figure 2(C, D)).
>
> As expected, the the focus-dilution cycles is still clearly observable. This confirms that our theoretical predictions hold true under vanilla SGD and are not merely artifacts of the Adam optimizer.
>
> Relatedly, to address the reviewer’s concern that stochastic noise may affect saddle-to-saddle dynamics, we further increased the stochasticity in our synthetic experiments by reducing the batch size while keeping the total number of training steps fixed. The corresponding results are shown in https://ibb.co/278t9GVF. We find that the saddle-to-saddle picture gradually degrades as randomness increases. At a moderately reduced batch size (e.g., bs=256), the pattern remains highly robust, while at a much smaller batch size (e.g., bs=32), the later cycles become increasingly noisy and less clearly separated. Importantly, the early stages (Stages I–IV) remain clearly visible even at bs=32, indicating that the core stage-transition mechanism is robust to mini-batch noise. We have added these ablation results and discussion to the revised appendix.
>
>
> ### Weakness 2 and Q4
> Yes, there is a distinct quantitative relationship. Under our theoretical framework, assuming an ideal setting with D transition states (vocabulary size), the system will sequentially undergo exactly D−1 focus-dilution cycles before full convergence.
>
> However, emperically, observing all D−1 distinct cycles perfectly can be difficult. This is due to a combination of factors: the randomness of initialization, the gap between gradient desent and gradient flow, and the accumulation of minor errors over long training steps. As a result, while the early stages follow the theoretical D−1 sequence strictly, the later stages often experience overlapping or aliasing between the remaining tokens. We will clarify this theoretical bound and its practical caveats in the revised manuscript.

---

### Official Review · Reviewer_wfbF · 2026-03-19

**Soundness:** 3
**Presentation:** 3
**Significance:** 3
**Originality:** 3
**Overall Recommendation:** 5
**Confidence:** 3

**Summary:**

This paper identifies a recurring focus-dilution cycle in the training dynamics of attention in a one-layer transformer trained on Markov chain data via gradient flow. The authors decompose this cycle into four stages: (1) initial condensation, where embeddings and projection matrices $W_0, W_1$ rapidly collapse to a rank-one structure while the attention parameters $W_Q, W_K$ remain near zero. (2) growth of attention, where $W_Q, W_K$ align with the unstable eigendirection of the Jacobian at a saddle point, inducing a frequency-driven bias toward high-frequency tokens; (3) dilution of attention, caused by a mass-redistribution mechanism in the embeddings that undermines the earlier focus and (4) emergence of a new embedding direction, where asymmetries among low-frequency tokens break a degenerate critical point on the rank-one manifold, opening new directions for $W_0$ and restarting the cycle. The theoretical analysis is built on stage-wise linearization around successive critical points in the population gradient flow. Experiments on synthetic Markov data and comparisons on WikiText and TinyStories are provided.

**Compliance With Llm Reviewing Policy:**

Affirmed.

**Final Justification:**

My main concerns were whether the cycle appears with skip connections and positional encodings, technical gaps in stage transitions, and data generality. The rebuttal addressed these: skip connections preserve the cycle, the new rigorous transition analysis fills the most significant technical gap, and the lag-2 experiment shows generality beyond lag-1. Positional encodings weaken the cycle on first-order Markov data, but this reflects the triviality of fixed-lag attention rather than a fundamental limitation. The mathematical framework remains a solid contribution to training dynamics.

**Key Questions For Authors:**

1. **Positional encodings and skip connections:** Can you provide any theoretical argument or experimental evidence that the focus-dilution cycle persists when positional encodings or skip connections are added? An experiment with a residual connection on your synthetic data would be informative. If the cycle disappears, how do you interpret the significance of the current findings?

2. **Transitions between stages:** Is it possible to provide any bound on the trajectory during the transition from one linearization window to the next? For instance, after the $\Theta(\log(1/\varepsilon))$ window around $\theta = 0$, can you guarantee the trajectory is within the linearization radius of $\theta_c^1$? Even a numerical verification would strengthen the argument.

3. **Off-manifold error:** How sensitive are the Stage II-III results to the trajectory not being exactly on the rank-one manifold $\mathcal{W}$? Can you bound the off-manifold component and show it remains negligible throughout these stages?

4. **Generalization of Stage IV beyond $d = 3$:** For the synthetic experiments with $d = 4$ tokens, does the Lyapunov-Schmidt analysis still apply? Can you provide numerical evidence (eigenvalue spectra of the Hessian at the degenerate critical point) confirming the two-scale separation for $d > 3$?

5. **Multiple high-frequency tokens:** What happens when the stationary distribution has two or more tokens with comparably high frequency? Does the focus stage still pick a single dominant token, or does the dynamics qualitatively change?

**Limitations:**

yes

**Strengths And Weaknesses:**

### Strengths

**S1 (Originality).** The key novelty lies in studying the co-evolution of embeddings ($W_0, W_1$) and attention parameters ($W_Q, W_K$) simultaneously. As far as I am aware, most prior theoretical work on transformer training dynamics either fixes the embedding and studies attention weights, or vice versa. This joint treatment reveals the feedback loop between focus and dilution that would be invisible under either isolation. The identification of the cyclical pattern itself is an interesting insight.

**S2 (Soundness of the mathematical framework).** The stage-wise linearization approach around successive critical points is technically sound. The paper identifies the block-diagonal structure of the Jacobian at the first critical point, showing the decoupling of the $(W_0, W_1)$ and $(W_Q, W_K)$ subsystems, and the analysis of the mass-redistribution mechanism via the reduced dynamics on the rank-one invariant manifold is interesting. The Lyapunov-Schmidt reduction for analyzing the degenerate critical point is technically non-trivial.

**S3 (Clear stage decomposition).** The decomposition into four stages with explicit critical points and transitions provides a clear narrative. Figure 1 gives a helpful overview, and Figure 2 provides convincing empirical validation of each predicted stage on synthetic data.

### Weaknesses

**W1 (Missing skip connections and positional encodings).** The model studied omits positional encodings and skip connections. These components could change the learning problem in ways that could trivialize or eliminate the focus-dilution cycle:

- *Positional encodings:* For Markov data, the next token depends only on the current (last) token. With positional encodings, there is a trivial solution where the attention learns to attend exclusively to the last position, bypassing any frequency learning entirely. The focus-dilution cycle may threfore be an artifact of the model's inability to use positional information, not an intrinsic property of attention.
- *Skip connections:* With a residual connection, the identity mapping already provides direct access to the input token representation at every position. In particular, the model can already produce non-trivial predictions through the skip path alone, so the rank-one condensation of $W_0$ in Stage I might not occur. More critically, the skip connection would make the attention block's contribution additive and small initially, potentially eliminating the saddle-point structure that drives the entire cycle.

The paper does not discuss these omissions or argue why the discovered dynamics would survive their inclusion. It therefore remains unclear whether focus-dilution is a property of attention learning or of this specific minimal architecture.

**W2 (Technical gaps in the stage-wise analysis).** While the individual stage analyses are sound, some technical gaps weaken the overall argument:

- *Uncontrolled transitions between stages.* Each stage is analyzed via local linearization around a critical point, valid for a $\Theta(\log(1/\varepsilon))$ time window. However, the paper dosen't rigorously prove that the trajectory exiting one linearization window enters the basin of the next critical point. For example, after Stage I, the trajectory is predicted to be near $\theta_c^1$, but the transition from the linearization around $\theta = 0$ to the linearization around $\theta_c^1$ involves a gap where neither linearization is valid. The paper relies on empirical evidence to bridge these gaps, which weakens the theoretical contribution.

- *Approximate entry into the rank-one manifold $\mathcal{W}$.* Proposition 3.6 proves that $\mathcal{W}$ is invariant, but the trajectory after Stage I only *approaches* $\mathcal{W}$ asymptotically -- it never exactly enters it. The reduced dynamics (Eq. 20) and all subsequent Stage II-III analysis assume exact membership in $\mathcal{W}$. The approximation error from not being exactly on $\mathcal{W}$ is never bounded. How large can the off-manifold component be, and does it affect the qualitative conclusions?

- *The reduced dynamics (Eq. 20) implicitly assume $\lambda_Q = \lambda_K$.* In the rank-one parametrization $W_Q = \lambda_Q \alpha_1 \tilde{\alpha}_1^\top$, $W_K = \lambda_K \alpha_1 \tilde{\alpha}_1^\top$, the equation $\dot{\eta} = -2\eta  \gamma^\top \frac{\partial \mathcal{L}}{\partial \Phi} \gamma$ is correct only when $\lambda_Q = \lambda_K$ . This equality is preserved by the dynamics during the focus phase, but the assumption should be explicitly stated.

- *Theorem 3.7 lacks error bounds.* The mass-redistribution result is derived as a first-order perturbation result, but it is stated in the main text without error bounds or a validity time window. Since this is a linearized result, it should hold only while higher-order terms remain negligible. The paper should clarify the regime of validity.

**W3 (Symmetry assumptions in Stage IV analysis).** The analysis of the symmetry-breaking mechanism (Section 3.4) restricts to $d = 3$ with a specific perturbation structure. While the paper acknowledges this, it is unclear how the mechanism generalizes.

**W4 (Markov data specificity).** The specific transition matrix $P = \lambda I + (1 - \lambda) \mathbf{1} \pi^\top$ and the two-group structure (one high-frequency token vs. symmetric low-frequency tokens) are designed to produce the focus-dilution cycle. It is unclear whether the phenomenon persists under transitions with richer spectral structure or distributions where multiple tokens have comparable frequencies. Related concurrent works studying transformer learning on Markov data with different structures,  random walks on circles (Shi & Cao, "Towards Understanding Transformers in Learning Random Walks", NeurIPS 2025) and interleaved Markov chains where the relevant token is not at lag 1 (D'Angelo et al., "Selective Induction Heads: how transformers select causal structures in context", ICLR 2025) could be discussed; in particular, does the focus-dilution cycle persist when the relevant token is not the most recent one or depends on a context-dependent lag?

---

> ### Author Rebuttal · Authors · 2026-03-27
>
> We sincerely thank the reviewer for the constructive feedback.
> ### W1 & Q1
> We added three experiments with residual connection (https://ibb.co/9Hv2QRZN), learnable absolute positional embeddings(APE) and RoPE (https://ibb.co/3mkvFsbv) and drew like Fig.2.
>
> For skip connections, the multi-stage focus-dilution structure remains clearly visible, showing that residual connections do not qualitatively alter the mechanism studied in this paper. Regarding condensation, prior work [1] shows that it persists even in the presence of residual connections.
>
> For positional embeddings, in first-order Markov setting, they provide a strong shortcut and thus weaken the focus-dilution cycle in attention. However, this does not make the mechanism an artifact. Instead, it shows that focus-dilution naturally arises from attention-embedding coupling when no positional shortcut is available. Notably, we still observe a multi-stage learning process in the embeddings.
>
> We will revise the paper and include the experiments to clarify the boundary of applicability of our conclusions.
>
> ### W2.1 & Q2
> We add a rigorous analysis from $\theta = 0$ to $\theta = \theta_c^1$ in revision. The key ingredient is no longer linearization, but a refined estimate that controls the normal directions and the attention parameters throughout the transition interval. Similar issues were also discussed in [2,3].
>
> Specifically, we decompose the parameters into the condensation directions and the normal directions. Let
> $$
> q=\frac{\pi}{\Vert\pi\Vert}, \qquad
> u=\frac{\pi-\frac{1}{d}\mathbf{1}}{\Vert \pi-\frac{1}{d}\mathbf{1}\Vert},
> $$
> and decompose the outer-layer parameters as
> $$
> W_0 = qq^\top W_0 + R_0, \qquad
> W_1 = W_1 uu^\top + R_1.
> $$
> We prove that the remainders satisfy
> $$
> \frac{d}{dt}\bigl(\Vert R_0\Vert+\Vert R_1 \Vert\bigr)
> \lesssim
> \Vert W_Q \Vert^2+\Vert W_K \Vert^2+\Vert R_0\Vert^2+\Vert R_1 \Vert^2,
> $$
> so the normal components remain controlled throughout the transition stage. As a result, the leading-order dynamics are confined to the reduced subsystem governed by $q^\top W_0$ and $W_1u$.
>
> Defining further
> $$
> z = q^\top W_0 W_1 u,
> $$
> we reduce the system to an effective scalar dynamics
> $$
> \dot z \approx 2\Vert \pi \Vert\sqrt{\frac{d}{d-1}} z
> \left(
> \pi_1-\frac{1}{1+(d-1)e^{-c_\pi z}}
> \right).
> $$
> We then show that the unique nonzero root of the right-hand side is exactly corresponding to the critical point $\theta_c^1$. This allows us to rigorously conclude that the trajectory enters an O$(\delta)$-neighborhood of $\theta_c^1$ within an explicit and controlled time.
>
> ### W2.2 & Q3
> We agree that the trajectory does not enter the invariant manifold $\mathcal{W}$ exactly. Our Stage II--III analysis is local and only requires proximity to $\mathcal{W}$, supported by previous cross-stage analysis and subsequent linearization. Near $\mathcal{W}$, linearization suggests that the off-manifold component is expected to remain higher-order on the time scale considered here, so it does not affect the qualitative conclusions. A long-time bound requires finer normal control and is left for future work. This is also consistent with Fig. 2A and Fig.2 B1/ B2, where the off-manifold component appears higher-order relative to the dominant rank-one mode.
>
> ### W2.3
> We agree that the dilution-stage analysis implicitly assumes $\lambda_Q = \lambda_K$. We will state this assumption explicitly in the revision.
>
> ### W2.4
> We agree that Theorem 3.7 is first-order perturbative, and we will clarify its regime of validity in the revision.
>
> ### W3 & Q4
> The symmetry breaking mechanism in Section 3.4 can generalize to $d≥4$ within our Lyapunov-Schmidt framework; $d>3$ only introduces additional technical complexity rather than new mechanisms. We leave higher-dimensional extensions and experimental validation to future work.
>
> ### W4 & Q5
> To address the possibility that the relevant token is not the most recent one, we add an experiment on a lag-2 interleaved Markov chain.
>
> Specifically, we construct two independent chains: $\{x_i\}$,  $x_i\in\{0,1,2\}$, and $\{y_i\}$, $y_i\in\{4,5,6\}$. We then interleave them as
> $$
> (x_1,y_1,\dots,x_n,y_n)\quad(y_1,x_1,\dots,y_n,x_n).
> $$
> In this setting, predicting the current token comes from the one before the previous one, rather than the previous one.
>
> Our experiments show that the multi-stage learning on the embedding still persists (embedding norm: https://ibb.co/FkqzVRQZ). In particular, the first stage is largely consistent with our paper. In the later stages, however, the dynamics change: since tokens $0$ and $4$ are both highest-frequency tokens, they simultaneously grow new directions.
>
> This experiment suggests that the mechanism is not limited to the lag-1 setting, although its detailed behaviour depends on the structure of the task. We hope this additional experiment helps address the concern raised in Q5.
>
> [1]https://arxiv.org/abs/2510.06954
>
> [2]https://arxiv.org/abs/2503.06982
>
> [3]https://arxiv.org/abs/2410.20119

---

> > ### Author Rebuttal · Reviewer_wfbF · 2026-04-03
> >
> > The rebuttal meaningfully strengthens the paper. The skip connection experiment confirms the cycle is robust to residual pathways, the rigorous stage transition analysis fills the most substantive technical gap from the original submission, and the lag-2 experiment demonstrates the dynamics extend beyond the simplest Markov setting. Regarding positional encodings: while PEs weaken the cycle on first-order Markov data, this is expected, as attending to a fixed lag is trivially solved by position. The more interesting regime is when the relevant context is not at a predictable position, where the attention-embedding coupling that the paper analyzes becomes the primary learning mechanism. The mathematical framework remains a solid contribution to the training dynamics literature. I raise my score to accept.

---

### Decision · Program_Chairs · 2026-04-30

**Decision:**

Accept (spotlight)

**Comment:**

This paper identifies a recurrent focus-dilution cycle in the training dynamics of attention mechanisms, providing a rigorous gradient flow analysis in a one-layer Transformer on Markovian data and validating the theory on both synthetic data and standard NLP benchmarks. All reviewers ultimately recommend acceptance with strong scores, praising the originality and depth of the analysis, and the clear empirical support. Initial concerns regarding architectural simplifications (skip connections, positional encodings), some technical points, and experimental reproducibility were convincingly addressed in the rebuttal, leading all reviewers to raise or confirm their positive assessments. The paper makes a substantial contribution to the theory of Transformer training dynamics and is ready for acceptance.